# FIGMA2CODE: AUTOMATING MULTIMODAL DESIGN TO CODE IN THE WILD

**Yi Gui**[1][*], **Jiawan Zhang**[1][*], **Yina Wang**[1], **Tianran Ma**[1], **Yao Wan**[1][†], **Shilin He**[2],
**Dongping Chen**[3], **Zhou Zhao**[4], **Wenbin Jiang**[1], **Xuanhua Shi**[1], **Hai Jin**[1], **Philip S. Yu**[5]
[1]Huazhong University of Science and Technology, [2]Independent Researcher
[3]University of Maryland, [4]Zhejiang University, [5]University of Illinois Chicago
`wanyao@hust.edu.cn`

## ABSTRACT

Front-end development constitutes a substantial portion of software engineering, yet converting design mockups into production-ready *User Interface* (UI) code remains tedious and costly. While recent work has explored automating this process with *Multimodal Large Language Models* (MLLMs), existing approaches typically rely solely on design images. As a result, they must infer complex UI details from images alone, often leading to degraded results. In real-world development workflows, however, design mockups are usually delivered as Figma files—a widely used tool for front-end design—that embed rich multimodal information (e.g., metadata and assets) essential for generating high-quality UI. To bridge this gap, we introduce FIGMA2CODE, a new task that advances *design-to-code* into a multimodal setting and aims to automate *design-to-code* in the wild. Specifically, we collect paired design images and their corresponding metadata files from the Figma community. We then apply a series of processing operations, including rule-based filtering, human and MLLM-based annotation and screening, and metadata refinement. This process yields 3,055 samples, from which designers curate a balanced dataset of 213 high-quality cases. Using this dataset, we benchmark ten state-of-the-art open-source and proprietary MLLMs. Our results show that while proprietary models achieve superior visual fidelity, they remain limited in layout responsiveness and code maintainability. Further experiments across modalities and ablation studies corroborate this limitation, partly due to models' tendency to directly map primitive visual attributes from Figma metadata.[1]

## 1 INTRODUCTION

Front-end development is a cornerstone of modern software engineering, often consuming a substantial share of the overall workload (Wu et al., 2024b). In practice, designers deliver design mockups, while developers translate them into UI code that reflects the intended visual appearance—a workflow that remains costly and labor-intensive. This has long motivated the pursuit of fully automated *design-to-code*, a vision with enduring academic and practical significance. With the recent emergence of powerful MLLMs, this long-standing vision is no longer merely conceptual but is now on the horizon.

Inspired by earlier machine learning–based approaches (Beltramelli, 2017; Robinson, 2019), a flurry of recent studies has advanced the *design-to-code* task by leveraging MLLMs. Some works introduce large-scale synthetic (Laurençon et al., 2024) or real-world datasets (Gui et al., 2025a) to train or finetune MLLMs (Yun et al., 2024), enabling them to convert design images into UI code. Others contribute benchmarks that aim at systematically evaluating and pushing the boundaries of MLLMs on the *design-to-code* task. Among them, Design2Code Si et al. (2024) presents the first real-world evaluation dataset together with dedicated metrics for assessing UI code generation, and explores different approaches to this task, such as direct prompting and text-augmented prompting. In parallel,

---

[*]Equal Contribution, [†]Corresponding Author
[1]The codebase and dataset are available at ⬡ GitHub and 🤗 Hugging Face.

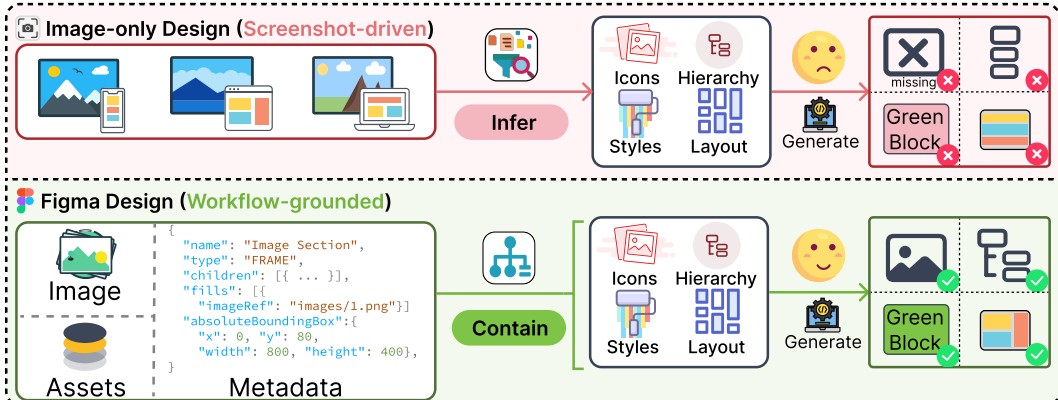

Figure 1: Comparison of UI generation from image-only and multimodal (Figma) design.

several studies have also investigated related challenges, including UI code repair and editing (Xiao et al., 2025), as well as code generation from complex layouts (Gui et al., 2025b).

Despite the promising performance reported in previous studies, there remains a substantial gap before MLLM-based automated *design-to-code* can be applied in industrial-scale development. Most previous efforts have focused on UI code generation from image-only designs, where the typical application scenario invokes replicating designs from screenshots of existing UI pages, similar to *screenshot-to-code* (Raja, 2024). As illustrated in Figure 1, this approach implicitly requires inferring UI details—including icons, hierarchy, styles, and layout—directly from design images, which has long been a challenging problem (Wu et al., 2021). Moreover, the absence of essential assets such as background images and button icons often leads to degraded UI code. In a standard workflow of software development, design mockups are not limited to static images but also include metadata (e.g., hierarchy, layout, and styles) and assets (e.g., icons and background images), most prominently as Figma (Figma, Inc.) files. These files contain rich information otherwise missing in image-only designs, thereby enabling higher-quality UI code generation.

To advance toward fully automated UI code generation, we propose a new task, FIGMA2CODE, aiming to move *design-to-code* research beyond image-only approaches toward a more realistic and industry-relevant setting grounded in real-world scenarios. We begin by collecting a large corpus of design files from the Figma community, spanning diverse platforms and content, where each sample consists of a design image paired with its metadata files in JSON format. Each file is divided into independent pages, which are subsequently refined through heuristic filtering and manual review. We further integrate dependent resources and refine the metadata through structure pruning and abstraction. In parallel, annotators label attributes such as complexity, platform, and quality, while an MLLM is employed to categorize samples by content. Through this pipeline, we obtain a base dataset of 3,055 samples. We then perform stratified sampling followed by expert review, yielding 213 high-quality and diverse samples that form the FIGMA2CODE dataset.

Using this dataset, we systematically evaluate both open-source and proprietary MLLMs on the FIGMA2CODE task with metrics covering visual fidelity and code quality in terms of layout responsiveness and code maintainability. Results show that while proprietary models achieve significantly higher visual fidelity, they still lag in code quality, leaving a substantial gap from industrial-grade UI code. We further assess UI code generation across different modalities and methods, conduct ablation studies on five key components in Figma metadata, and perform qualitative case analyses. The results indicate that Figma metadata substantially enhances visual fidelity in the task, introduces challenges for responsiveness and maintainability—partly due to models' tendency to directly map absolute coordinates and primitive visual attributes from the metadata.

Overall, the key contributions of this work are threefold:

- **New Problem and Dataset.** We introduce a new task, FIGMA2CODE, advancing *design-to-code* research beyond image-only methods toward a multimodal, industry-relevant setting. To this end, we build the FIGMA2CODE dataset, comprising 213 diverse, high-quality samples.

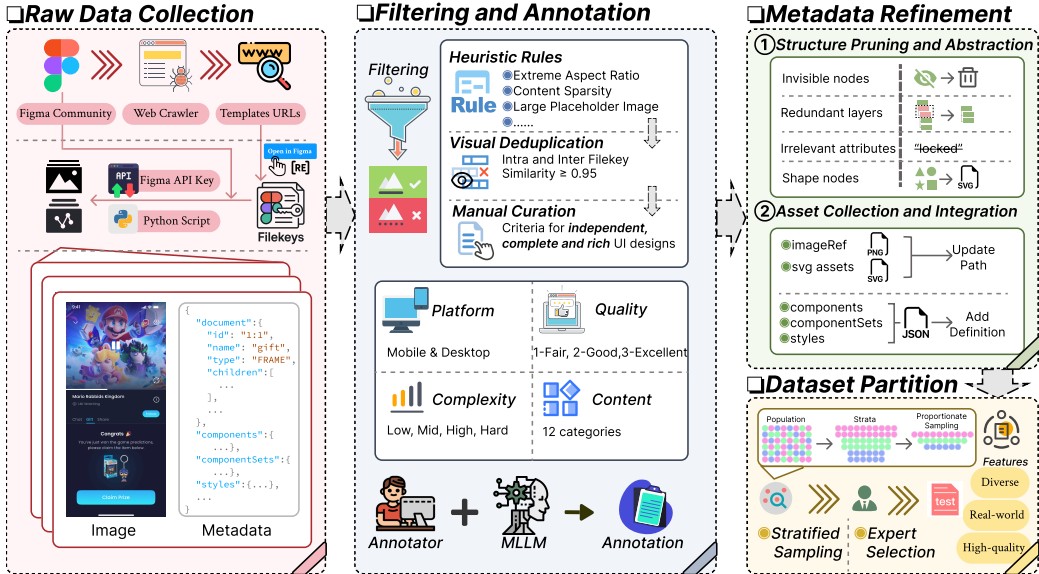

Figure 2: The pipeline of constructing the FIGMA2CODE dataset.

- **Comprehensive Benchmark.** We are the first to establish a systematic evaluation framework that assesses not only visual fidelity but also code quality for multimodal *design-to-code* generation, and comprehensively benchmark state-of-the-art MLLMs on the task.
- **Findings and Implications.** Our studies indicate that current MLLMs struggle to balance visual fidelity with code quality in the FIGMA2CODE task, a challenge arising from integrating Figma metadata. These findings highlight opportunities for future research in this domain.

## 2 PROBLEM FORMULATION

The FIGMA2CODE task aims to convert Figma design artifacts into executable user interface (UI) code. Unlike previous *design-to-code* (Si et al., 2024) settings that rely solely on visual inputs, Figma designs provide multimodal information, including ① a JSON metadata file describing the hierarchical structure and properties of UI elements, ② associated design assets such as icons and images, and ③ a rendered screenshot of the full design. The objective is to generate corresponding UI code (e.g., HTML/CSS, React, Flutter) that meets three essential criteria: **Visual Fidelity**, faithfully reproducing the intended appearance of the original design; **Layout Responsiveness**, enabling graceful adaptation across devices and screen sizes; and **Code Maintainability**, ensuring structural qualities that facilitate modularity, reuse, and long-term evolution.

Formally, the task takes as input $I = (M, A, V)$, where $M$ denotes the JSON metadata, $A$ the set of design assets (e.g., icons and images), and $V \in \mathbb{R}^{H \times W \times 3}$ the screenshot of the Figma design. The output is a codebase $C$ in a specific UI language or framework, with $\text{Render}(C)$ denoting its rendered output. The objectives are threefold: ① minimize the perceptual dissimilarity $D(V, \text{Render}(C))$, ② maximize the *Layout Responsiveness Score (RS)*, and ③ maximize the *Code Maintainability Score (MS)*. The joint optimization problem is:

$$\hat{C} = \arg\max_C \Big[ -\alpha \cdot D(V, \text{Render}(C)) + \beta \cdot RS(C) + \gamma \cdot MS(C) \Big],$$

where $\alpha, \beta, \gamma \geq 0$ balance the three objectives.

## 3 FIGMA2CODE: THE DATASET

### 3.1 DATASET CONSTRUCTION

This work aims to construct a high-quality and diverse Figma design dataset derived from real-world design practices to support and advance research in automatic front-end code generation. Figure 2

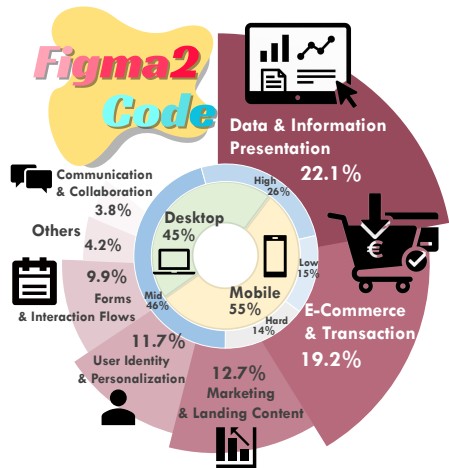

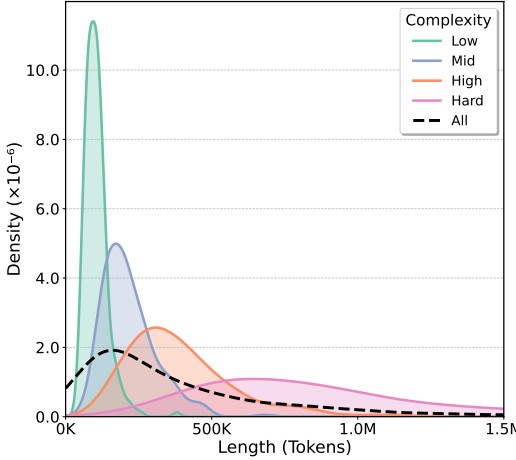

Figure 3: Dataset statistics over different content, complexities and platforms.

Figure 4: Estimated token distribution over different complexities.

illustrates our meticulous data construction pipeline (details in Appendix C), which consists of four main stages: raw data collection, filtering and annotation, metadata refinement and dataset partition.

**Raw Data Collection.** To ensure that our dataset reflects real-world design practices, we select the Figma community as our primary data source. The Figma community is an open and widely used platform where professional designers and developers share real projects, templates, and UI components, making it a natural and reliable source for authentic design artifacts. To guarantee diversity, we perform balanced crawling across seven categories—including blogs, advertisements, and mobile applications. This process yields an initial collection of about 2,100 design files. Since each design file typically contains multiple pages, we further decompose them into about 30,000 independent pages, and use the page as the unit of our dataset. These pages provide a rich and varied foundation for the subsequent filtering and refinement process.

**Data Filtering and Annotation.** The raw corpus, collected from the open Figma community, inevitably contains many irrelevant or low-quality samples, as files are contributed by diverse users and often include blank pages, non-UI artifacts (e.g., presentations), or overly simplistic drafts. In addition, pages derived from the same design file were often highly redundant. To distill a dataset that is both high-quality and diverse, we design a rigorous multi-stage filtering pipeline (Details in Appendix C.2): ① Heuristic rule-based filtering, where we apply simple structural constraints (e.g., discarding pages with extreme aspect ratios or too few elements) to automatically remove obviously unsuitable cases, drastically reducing the data volume; ② Visual deduplication, where we generate CLIP embeddings for each page screenshot and remove perceptually similar pages (cosine similarity > 0.95), ensuring that each retained sample contributes unique visual information; and ③ Manual filtering, where design experts review the remaining pages following clear guidelines, eliminating incomplete or semantically trivial designs while retaining only coherent and meaningful UI layouts. This multi-stage pipeline effectively ensures both the quality and diversity of the final dataset, providing a robust foundation for subsequent benchmarking.

Following the filtering process, design experts perform multi-dimensional annotations on the qualified pages, including their platform, structural complexity, and design quality. Furthermore, to enrich the data with additional semantic information for finer-grained analysis and applications, we leverage a large language model to assist in annotating the content of each page, categorizing them into 12 major classes to describe their primary purpose. This human-in-the-loop process ultimately yields 3,055 high-quality, richly annotated design pages.

**Metadata Refinement.** The raw design metadata from Figma is extremely verbose, containing editor-specific properties and redundant structures that add substantial overhead and hinder effective code generation. Such artifacts not only inflate the representation, but also obscure the essential design semantics required for model learning. To transform this raw data into a clean, lightweight, and structured representation, we perform a two-step post-processing procedure.

Table 1: Overview of existing *design-to-code* benchmark datasets.

| | Size | Source | Design Format | | | Dimensions | | |
|---|---|---|---|---|---|---|---|---|
| | | | Image | Meta | Assets | Platform | Complex | Content |
| pix2code (Beltramelli, 2017) | 1,742 | Synth. | ✓ | ✗ | ✗ | ✓ | ✗ | ✗ |
| Design2Code (Si et al., 2024) | 484 | Real | ✓ | ✗ | ✗ | ✗ | ✗ | ✗ |
| IW-Bench (Guo et al., 2025) | 1,200 | Real | ✓ | ✗ | ✗ | ✗ | ✓ | ✗ |
| WebCode2M (Gui et al., 2025a) | 768 | Real | ✓ | ✗ | ✗ | ✗ | ✓ | ✗ |
| MRWeb (Wan et al., 2024a) | 500 | Mixed | ✓ | ✗ | ✓ | ✗ | ✓ | ✗ |
| CC-HARD (Gui et al., 2025b) | 128 | Real | ✓ | ✗ | ✗ | ✗ | ✗ | ✗ |
| **FIGMA2CODE** (Ours) | 213 | Real | ✓ | ✓ | ✓ | ✓ | ✓ | ✓ |

▷ Structure Pruning and Abstraction. This step aims to streamline the raw Figma JSON and distill the core UI structure and attributes. Specifically, we perform a series of refinement operations: removing nodes that do not contribute to the final rendering (e.g., invisible or fully occluded elements), flattening redundant container hierarchies, and filtering out metadata attributes specific to the Figma editor that are irrelevant to UI reproduction. In addition, groups of vector nodes identified as composing the same icon are merged and abstracted into a single complete icon, which we export as an SVG file and reintegrate into the metadata as a rectangle node with a local image reference. These refinements substantially reduce the sequence length of the structured representation while preserving its essential semantics.

▷ Asset Collection and Integration. UI designs frequently rely on external assets like bitmap images and vector icons. To create fully self-contained data samples, we identify and localize all external dependencies. This process involves batch downloading necessary images, components, and style resources, deduplicating images with identical content, providing local relative paths for images within the data structure, and integrating components and style definitions, thereby ensuring the completeness and portability of each sample.

**Dataset Partition.** To support both robust evaluation and future research, we partition our base dataset which comprises 3,055 samples, into a benchmark dataset and an auxiliary dataset. The benchmark dataset is constructed through stratified sampling followed by expert selection (Details in Appendix C.5), ensuring its distribution across key dimensions—platform, complexity, and content category—mirrors that of the full dataset. From this stratified pool, designers further select representative cases to ensure both representativeness and quality. This process yields a high-quality benchmark dataset of 213 samples, while the remaining 2,842 auxiliary samples are also released to the research community.

## 3.2 DATA CHARACTERISTICS

Compared with existing *design-to-code* benchmarks (Table 1), our dataset is built from real design mockups and provides design images, metadata, and assets, all of which are essential for generating faithful, high-quality UI code. We further supply annotations along three key dimensions (i.e., platform, complexity and content) to support future research. Overall, the dataset has the following two key characteristics: ❶ Rich diversity across multiple dimensions. As shown in Figure 3, it covers scenarios such as data visualization, presentation, e-commerce, marketing, user identity, interaction flows, and communication, while maintaining a balanced distribution across semantic categories that reflect real-world practices. In terms of complexity, it spans a clear hierarchy from simple cases to mid-level designs and highly complex interfaces, with token length distributions (Figure 4) highlighting these differences. ❷ High quality. All samples are collected from the Figma community and undergo heuristic filtering, visual deduplication, and expert inspection to eliminate noise and non-UI pages, followed by metadata refinement. Each page is a self-contained sample with complete assets, styles, and components, and multiple vector instances of the same icon are consolidated into a single exported SVG file. This construction addresses layout and resource gaps in prior work and establishes a solid foundation for generating high-quality UI. Please see more details in Appendix D.

Table 2: Benchmarking the multimodal code generation capabilities of state-of-the-art MLLMs from leading vendors when provided with design images and metadata as input.

| Model | Visual Fidelity | | Responsiveness (%) | | Maintainability (%) | |
|---|---|---|---|---|---|---|
| | VES (↑) | MAE (↓) | RUR (↑) | APR (↓) | STR (↑) | AVU (↓) |
| Llama 4 Maverick (MetaAI, 2025) | 0.6902 | 0.2266 | 3.30 | 6.09 | 25.28 | 7.71 |
| Llama 4 Scout (MetaAI, 2025) | 0.6184 | 0.2375 | 4.72 | **1.42** | **35.76** | 0.87 |
| Qwen 2.5 VL (Alibaba, 2025) | 0.6516 | 0.2120 | 4.40 | 2.66 | 29.54 | **0.15** |
| ERNIE 4.5 424B VL (Baidu, 2025) | 0.6983 | 0.2198 | **4.81** | 2.83 | 32.03 | 2.06 |
| Nova Pro v1 (Amazon, 2025) | 0.5993 | 0.2496 | 4.59 | 3.84 | 29.28 | 1.53 |
| Gemini 2.5 Pro (Google, 2025) | 0.8110 | 0.1936 | 4.43 | 10.51 | 28.98 | 25.46 |
| Grok4 (xAI, 2025) | 0.7997 | **0.1822** | 2.30 | 31.09 | 13.88 | 49.97 |
| Claude Opus 4.1 (Anthropic, 2025) | 0.7761 | 0.1911 | 1.05 | 9.62 | 19.79 | 23.29 |
| GPT-4o (OpenAI, 2024) | 0.7405 | 0.2227 | 3.72 | 2.94 | 25.21 | 2.46 |
| GPT-5 (OpenAI, 2025) | **0.8405** | 0.1874 | 1.73 | 14.35 | 15.37 | 37.72 |

## 4 BENCHMARKING ON FIGMA2CODE

### 4.1 EVALUATION METRICS

Even within the same UI technology stack (e.g., HTML/CSS variants on the web or declarative frameworks on mobile), implementations of a visually identical UI can differ substantially. To accommodate this diversity, the evaluation relies on **reference-free metrics**, using the design image as the primary ground truth while incorporating constraints on responsiveness and maintainability.

**Visual Fidelity.** We evaluate the visual fidelity of the generated UI code with respect to the original design mockups along two dimensions: high-level semantic similarity and low-level pixel similarity. For semantic similarity, we compute **Visual Embedding Similarity (VES)** via the cosine similarity between the embeddings of the generated page screenshot $I$ and the original design image $\hat{I}$, expressed as $\cos(\text{Encode}(I), \text{Encode}(\hat{I}))$. We adopt DINOv2 (Oquab et al., 2023) as the image encoder instead of OpenCLIP (Cherti et al., 2023), since it is a self-supervised vision foundation model that learns robust and transferable visual representations, surpassing OpenCLIP on most image- and pixel-level benchmarks. For pixel-level similarity, we employ the normalized **Mean Absolute Error (MAE)** of the generated page screenshot and the design image, which has been shown in a prior study (Wan et al., 2024a) to correlate strongly with human perceptual preferences.

**Layout Responsiveness.** Responsiveness is a fundamental requirement for front-end code quality, as modern user interfaces are expected to adapt seamlessly across heterogeneous devices (desktop, tablet, mobile) and varying screen resolutions. Inspired by (Parlakkılıç, 2021; Bhanarkar et al., 2023), we primarily assess the responsiveness along two dimensions: ① the **Relative Unit Ratio (RUR)** (e.g., *%*, *em*, *rem*, *vw*/*vh*, *fr*) used in layout-related CSS properties, which captures the intrinsic scalability of the design; and ② **Absolute Positioning Ratio (APR)**, the prevalence of out-of-flow positioning (absolute/fixed) that may undermine adaptability across diverse viewports.

**Code Maintainability.** Maintainability is another key dimension of front-end code quality. Unlike static mockups, production-ready front-end code must be extensible (Gharachorlu, 2014) and semantic (Deng et al., 2022), enabling future developers to easily understand, modify, and reuse it. We primarily assess maintainability along two dimensions: ① **Semantic Tag Ratio (STR)**, defined as the proportion of semantic HTML elements (e.g., `header`, `section`, `article`) among all DOM nodes, reflecting structural clarity and accessibility; and ② **Arbitrary Value Usage (AVU)**, the proportion of class tokens using arbitrary value syntax (e.g., `w-[123px]`, `bg-[#ff0]`), indicating deviations from standardized design tokens.

### 4.2 EVALUATED METHODS

We study generation approaches across three modalities: image-only, metadata-only, and multimodal (image + metadata), to systematically evaluate the effect of different inputs. In the image-only setting, we adopt two methods from Design2Code (Si et al., 2024), namely **Direct Prompting** and

Table 3: Benchmarking different approaches on the task using the ERNIE 4.5 424B VL model under different modalities (🖼 Design image, </> Metadata, or both).

| Method | Input | Visual Fidelity | | Responsiveness (%) | | Maintainability (%) | |
|---|---|---|---|---|---|---|---|
| | | VES (↑) | MAE (↓) | RUR (↑) | APR (↓) | STR (↑) | AVU (↓) |
| Direct Prompting | 🖼 | 0.5653 | 0.2203 | 6.14 | **0.01** | 31.86 | **0.00** |
| Text-Augmented | 🖼 | 0.5683 | 0.2145 | **7.02** | **0.01** | 29.67 | **0.00** |
| Direct Prompting | </> | 0.6801 | 0.2101 | 4.97 | 5.01 | 21.46 | 6.09 |
| Template Conversion | </> | 0.6219 | **0.1205** | 0.00 | 58.0 | 0.01 | 24.94 |
| Direct Prompting | 🖼 + </> | 0.6923 | 0.2228 | 4.81 | 2.83 | **32.03** | 2.06 |
| F2CAGENT | 🖼 + </> | **0.7990** | 0.1923 | 4.69 | 13.57 | 28.57 | 16.71 |

**Text-Augmented**. In the metadata-only setting, we evaluate the FigmaToCode (Ferrari, 2025) plugin (termed **Template Conversion**) and implement a vanilla approach that instructs MLLMs to convert JSON metadata into responsive and maintainable UI code. In the multimodal setting, we further design a vanilla approach that integrates both images and metadata. For clarity, we denote all vanilla prompting approaches collectively as **Direct Prompting** (Details in Appendix F).

Building on ReAct (Yao et al., 2022), we develop a preliminary agentic workflow for the FIGMA2CODE task, termed F2CAGENT (detailed in Appendix F.2). Inspired by Ferrari (2025), F2CAGENT first converts raw Figma JSON into an Intermediate Representation (IR) by restructuring the node hierarchy, preserving design attributes, and inlining component and style dependencies. The IR is then translated into UI code via rule-based templates. Implemented without the Figma sandbox (Figma, Inc.), this step often introduces visual and structural flaws. To address these issues, we adopt a ReAct-style refinement loop, where a critic detects deficiencies and a refiner iteratively improves visual fidelity and code quality.

## 4.3 MULTIMODAL CODE GENERATION CAPABILITIES OF MLLMS

To comprehensively evaluate the multimodal code generation capabilities of MLLMs on the FIGMA2CODE task, we adopt ten flagship models from leading vendors, covering both open-source and proprietary solutions. Each model is provided with the design image and corresponding Figma metadata in JSON format and is instructed to generate UI code that reconstructs the target design while explicitly ensuring layout responsiveness and code maintainability.

**Under our benchmark and unified protocol, current MLLMs still struggle to balance visual fidelity with code quality in the FIGMA2CODE task**. As shown in Table 2, GPT-5 and Gemini 2.5 Pro achieve the highest visual fidelity, with VES scores of 0.8405 and 0.8110 and corresponding MAE values of 0.1874 and 0.1936. Claude Opus 4.1 and Grok4 also perform competitively in this dimension. By contrast, open-source models such as Llama 4 Scout, which records a VES of only 0.6184 and an MAE of 0.2375, fall far behind, highlighting the advantage of proprietary models in multimodal alignment for reconstructing design appearance. The pattern, however, reverses when considering layout responsiveness and code maintainability. Llama 4 Scout generates the cleanest and most responsive code, with an APR of 1.42% and STR of 35.76%. Grok4, despite its strong MAE of 0.1822, suffers from extremely poor responsiveness and maintainability, with APR rising to 31.09% and AVU reaching 49.97%. Even GPT-5, the strongest in visual fidelity, shows a high AVU of 37.72%, reflecting a frequent reliance on absolute positioning and non-standard style tokens.

## 4.4 CODE GENERATION PERFORMANCE ACROSS MODALITIES

To examine the impact of different modalities on the FIGMA2CODE task, we derive three input variants from the FIGMA2CODE dataset: image only, metadata only, and image + metadata. For each variant, we evaluate two corresponding methods using ERNIE 4.5 424B VL as the backbone model.

**While Figma metadata greatly improves visual fidelity, it also introduces rigidity that undermines responsiveness and maintainability**. As shown in Table 3, metadata-only methods achieve higher fidelity than image-only ones—for example, Direct Prompting reaches a VES of 0.6801 compared with 0.5653 for its image-only counterpart—while multimodal inputs perform best, with F2CAGENT attaining a VES of 0.7990, the highest across all methods. By contrast, image-only

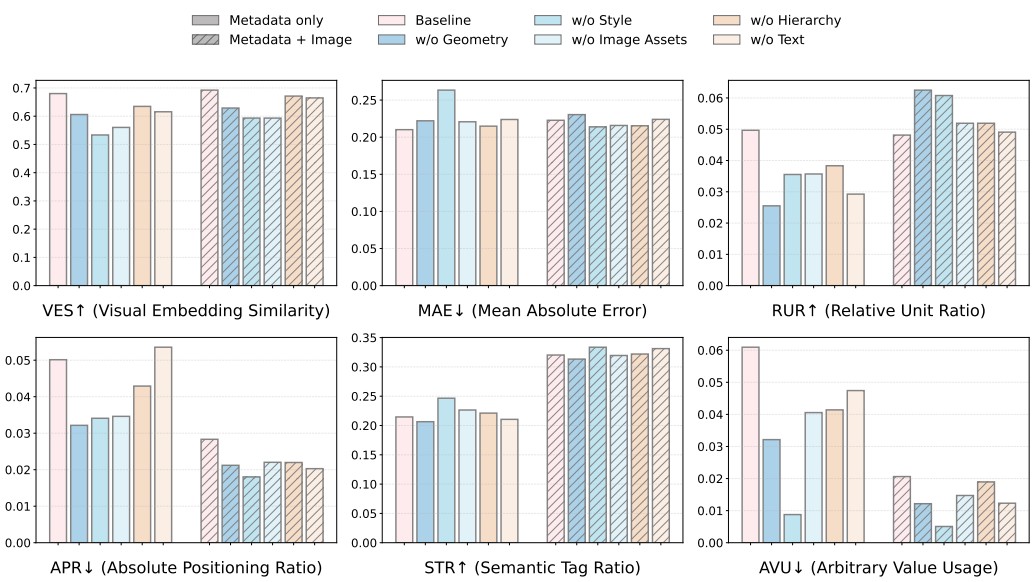

Figure 5: Ablation study on five metadata components. Using the ERNIE 4.5 424B VL model, we conduct *Figma-to-code* generation under both metadata-only and metadata+image settings, where each metadata component—geometry, style, image assets, hierarchy, and text—is removed in turn.

methods produce the most responsive and maintainable code, with APR and AVU values near zero, whereas metadata-based approaches often yield rigid layouts and arbitrary style tokens, as seen in APR 5.01% for metadata prompting and 13.57% for F2CAGENT. Template Conversion shows the most extreme case: despite a competitive MAE of 0.1205, its responsiveness and maintainability collapse, with RUR falling to 0%, STR to 0.01%, and APR soaring to 58.0%. **Overall, F2CAGENT achieves the strongest visual fidelity while partially alleviating structural issues compared to metadata-only prompting** (please see case studies in Appendix J).

### 4.5 HOW DOES METADATA INFLUENCE GENERATION PERFORMANCE?

To disentangle the effects of Figma metadata in the FIGMA2CODE task, we perform an ablation study on five key components: textual content, style attributes, hierarchical structure, image assets, and geometric information (details in Appendix G). Using ERNIE 4.5 424B VL as the backbone model, we conduct the ablation studies under both metadata-only and image + metadata conditions.

**Multimodal input mitigates degradation caused by ablations, highlighting the complementary strengths of visual and metadata signals**. As shown in Figure 5, drops in VES or rises in APR are less pronounced—indicating that visual and metadata signals work together to stabilize generation quality. **Style and image assets are most critical for visual fidelity**: removing style lowers VES from approximately 0.65 to below 0.55 and raises MAE above 0.25, while ablating image assets produces a similar decline, underscoring their importance for faithfully reconstructing visual appearance. By contrast, responsiveness is most sensitive to geometry and hierarchy: removing either halves RUR and raises APR, showing that these cues are essential for flexible, adaptive layouts. Interestingly, ablating style or geometry slightly improves maintainability, as reflected by higher STR and lower AVU, suggesting that current MLLMs struggle to fully exploit these signals and instead fall back on arbitrary tokens or non-semantic structures.

### 4.6 QUALITATIVE ANALYSIS

We present a representative UI code snippet generated by Grok4 alongside a manually created golden case, as illustrated in Figure 6. The Grok4 output relies heavily on absolute positioning, which hinders seamless adaptation across devices of different resolutions. It also overuses arbitrary style tokens and non-semantic $<div>$ elements, introduces unnecessary hierarchy, and thus produces code that is difficult to reuse and evolve. In contrast, the manually crafted golden case uses readable semantic

```
<div class="absolute left-[20px] top-[56px] w-[335px] h-[44px]">     <header class="absolute left-5 top-14
  <div class="absolute left-0 top-0 w-[44px] h-[44px]">              w-[335px] h-11 flex items-center">
    <div class="absolute left-0 top-0 w-[44px] h-[44px] rounded-      <button class="flex items-center
    full backdrop-blur-[30px] bg-[#f7f7fb]"></div>                     justify-center w-11 rounded-full
    
    top-[16.25px] w-[5.5px] h-[11.5px]" alt="Back">                    <img src="./arrow.svg" alt="Back"
  </div>                                                               class="w-[5.5px] h-[11.5px]">
</div>                                                              </button></header>
```

Figure 6: HTML snippet generated by Grok4 (left) and the golden implementation (right). The former overuses absolute positioning, arbitrary style tokens, and non-semantic $<div>$, while the latter uses semantic tags and flex layout, improving responsiveness and maintainability.

tags, Tailwind predefined classes, and flexible layouts, resulting in higher responsiveness and maintainability. This example underscores the limitations of proprietary MLLMs in the FIGMA2CODE task and demonstrates that automated *design-to-code* with MLLMs still faces significant challenges. We also present some generation cases in Appendix J.

## 5 RELATED WORK

**Multimodal Code Generation.** Recent MLLMs such as GPT-4o (Hurst et al., 2024), Gemini (Reid et al., 2024), Claude (Anthropic), MiniGPT-4 (Zhu et al., 2023), LLaVA (Liu et al., 2023), DeepSeek-VL (Lu et al., 2024), Qwen2.5-VL (Alibaba, 2025), and InternVL3 (Zhu et al., 2025) demonstrate strong multimodal grounding but still struggle on domain-specific code generation. To address this, specialized benchmarks and systems have emerged: ChartMimic (Shi et al., 2024), Plot2Code (Wu et al., 2024a), and ChartCoder (Zhao et al., 2025) target chart-to-code tasks; HumanEval-V (Zhang et al., 2024) and CodeVision (Xu & Sheng, 2025) test coding with diagrams; code generation has also been explored as visual reasoning in VQA (Shen et al., 2024; Sur'is et al., 2023). Datasets such as MMCode (Li et al., 2024) and VisCodex (Jiang et al., 2025) reveal the limitations of current LMMs and propose vision-language–code integration. Overall, MLLMs present significant opportunities and potential for advancing multimodal code generation.

**Design to Code.** Early *design-to-code* systems explored deep learning on simple cases (Beltramelli, 2017; Chen et al., 2018). Recent work emphasizes layout reasoning and iterative refinement: layout-as-thought (Gui et al., 2025b) decomposes designs into sequential blocks; feedback-driven systems (DeclarUI (Zhou et al., 2025), WAFFLE (Liang et al., 2024), UICoder (Wu et al., 2024b)) refine outputs through compiler or training signals; and hierarchical pipelines (DCGen (Wan et al., 2024b), LaTCoder (Gui et al., 2025b)) improve composability by dividing complex UIs. Extensions include coarse-to-fine strategies (UICopilot (Gui et al., 2025c)) and structural fidelity via hierarchy-aware grouping (DesignCoder (Chen et al., 2025)). These advances are supported by large-scale benchmarks—WebUI (Wu et al., 2023), Design2Code (Si et al., 2024), Web2Code (Yun et al., 2024), IW-Bench (Guo et al., 2025), and WebCode2M (Gui et al., 2025a)—which standardize evaluation on both code correctness and layout fidelity. However, most existing approaches remain limited to image-only UI generation and rarely incorporate industry-grounded design drafts (e.g., Figma), resulting in a considerable gap from practical industrial applications.

## 6 DISCUSSION AND CONCLUSION

Although our work represents an important step in advancing *design-to-code* research beyond image-only methods, it remains an initial exploration that defines the scope of this line of study rather than a definitive solution. To establish a principled foundation and ensure tractability, reproducibility, and comparability across models, we adopt several pragmatic design choices. For instance, we standardize the generation target to Tailwind CSS—widely adopted in practice and modular in structure—which serves as a practical baseline without loss of generality. Likewise, we introduce streamlined evaluation criteria that, while simple, capture multiple dimensions of code quality and motivate future extensions toward richer evaluation signals. Our proposed agentic workflow, F2CAGENT, demonstrates promising gains in visual fidelity for a weaker open-source model (ERNIE 4.5 424B VL), while also highlighting opportunities to strengthen responsiveness and maintainability. Yet, the broader

challenge of reconciling visual fidelity with robust engineering quality continues to define the frontier of multimodal *design-to-code* research.

In conclusion, we formally introduce the FIGMA2CODE task and present the first dataset constructed from real-world Figma design files integrating images, metadata, and assets. Through systematic evaluation of state-of-the-art MLLMs, we show that these models still struggle to balance visual fidelity with engineering practicality in *Figma-to-code* generation: while Figma metadata substantially enhances visual fidelity, it simultaneously introduces challenges for responsiveness and maintainability. We believe this work represents a significant step toward practical and scalable design-to-code automation. For future work, we aim to explore reinforcement learning–based agents and more reliable agentic workflows to address these limitations, as well as diffusion models to improve the efficiency of UI code generation.

## ACKNOWLEDGEMENTS

This work is supported by the Major Program (JD) of Hubei Province under Grant No.2023BAA024, and the National Natural Science Foundation of China under Grant No.U24A20326. This work is also partially sponsored by CCF-Huawei Populus Grove Fund (No. HuaweiSE2025027).

## ETHICS STATEMENT

Our dataset is sourced from the public Figma community. To mitigate ethical risks, such as the inclusion of private or inappropriate content, we conducted a rigorous manual filtering process in which design experts reviewed every sample (see Section 3.1 and Appendix C.2). This ensures that the final dataset is appropriate for research use and fully complies with the platform's terms of service. Details regarding licensing and redistribution compliance are provided in Appendix B.

## REPRODUCIBILITY STATEMENT

To ensure reproducibility, we detail our dataset construction in Section 3.1 and Appendix C, and our evaluation framework in Section 4.1 and Appendix E. Additional implementation details are provided in Appendix D. We will release the FIGMA2CODE dataset along with our codebase and experimental scripts to enable full verification of results and facilitate future research. To further strengthen reproducibility, we set the temperature parameter of all MLLM invocations to 0 and fix random seeds for Python's `random`, `torch`, and `numpy` packages. Detailed inference protocols for invoking MLLMs are presented in Appendix F.3.

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

# Part I

# Appendix

## Table of Contents

## A    USAGE OF LARGE LANGUAGE MODELS

In compliance with the ICLR 2026 policies on the use of Large Language Models (LLMs), we hereby disclose the usage of such models in the preparation of this manuscript. Our use of LLMs can be categorized into three main areas: data annotation as part of our research methodology, assistance with code implementation, and writing assistance for the manuscript.

**LLM in Paper Writing.**   We utilize LLMs to aid in the writing process, primarily for improving grammar, enhancing clarity, and refining the phrasing of the manuscript. The goal is to improve the overall readability and quality of the text.

**LLM in Data Annotation.**   As detailed in Section 3.1 and Appendix C.3, an LLM (GPT-4o-mini) is employed as a tool to assist in the semantic annotation of our dataset. The model's role is to categorize UI design pages into predefined content classes and generate concise descriptions. This process is conducted under a human-in-the-loop framework, where the LLM-generated annotations are part of a broader supervised data curation pipeline. The prompts and methodology are explicitly documented to ensure transparency.

**LLM in Code Implementation.**   LLMs are utilized to assist in the development and debugging of scripts for our experimental framework. This includes generating code snippets for data processing, implementing evaluation metrics, and refining algorithms.  All LLM-assisted code is carefully reviewed, tested, and verified by the authors to ensure its correctness and efficiency.

**Author Responsibility.**   In accordance with the ICLR policy, the authors have meticulously reviewed, revised, and validated all content produced with the assistance of LLMs. We assume full responsibility for all statements, claims, and any potential inaccuracies within this submission. The final content of this paper, including all the code and text, reflects the authors' own work and conclusions.

## B    LICENSING AND REDISTRIBUTION COMPLIANCE

All design files in our dataset are sourced from the **Figma community**, which allows users to publicly share design artifacts. In accordance with Figma's licensing framework,[2] free community files are distributed under the **Creative Commons Attribution 4.0 (CC BY 4.0)** license, which permits sharing and adaptation (including for commercial use) provided that appropriate credit is given. Paid resources, in contrast, are governed by the **Community Paid Resource License** and cannot be redistributed in their original form. To ensure compliance, we conduct a systematic review of the license and content before redistribution.

**License Inspection.**   For each Figma file, we record its source URL, author information, and license type (free CC BY or paid). When explicit license information was missing, ambiguous, or marked as paid, we exclude the corresponding raw assets from redistribution.

**Attribution.**   For files under CC BY 4.0, we provide author credit in our dataset metadata and release notes, satisfying the attribution requirement.

**Asset Handling.**   For design assets such as images and icons, we localize them into our dataset package only when redistribution was explicitly permitted.  Otherwise, we retain only rendered screenshots and structural annotations, while pointing users to the original Figma source.

**Content Filtering.**   We manually filter out files containing sensitive, offensive, or copyrighted third-party material to avoid inappropriate redistribution.

**Usage Restriction.**   We release the dataset strictly for **non-commercial research purposes**. Any commercial use of the design files or assets remains subject to the original Figma license terms and, where applicable, the Paid Resource License.

Through these steps, we ensure that all redistributed components of FIGMA2CODE comply with licensing requirements. Ambiguous or paid cases are handled conservatively by limiting distribution

---

[2]https://help.figma.com/hc/en-us/articles/360042296374-Figma-Community
-copyright-and-licensing

to derived representations only (e.g., screenshots and metadata), while free CC BY resources are redistributed with proper attribution.

## C    DETAILS OF DATASET CONSTRUCTION

This section provides a comprehensive account of the methodologies and protocols implemented at each stage of the FIGMA2CODE dataset's construction.

### C.1    DATA ACQUISITION PIPELINE

This subsection details the technical approach for collecting the initial corpus of raw design files from the Figma community.

**Automated Crawler Implementation.**    To ground our dataset in authentic, real-world design practices, we choose the Figma community as our primary data source. We implement an automated data collection script using Python with the Selenium WebDriver framework. This script programmatically controls a web browser to navigate the website's dynamic-loading content, systematically crawling the unique identifiers (`filekeys`) of publicly available design files. The collection process involves two main stages. First, the script accesses a predefined list of Figma community category pages, parsing up to 300 design template links from each. Second, after deduplicating this aggregated list of links, the script iterates through each unique URL, simulating a user click on the "Open in Figma" button. In the URL of the newly opened browser tab, the `fileKey` is then precisely extracted using the regular expression `/(file|design)/(^/]+)/`.

**Metadata and Image Retrieval via Figma API.**    Using the collected `filekeys`, we leverage the official Figma API to retrieve the complete metadata for each design file in a structured JSON format. Within this hierarchical data, we identify the target design pages by selecting nodes of type `FRAME` that are direct children of a `CANVAS` node. The unique IDs of these `FRAME` nodes are then collected. In the final step, these IDs are used to make subsequent API calls to download the high-resolution rendered image of each respective page, thereby assembling our initial raw dataset.

**Target Source Specification.**    We target free "files" resources across a diverse range of application scenarios to ensure design diversity. The specific Figma community URLs include:

- https://www.figma.com/community/website-templates
- https://www.figma.com/community/website-templates/blog
- https://www.figma.com/community/web-ads
- https://www.figma.com/community/design-templates
- https://www.figma.com/community/mobile-apps
- https://www.figma.com/community/portfolio-templates
- https://www.figma.com/community/resume-templates

### C.2    MULTI-STAGE CURATION PROTOCOL

This subsection outlines the rigorous filtering and selection process designed to distill high-quality samples from the raw data corpus, which contained a significant number of irrelevant or low-quality samples.

**Heuristic Pre-filtering Rules.**    To efficiently process the massive raw data, we first discard pages that are obviously unsuitable according to four heuristic rules: ① Size Validity—pages with a bounding-box width or height $\leq 0$; ② Extreme Aspect Ratio—the ratio of the longest to shortest side exceeds 5:1, typical of non-standard UI designs such as flowcharts or banners; ③ Content Sparsity—fewer than three direct child nodes, indicating a near-blank canvas; ④ Dominant Placeholder Image—any single image node occupies $\geq 80\%$ of the total page area.

**CLIP-based Visual Deduplication.**   To further improve visual diversity, we leverage the pre-trained CLIP model (ViT-B/32) to detect and eliminate perceptually near-duplicate pages in two stages: (1) Intra-Filekey Deduplication clusters pages within each Figma file whose cosine similarity exceeds 0.95, retaining only the page with the largest image file size as a fidelity proxy; (2) Inter-Filekey Deduplication pools the survivors and repeats the same clustering across the entire corpus, removing residual duplicates that span multiple files. This dual-stage strategy balances coverage and uniqueness while remaining computationally tractable.

**Manual Screening Criteria.**   The final filtering stage involve manual review by design experts to ensure semantic quality and relevance. The primary objective is to remove non-conforming pages while retaining only those representing complete, self-contained UI designs. Pages are discarded if they meet any of the following criteria: (1) Incomplete or Missing Content—designs containing only a single icon, image, or otherwise truncated layout; (2) Non-UI Artifacts—design-system documentation, Figma covers, or presentation slides; (3) Overly Simple Layouts—splash screens or basic forms lacking sufficient structure; (4) Abnormal Composition—large areas of meaningless whitespace or chaotic arrangement; (5) Highly Similar Variants—near-duplicates differing only in theme, color, or minor details; and (6) Sensitive or Inappropriate Content—private information, violence, or any other unethical material. This manual sweep ensures that only coherent, self-contained, and ethically sound UI designs enter the final benchmark.

## C.3   SEMANTIC ANNOTATION AND ENRICHMENT

Figure 7: Prompt for using GPT-4o in additional annotation.

This subsection describes the process of annotating the curated dataset with rich, multi-dimensional labels to facilitate detailed analysis.

**Manual Annotation Dimensions.**   Using annotation interface shown in Figure 8, each retained page is examined along three orthogonal axes under strict guidelines to guarantee inter-annotator consistency:

① **Platform**—label the target environment as *Mobile* or *Desktop*. *Decision cues:* (i) **Canvas/ratio**: portrait, typical mobile frames (e.g., 375×812, 390×844) indicate Mobile; landscape, desktop frames (e.g., 1440×900, 1920×1080) indicate Desktop. (ii) **Context**: phone/tablet apps or webpages rendered on phones count as Mobile; desktop apps or webpages rendered on desktop count as Desktop. (iii) **UI patterns**: Mobile often shows status bar (signal/battery/time), top app bar with back, bottom tab bar, drawer; Desktop often shows top horizontal nav, side nav, breadcrumbs, denser clickable links. Resolve ambiguity by prioritizing concrete UI patterns over canvas alone.

② **Complexity**—grade structural richness as *Low*, *Mid*, *High*, or *Hard* using three lenses: hierarchy depth, component diversity, and layout organization. *Low*: 1–2 levels; ≤5 basic components (text, button, image, input, icon, etc.); single or simple two-column layout; ample whitespace; typical of login, simple forms, onboarding. *Mid*: 2–3 levels; 6–10 component types; multi-column or card layouts; lists/grids of moderate complexity; clear visual hierarchy; typical of product lists, profile, settings. *High*: 3–4 levels; 11–15 component types; complex multi-column; includes charts, complex tables, or multi-level menus; high information density and rich functions; typical of dashboards or analysis. *Hard*: ≥4 levels; ≥16 component types; highly intricate layouts with many interactive/dynamic modules; integrated multi-module workspaces; complex information architecture and high cognitive load; typical of comprehensive admin suites or multi-tool workbenches. When in doubt, select the level matching the most demanding criterion among depth, diversity, or layout complexity.

③ **Quality Rating**—score visual and experiential quality on a 3-point scale: *1–Fair*, *2–Good*, *3–Excellent*. *1–Fair*: basic color pairing without conflict but lacks highlights; conventional typography with limited hierarchy; usable but average experience; styles mostly consistent yet coarse in details; information clarity is adequate, interactions minimally considered. *2–Good*: coordinated colors and clear visual layers; appropriate typography with readable hierarchy; strict adherence to design tokens; consistent components and spacing; smooth basic UX with clear information and standard usability. *3–Excellent*: outstanding color and strong visual impact; refined, layered typography; meticulous, innovative yet consistent component design; superior UX with intuitive flows, precise information delivery, and attention to accessibility; demonstrates thoughtful, possibly novel interactions. Annotators should rate based on the overall impression, resolving ties by weighing consistency and UX clarity over mere visual flair.

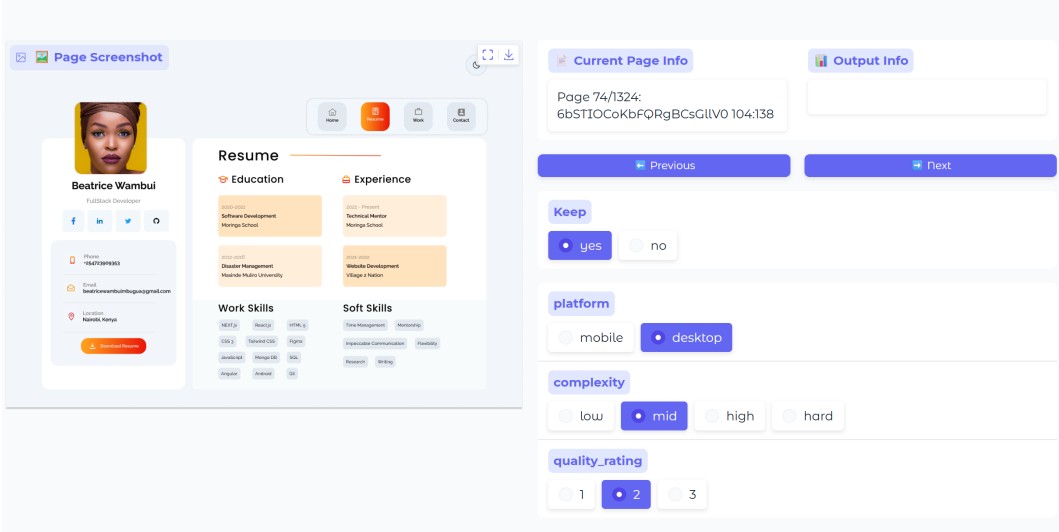

Figure 8: The custom-built interface for manual annotation of design pages. From this UI, design experts assign labels for Platform (Mobile/Desktop), Complexity (Low, Mid, High, Hard), and Quality Rating (1-Fair, 2-Good, 3-Excellent) to each curated design.

**MLLM-Assisted Content Categorization.** To further enhance data richness while reducing manual effort, we use `gpt-4o-mini` to annotate the `content` and `description` of the retained pages. We establish a comprehensive classification system dividing page content into 12 main cate-

gories: ① User Identity & Personalization—user profiles, account setting; ② Marketing & Landing Content—landing pages, blogs, product reviews; ③ E-Commerce & Transactions—product browsing, payment, order management; ④ Data & Information Presentation—dashboards, data reports, list feeds; ⑤ Forms & Interaction Flows—forms, filters, search functions, modals; ⑥ Communication & Collaboration—instant messaging, community discussions; ⑦ Support, Guidance & Onboarding—help centers, tutorials, onboarding guides; ⑧ Notifications & States—notifications, warnings, error/empty state pages; ⑨ Media & Entertainment—music players, photo albums, games; ⑩ Scheduling & Activities—calendars, appointments, itinerary planning; ⑪ Health & Lifestyle—fitness tracking, meal plannings; ⑫ Others—categories not fitting above, such as contracts or maps.

## C.4 Data Transformation and Structuring

This subsection details the post-processing steps for converting verbose Figma JSON, which contains editor-specific properties, into a clean, lightweight, and model-friendly structured representation.

**Structure Pruning and Abstraction.** This step streamlines the data structure and extracts the core UI information.

- **Node Refinement:** We remove nodes that are ineffective in the final rendering. This includes: (a) invisible nodes with `visible: false` or `opacity: 0`; (b) empty shape nodes with no `fills` or `strokes` styles; (c) nodes completely occluded by opaque elements above them, identified via a reverse Z-order traversal algorithm.
- **Layer Refinement:** We flatten the node tree by removing redundant nesting. For container nodes (`FRAME` or `GROUP`) that contain only a single child and have no style information of their own, we remove them and promote their child node to the container's original level. For layer groups that distribute a single set of style information across multiple layers, we merge the multi-level style information to reduce nesting depth.
- **Property Refinement:** We filter out metadata properties specific to the Figma editor (e.g., `locked`, `exportSettings`), as these are irrelevant to the final UI code structure.
- **Asset Abstraction:** Figma allows designers to freely assemble icons, resulting in some icons being composed of multiple shape nodes. We identify shape nodes belonging to the same icon using heuristic rules (node type is one of `VECTOR`, `STAR`, `LINE`, `ELLIPSE`, `REGULAR_POLYGON`, `BOOLEAN_OPERATION`, or the name contains "merge"). These are merged into a unified `SVG_ASSET` abstract node for subsequent export as static assets.
- **Coordinate and Numerical Value Standardization:** We convert the absolute coordinates of all nodes to relative coordinates with the page's root node as the origin. Since high precision is not required for implementation, we also standardize all floating-point values to three decimal places to reduce input size.

**Asset Collection and Integration.** To create fully self-contained data samples, we identify and localize all external dependencies. This involves:

- **Asset Identification and Download:** We identify and collect all external resources for localization and integration. This primarily includes: (a) `imageRef` identifiers in the `fills` property (bitmaps); (b) the ID of `SVG_ASSET` nodes (vector graphics); (c) the `Key` for `components`, `componentSets`, and `styles` defined at the top level of the file. After collection, the corresponding PNG, SVG, and JSON definition files are batch-downloaded via the Figma API.
- **Asset Deduplication & Integration:** The goal of this stage is to eliminate resource redundancy and associate local resources with the node tree.
  - ▷ *Resource File Deduplication:* While assets identified by a `key` are often unique, assets identified by a node ID commonly have identical duplicates (e.g., the same icon used in multiple places). These duplicates typically share the same name in the node tree. To reduce storage overhead, we perform a deduplication process: first, we group files by node name, and then we perform a content comparison within each group. A mapping from duplicate IDs to a primary ID is established for content-identical resources.
  - ▷ *Path Association and Format Unification:* We update the `imageRef` in the node tree to a local relative path. The previously abstracted `SVG_ASSET` nodes are converted into a standard

RECTANGLE node, and its `fills` property is set to an `IMAGE` type fill pointing to the local SVG file. This approach unifies the handling of bitmaps and vector graphics, maintains Figma's syntax, and simplifies the processing logic for downstream models. Concurrently, the JSON definitions for `components`, `componentSets`, and `styles` are embedded into the main data file to ensure self-containment of information.

## C.5 DATASET PARTITIONING STRATEGY

To support robust model evaluation, we partition our final collection of 3,055 designs. A high-quality test set of 213 samples is constructed using a rigorous process of stratified sampling followed by manual curation, with the remaining 2,842 samples forming the training set.

**Stratified Sampling Framework.** Our core strategy is Stratified Sampling to ensure the diversity and representativeness of the test set across key features. We construct a hierarchical structure based on the following four core dimensions: ① Platform — Mobile *vs.* Desktop; ② Complexity — Low, Mid, High, Hard; ③ Quality Rating — 2, 3 (samples with a rating of 1 are excluded to retain higher-quality samples); ④ Content Category — 12 main functional categories. We calculate the proportion of samples for each combination of the above dimensions (i.e., each "stratum") in the total dataset and allocate samples for the test set according to these proportions, ensuring it serves as an unbiased microcosm of the overall dataset.

**Expert-in-the-Loop Selection Process.** To guarantee the quality and typicality of the test samples, we design a two-stage selection process:

- **Phase 1: Automated Candidate Set Generation**
  We first run an automated script to draw an oversampled candidate set from each stratum based on the stratified sampling strategy. The number of samples in this candidate set ($N_{redundant}$) is dynamically adjusted based on the required number of samples for that stratum ($N_{target}$) and the number of remaining samples ($N_{remain}$), aiming to provide ample high-quality options for manual selection. The calculation is as follows:

$$N_{redundant} = \begin{cases} \min(N_{remain}, 6), & N_{target} < 3 \\ \min(N_{remain}, 3N_{target}), & 3 \leq N_{target} \leq 5 \\ \min(N_{remain}, 2N_{target}), & N_{target} > 5 \end{cases}$$

- **Phase 2: Expert Manual Selection**
  The generated candidate set (including image previews and stratum information) is submitted to design experts. Experts review all candidate samples within each stratum and select those that best represent the category and have the clearest design. This "human-in-the-loop" process leverages both the objectivity and efficiency of automated scripts for handling large-scale data and the profound insights of human experts in aesthetics and functional understanding. This ultimately ensures the quality and typicality of every sample in the test set.

Through this process, we construct a high-quality, well-distributed, and representative test dataset. This dataset not only provides a solid foundation for our model evaluation but also serves as a reliable benchmark for future research.

## D DATASET ANALYSIS

We conduct a statistical analysis of the FIGMA2CODE dataset to demonstrate its diversity and complexity.

**Composition and Diversity.** The dataset contains 45% mobile and 55% desktop designs, reflecting a balanced coverage of platforms. In terms of complexity, 46% of the designs are medium, 26% high, 15% low, and 14% very high. Most designs are in English, with a small fraction in other languages. Quality ratings are nearly balanced, with 53% labeled as good and 47% as excellent. Light themes account for 74% of the dataset, while the remaining 26% are dark. Layout usage varies across samples: 3% adopt vertical layout, 16% use horizontal layout, 48% combine both vertical and

horizontal layouts, and 32% do not employ Auto Layout. As illustrated in Figures 9, these statistics highlight the dataset's diversity in platform, complexity, and general characteristics.

**Technical and Structural Features.** The metadata token length is mostly within 250,000, though a few samples reach up to 2,500,000. The maximum DOM depth mainly falls between 3–7. Node counts are typically below 200, though some designs exceed 700. The number of resource files is usually under 100, with a few samples reaching 350. Image resolutions cluster into two ranges: 800–1200 px width × 1500–2500 px height for mobile designs, and 2500–3000 px width × 1500–2500 px height for desktop designs. The variation of these structural properties across samples is visualized in Figures 10–15.

**Content and Functional Category Analysis.** We analyze the textual content and functional attributes of the designs to understand their distribution, as shown in Figure 16. Across functional categories, the dataset spans 12 types, with *Data & Information Presentation*, *E-Commerce & Transactions*, and *Marketing & Landing Pages* being the most common. To further capture semantic patterns, a word cloud is generated highlighting prominent keywords such as *design*, *designer*, *contact*, and *services*, as well as commerce-related terms (*product*, *shop*, *e-commerce*) and user interaction keywords (*account*, *payment*, *support*, *search*). The resulting word cloud is presented in Figure 17, collectively illustrating the dataset's semantic diversity and functional coverage.

In summary, FIGMA2CODE demonstrates clear advantages in scale, diversity, and structural complexity, making it more representative of the real-world challenges in *design-to-code* generation.

# E    IMPLEMENTATION OF EVALUATION METRICS

## E.1    VISUAL FIDELITY

- **Visual Embedding Similarity (VES).** This term evaluates the semantic fidelity of the generated UI code with respect to the original design mockup. It is defined as $\text{VES} = \cos(\text{Encode}(I), \text{Encode}(\hat{I}))$, where $I$ denotes the generated page screenshot and $\hat{I}$ the original design image, and $\texttt{Encode}$ is a vision encoder. We adopt DINOv2 as the encoder instead of OpenCLIP, since DINOv2 is a self-supervised vision foundation model that learns robust and transferable visual representations, surpassing OpenCLIP on most image- and pixel-level benchmarks. In contrast, CLIP's image–text contrastive objective primarily emphasizes coarse-grained semantic alignment and is less sensitive to fine-grained layout or visual variations.

- **Mean Absolute Error (MAE).** This term measures the pixel-level fidelity of the generated UI code with respect to the design mockup. It is defined as the normalized mean absolute error between the generated page screenshot $I$ and the design image $\hat{I}$:

$$\text{MAE} = \frac{1}{N} \sum_{i=1}^{N} |I_i - \hat{I}_i|$$

where $I_i$ and $\hat{I}_i$ denote the pixel values at position $i$, and $N$ is the total number of pixels. Normalization ensures comparability across different resolutions. Prior work Wan et al. (2024a) has shown that MAE correlates strongly with human perceptual preferences, making it an effective proxy for visual fidelity at the pixel level.

## E.2    LAYOUT RESPONSIVENESS.

- **Relative Unit Ratio (RUR)** Relative units (e.g., %, `em`, `rem`, `vw/vh`, `fr`) enable layouts to scale across devices. We define $\text{RUR} = \frac{N_{\text{rel}}}{N_{\text{layout}}}$, where $N_{\text{rel}}$ counts the usage of relative units in layout-related CSS properties and $N_{\text{layout}}$ is the total number of layout-related declarations (sum of both absolute layouts and relative layouts). For example, using `width:50%` instead of `width:500px` increases the RUR.

- **Absolute Positioning Ratio (APR).** Excessive absolute/fixed positioning reduces adaptability across viewports. We define $\text{APR} = \frac{N_{\text{abspos}}}{N_{\text{position}}}$, where $N_{\text{abspos}}$ is the number of elements with `position:absolute` or `position:fixed`, and $N_{\text{pos}}$ is the total number of positioned

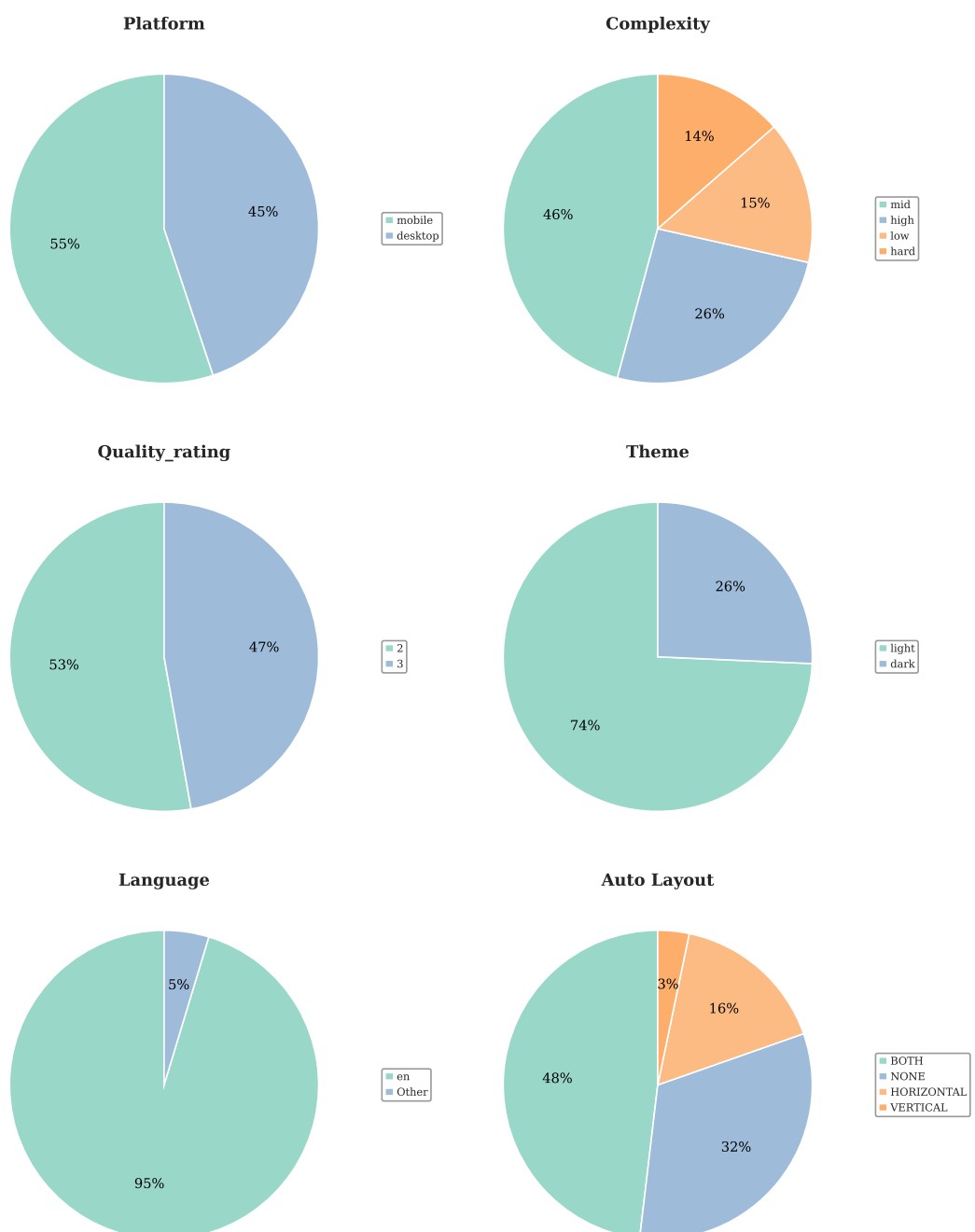

Figure 9: Distribution of dataset samples across multiple attributes, including platform, complexity, quality rating, theme, language, and auto layout.

elements. For example, replacing `position:absolute` with a flexbox-based layout lowers the APR.

- **Flex/Grid Utilization (FU).** Modern CSS layout models improve structured and adaptive reflow. We define $\text{FU} = \frac{N_{\text{flex/grid}}}{N_{\text{containers}}}$, where $N_{\text{flex/grid}}$ counts container elements adopting Flexbox or Grid, and $N_{\text{containers}}$ is the total number of container elements. For example, using a grid layout for a gallery instead of manual floating increases the FU.

- **Breakpoint Coverage (BC).** Responsive design requires targeting multiple standard viewports. We define $\text{BC} = \frac{N_{\text{covered}}}{N_{\text{std}}}$, where $N_{\text{covered}}$ is the number of standard viewport ranges ( S / M / L

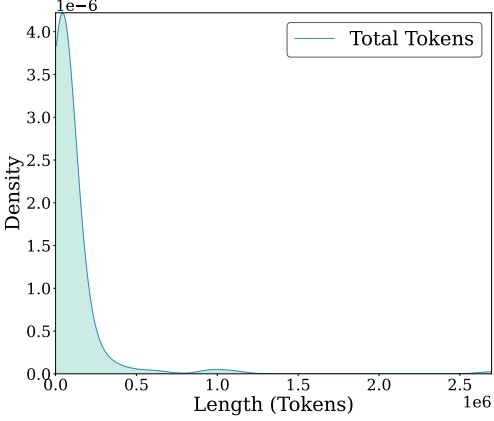

Figure 10: Estimated total token length distribution.

Figure 11: DOM depth distribution.

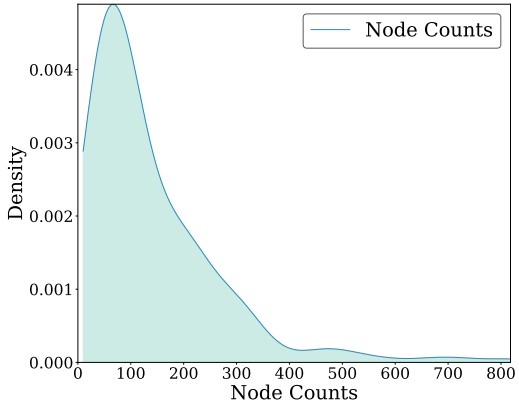

Figure 12: Node counts distribution.

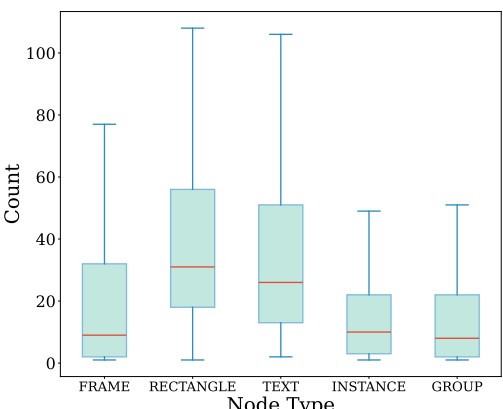

Figure 13: Different node types distribution.

/ XL) explicitly targeted through media queries or utility prefixes, and $N_{\text{std}}$ is the total number of such standard ranges. For example, defining styles for both mobile and desktop viewports increases the BC compared to only supporting one.

The weights $w_1 \ldots w_5$ balance the contribution of different aspects; in this work we set them heuristically, but they can be tuned or learned from human preference data in future studies.

### E.3 CODE MAINTAINABILITY

- **Semantic Tag Ratio (STR).** Semantic HTML improves code readability and accessibility. We define $\text{STR} = \frac{N_{\text{semantic}}}{N_{\text{elem}}}$, where $N_{\text{semantic}}$ counts semantic elements (e.g., `header`, `nav`, `main`, `article`, `ul/li`), and $N_{\text{elem}}$ counts all DOM elements. For example, using `<button>` instead of a `<div>` with `onclick` improves the score.

- **Arbitrary Value Usage (AVU).** Arbitrary values in utility classes reduce consistency with design tokens. We define $\text{AVU} = \frac{N_{\text{arbitrary}}}{N_{\text{class}}}$, where $N_{\text{arbitrary}}$ counts class tokens using arbitrary value syntax (e.g., `w-[123px]`, `bg-[#ff0]`), and $N_{\text{class}}$ is the total number of class tokens. For example, replacing `w-[123px]` with a standardized spacing utility (e.g., `w-32`) lowers the AVU.

- **Inline Style Ratio (ISR).** Inline styles hinder reusability and separation of concerns. We define $\text{ISR} = \frac{N_{\text{style}}}{N_{\text{elem}}}$, where $N_{\text{style}}$ is the number of elements carrying inline style declarations and $N_{\text{elem}}$ is the total number of DOM elements. For example, using `` decreases the ISR if replaced with a reusable CSS class.

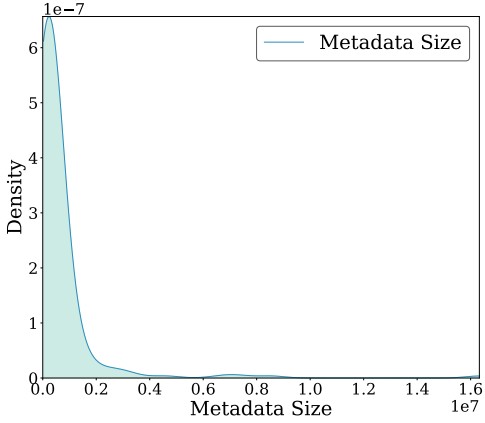

Figure 14: Estimated metadata size distribution.

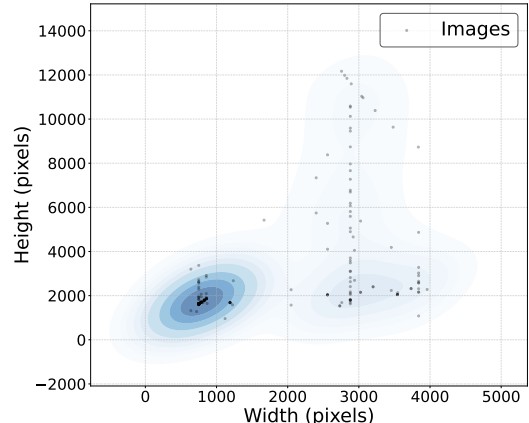

Figure 15: Scatter density visualization of image width and height.

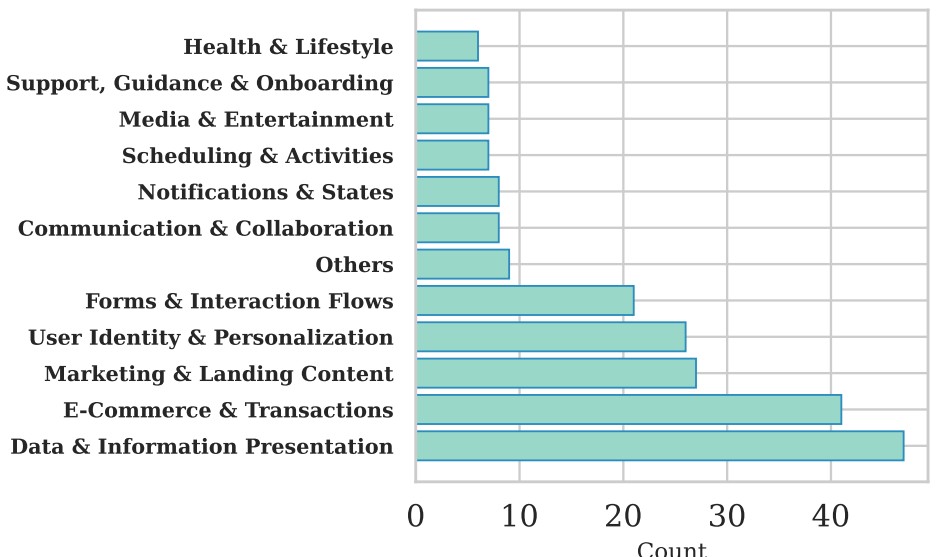

Figure 16: Different content type distribution across the dataset.

- **Custom Class Reuse (CCR).** Reusing custom (non-utility) classes reflects style modularization. We define $\mathrm{CCR} = \frac{N_{\mathrm{reused}}}{N_{\mathrm{custom}}}$, where $N_{\mathrm{reused}}$ is the number of custom class names applied to multiple elements (used frequency greater or equal to 2) and $N_{\mathrm{custom}}$ is the total number of distinct custom class names. For example, encapsulating button styles in a custom class used by 10 buttons yields a higher CCR than giving each button unique inline styles.

# F  IMPLEMENTATION OF BENCHMARKING METHODS

## F.1  DIRECT PROMPTING

**Design Rationale.**   This section presents the implementation of the *Direct Generation* baseline. The goal is to provide a fundamental, non-iterative benchmark for generating front-end code from design files. The core principle is to leverage the comprehension and generation capabilities of Large Multimodal Models (LMMs) to directly translate Figma design specifications into fully functional HTML code through a single-pass, structured prompt.

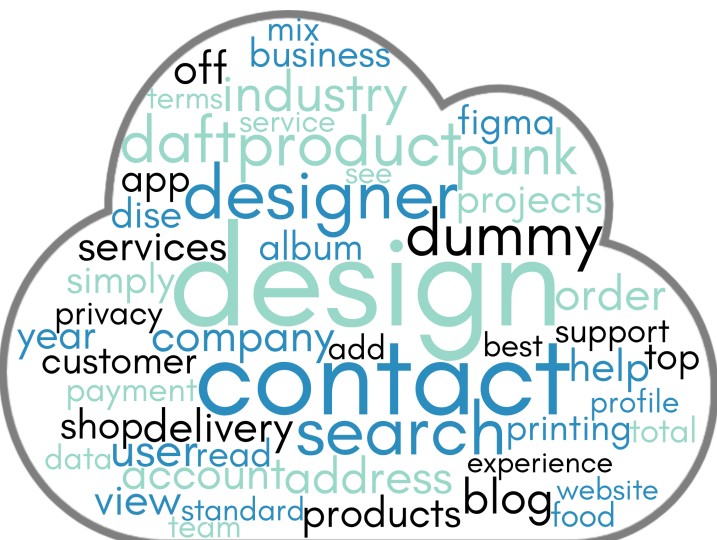

Figure 17: Word cloud of the most frequent terms in the dataset.

We adopt **Tailwind CSS** as the target code format because it is both widely used in real-world front-end development and structurally suitable for research. Compared with arbitrary CSS, Tailwind offers a standardized set of utility classes that reduce the output space while retaining sufficient expressiveness to reproduce complex layouts. This makes evaluation more consistent and reproducible across MLLMs. At the same time, Tailwind aligns with industrial workflows, ensuring that our benchmark is not purely synthetic but grounded in practical practice. Thus, Tailwind provides a reasonable and effective starting point for design-to-code generation research.

This baseline supports two input modalities to evaluate the impact of different information sources on code quality:

- **Single-modal (JSON-only):** Uses only the preprocessed Figma JSON as input, testing the model's ability to reconstruct a visual interface from purely structured data.
- **Multi-modal (JSON + Screenshot):** Provides both the Figma JSON and a corresponding screenshot. In this setup, the JSON is defined as the **authoritative source of truth** for layout, styling, and asset binding, while the screenshot serves only as an auxiliary reference to resolve ambiguities or supplement missing context.

**Prompt Engineering Strategy.** To ensure stability and reproducibility, we design a rigorous prompt engineering strategy. It relies on a meticulously crafted **System Prompt** that functions as a "behavioral contract" to guide the model's reasoning and output. Design data are then delivered in a structured format, preceded by a "content overview" to aid comprehension.

The system prompt is not a simple natural language request but a highly structured, machine-readable specification, built on three principles:

- **Principle 1: Clear Information Hierarchy.** Each input source is assigned a role and priority. In the multi-modal case (Figure 18), the JSON is explicitly designated as the non-negotiable source of truth for all design tokens and asset paths, while the screenshot is a secondary reference only. This prevents the model from making visual inferences that contradict structured data, ensuring fidelity to the original design.
- **Principle 2: Transparent Preprocessing.** To build a more reliable interaction with the model, we explicitly communicate all preprocessing steps. As shown in Figure 19, we inform the model that opaque image identifiers have already been resolved into relative paths. This prevents the model from guessing asset locations and grounds its output in the provided file structure.
- **Principle 3: Strict Output and Behavioral Constraints.** We impose strict constraints to prevent errors and ensure utility. The prompt mandates pure HTML output for direct automated evaluation.

The most critical rule is **Mandatory Image Asset Binding**, which requires the model to render an HTML element for every specified image asset using the exact provided path, eliminating hallucinated or missing assets.

---

**Prompt for JSON-only FIGMA2CODE Generation**

You are an expert front-end developer. Generate a complete HTML document using Tailwind CSS classes from the provided Figma JSON. Adjust the generation strategy according to the preprocessing notes in the input contract. The implementation should ensure that the code not only reproduces the design draft visually, but also enhances responsiveness and maintainability.

The Figma JSON is the primary source for layout, spacing, sizing, colors, typography, borders, radii, shadows/effects, and asset bindings.

**Input Contract:**

- **Contains:** Figma JSON (structured).
- **Preprocessing Notes:**
    - The provided Figma JSON is preprocessed.
    - Originally, Figma JSON `imageRef` values were opaque IDs (e.g., `347ba7a7c57adabed33deffd9e936c9b285f611e`).
    - These `imageRef` values have been replaced with actual local relative paths (e.g., `assets/foo.png`, `assets/bar.svg`).
    - Some vectors were merged and exported as local SVG files. A rectangle node was created to hold each exported SVG, with the local file path written into its `imageRef`.
    - Unnecessary properties have been removed to simplify the JSON.

**Output Specification:**

- **Format:** Valid HTML document only (no extra text before or after).
- **Example:**

```
<!DOCTYPE html>
<html lang="en">
<head> ... </head>
<body> ... </body>
</html>
```

- **Requirements:**
    - Return **only** the complete HTML document.
    - Do not wrap the output in JSON or any other structure.
    - The HTML must use Tailwind CSS classes and preserve the visual fidelity of the design.

**Constraints:**

1. **Mandatory Image Asset Binding:** For every visible node whose fills include an IMAGE with a non-empty, case-sensitive `imageRef` value (already a local relative path), you must render that exact asset path in the HTML. This is a hard requirement.
2. **Allowed Rendering for Image References:** Prefer `` for placed/raster/SVG assets. Use CSS `background-image` only when the design explicitly uses the image as a background fill. When using Tailwind arbitrary values, escape properly, e.g., `bg-[url('assets/foo.png')]`.
3. **Path Integrity:** Do not alter provided paths or their semantics.
4. **No Hallucinated Assets:** Do not introduce assets that are not explicitly specified in the JSON.
5. **External Resources:** No external URLs are permitted except the Tailwind CDN.

**Code Format:** The final output must be the complete HTML document.

---

Figure 18: System prompt for the JSON-only modality, illustrating the explicit definition of input contract, preprocessing notes, and constraints.

**Integration with Ablation Studies.** The process is designed to naturally support ablation studies. Before the Figma JSON is passed to the model, it can be programmatically "ablated" (e.g., by removing styling information, text content, or hierarchical structure). This ensures that in comparative experiments, all other aspects of the process—including the prompt's structure and content—remain identical. Any observed performance differences can therefore be reliably attributed to the specific information removed, ensuring the validity of experimental conclusions.

## F.2 F2CAGENT

To transcend the limitations of a single-pass generation approach, we designed and implemented a more sophisticated agent-based model. This model emulates a human developer's workflow by

---

**System Prompt for Multi-modal FIGMA2CODE (JSON + Screenshot) Generation**

You are an expert front-end developer. Generate a complete HTML document using Tailwind CSS from the provided preprocessed Figma JSON and page screenshot. Adjust the generation strategy according to the preprocessing notes in the input contract. The output must preserve the visual fidelity of the design while improving responsiveness and maintainability.

The Figma JSON is the authoritative source for layout, spacing, sizing, colors, typography, borders, radii, shadows/effects, and asset bindings. The screenshot is only a secondary reference to resolve ambiguities or fill in missing details, and must never override explicit JSON values.

**Input Contract:**

- **Contains:** Figma JSON (structured), Page Screenshot (image).
- **Preprocessing Notes:**
    - The provided Figma JSON is preprocessed.
    - Originally, Figma JSON `imageRef` values were opaque IDs (e.g., `347ba7a7c57adabed33deffd9e936c9b285f611e`).
    - These have been replaced with actual local relative paths (e.g., `assets/foo.png`, `assets/bar.svg`).
    - Some vectors were merged and exported as local SVG files. Rectangle nodes were created to hold these SVGs, with their file paths written into `imageRef`.
    - Unnecessary properties have been removed to simplify the JSON.

**Output Specification:**

- **Format:** Valid HTML document only (no extra text before or after).
- **Example:**

```
1  <!DOCTYPE html>
2  <html lang="en">
3  <head> ... </head>
4  <body> ... </body>
5  </html>
```

- **Requirements:**
    - Return **only** the complete HTML document.
    - Do not wrap the output in JSON or any other structure.
    - The HTML must use Tailwind CSS classes and preserve the visual fidelity of the design.

**Constraints:**

1. **Mandatory Image Asset Binding:** For every visible node whose fills include an IMAGE with a non-empty, case-sensitive `imageRef` value (already a local relative path), you must render that exact asset path in the HTML. This is a hard requirement.
2. **Allowed Rendering for Image References:** Prefer `` for placed/raster/SVG assets. Use CSS `background-image` only when the design explicitly uses the image as a background fill. When using Tailwind arbitrary values, escape properly, e.g., `bg-[url('assets/foo.png')]`.
3. **Path Integrity:** Do not alter provided paths or their semantics.
4. **No Hallucinated Assets:** Do not introduce assets that are not explicitly specified in the JSON.
5. **External Resources:** No external URLs are permitted except the Tailwind CDN.

**Code Format:** The final output must be the complete HTML document.

Figure 19: System prompt for the multi-modal (JSON + Screenshot) modality, illustrating the explicit definition of input contract, preprocessing notes, and constraints.

decomposing the complex code generation task into a sequential and iterative process, encompassing initial drafting, self-critique, and refinement. The agent operates on a simplified, structured *Intermediate Representation* (IR) of the design, rather than the raw Figma JSON, which enhances the focus and efficiency of each step.

The agent's workflow is orchestrated as a pipeline comprising three principal stages:

**Stage 1: Design Abstraction to Intermediate Representation (IR).** The initial stage of the pipeline is responsible for converting the verbose and complex Figma JSON into a streamlined, task-oriented IR. This abstraction process serves two primary goals:

- **Simplification and Normalization:** It distills the raw design data, which contains a plethora of properties irrelevant to code generation, into a structured format that exclusively retains essential information, such as node hierarchy, geometry, styling attributes, and textual content.

- **Semantic Enhancement:** The process enriches the representation by making implicit design semantics explicit. For instance, it might identify layout patterns (e.g., flexbox *vs.* grid), group related elements, or resolve asset paths, thereby creating a more actionable "blueprint" for the subsequent generation stage.

This conversion to an IR effectively provides the agent with a simplified and more coherent "world model" of the design, allowing it to reason about the layout and components more effectively than it could with the raw source file.

**Stage 2: Initial Code Generation from IR.** Following the creation of the IR, the pipeline generates an initial HTML draft. Unlike other stages that leverage LMMs, this step is intentionally deterministic, employing a **rule-based templating engine**. This engine traverses the structured IR and maps each node to a corresponding HTML element with appropriate Tailwind CSS classes based on a predefined set of rules.

The primary objective of this rule-based approach is to produce a structurally sound, albeit potentially naive, first version of the code. This initial draft faithfully represents the hierarchy and basic attributes defined in the IR, providing a consistent and predictable foundation for the subsequent refinement stage. By starting with a deterministic baseline, we ensure that the creative and corrective capabilities of the LMM in the refinement stage are focused on improving a solid structural base, rather than generating code from scratch.

**Stage 3: Iterative Self-Correction and Refinement.** This stage embodies the agentic behavior of the model, where it iteratively improves upon its own work. The self-correction loop consists of two key personas: a **Critic** and a **Refiner**.

- **The Critic:** The agent invokes the Critic persona to perform a self-evaluation, guided by the prompt shown in Figure 20. The Critic is instructed to analyze the currently generated HTML code against the original design specification (represented by the IR and the page screenshot). Its task is to identify discrepancies, logical errors, or areas for improvement, such as layout inaccuracies or styling deviations. The output of the Critic is a structured list of actionable feedback, which serves as a clear, machine-readable guide for the Refiner. The inclusion of this critique step is a configurable option within our experimental setup, allowing us to study the impact of explicit self-evaluation on final code quality.
- **The Refiner:** The Refiner's goal is to improve the code, following the instructions in its dedicated prompt (Figure 21). It receives the current version of the HTML as its primary input. When the Critic is active, the Refiner is additionally provided with structured feedback. The prompt instructs the Refiner to apply the suggested fixes or, if no critique is provided, to independently identify and implement improvements based on its own understanding of best practices. The Refiner then outputs a new, revised version of the HTML code.

This critique-and-refine cycle can be executed for a predetermined number of iterations or until the model determines that no further improvements can be made. This iterative process allows the agent to systematically address errors and converge towards a higher-fidelity final output, mirroring the cycles of coding and debugging in human software development.

### F.3 UNIFIED INFERENCE PROTOCOLS FOR MLLMS

To ensure fairness and reproducibility, we adopt a unified inference setup across all evaluated MLLMs.

- **Image Input Resolution.** In direct prompting, the screenshot input resolution is set to **2× the size specified in the top-level Figma metadata container**. To reduce token length, images are uploaded to a cloud service and passed to the API as secure URLs.
- **API Access.** We query **10 MLLMs via the OpenRouter API**, without imposing an artificial cap on the maximum output tokens. This allows each model to utilize its full context length.
- **Input of Critic for F2CAGENT.** For the critic phase of **F2CAGENT**, we encode the rendered page screenshot into **base64** and provide it as input to the MLLM.
- **Temperature Settings.** In the critic phase, we set **temperature = 0.5** to encourage exploration and creativity. For all other cases, we set **temperature = 0.0** to improve determinism and reproducibility.

---

**Prompt for the Critic Persona**

You are an expert UI/UX critic and front-end developer. Your task is to analyze the provided HTML code against a design specification (given as an Intermediate Representation and a screenshot). Identify all discrepancies and areas for improvement. Your response must be a single, valid JSON object containing a list of issues. Do not include any surrounding text or explanations.

**JSON Schema (All fields are mandatory):**

- **critique (Array of Objects):** A list of issue objects. Each object must contain:
  - **issue_type (String):** The category of the issue. Examples: *Layout, Styling, Component, Missing Content, Accessibility*.
  - **description (String):** A clear and concise explanation of the discrepancy.
  - **suggestion (String):** A specific, actionable recommendation for how to fix the issue in the code.

**Instructions:**

1. Compare the visual rendering of the HTML with the provided screenshot to identify layout and style mismatches.
2. Cross-reference the HTML against the Intermediate Representation (IR) to verify correct implementation of specified properties (e.g., colors, fonts, spacing).
3. For each identified issue, provide a precise description and a concrete suggestion for correction.
4. If the code is already perfect and requires no changes, return a JSON object with an empty `critique` array.

**Example:**

```
{
  "critique": [
    {
      "issue_type": "Styling",
      "description": "The primary action button's background color is incorrect. It appears as a standard
          blue but should be a specific shade of purple as per the design.",
      "suggestion": "In the button's class list, change 'bg-blue-500' to 'bg-purple-600' to match the IR
          and screenshot."
    },
    {
      "issue_type": "Layout",
      "description": "The user profile icons in the header are vertically stacked instead of being
          aligned horizontally in a row.",
      "suggestion": "Ensure the container div for the profile icons has the 'flex' and 'flex-row'
          Tailwind CSS classes applied."
    }
  ]
}
```

Figure 20: The prompt used to instruct the Critic persona for self-evaluation.

- **Prompt Templates.** Within the same experiment, **all MLLMs use the same prompt templates**, ensuring consistent task framing and fair comparison.

# G ABLATION STUDY ON FIGMA METADATA COMPONENTS

## G.1 OBJECTIVE AND RATIONALE

The primary objective of this ablation study is to quantitatively assess the relative importance of different components within Figma metadata for the task of automated UI code generation. Our central research question is: **"Which categories of information within the structured Figma metadata are most critical for generating high-fidelity, production-quality front-end code?"**

The rationale behind this study is to deconstruct the multimodal nature of UI design artifacts and understand the model's reliance on each data modality. A design's source-of-truth contains not only a visual representation (the rendered image) but also rich, structured metadata detailing its geometric layout, styling, content, and component hierarchy. By systematically removing (ablating) specific categories of this metadata and observing the corresponding degradation in the quality of the generated code (e.g., measured by visual similarity scores or structural correctness), we can infer the model's dependency on each ablated component. This provides critical insights into the model's internal reasoning and highlights which metadata components are indispensable versus those that can be robustly inferred from other available information, such as the visual rendering.

---

**Prompt for the Refiner Persona**

You are an expert front-end developer specializing in code refactoring and correction. Your task is to revise and improve the provided HTML document based on a set of critiques. Your response must be only the complete, revised HTML document. Do not include explanations, apologies, or any text outside the `<!DOCTYPE html>...</html>` block.

**Inputs You Will Receive:**

- **Current HTML:** The full HTML code of the current implementation.

- **Critique (Optional):** A JSON object containing a list of specific issues and suggestions for improvement.

**Instructions:**

1. Carefully review the entire current HTML code.

2. If a `critique` JSON is provided, systematically address every issue listed. Apply the suggestions to correct the code.

3. If no `critique` is provided, perform a general review of the code. Independently identify and fix any potential issues related to layout, styling, semantic correctness, or component structure.

4. Ensure your final output is a single, complete, and valid HTML document that incorporates all necessary corrections.

**Example Output:**

```html
<!DOCTYPE html>
<html lang="en">
<head>
    <meta charset="UTF-8">
    <meta name="viewport" content="width=device-width, initial-scale=1.0">
    <script src="https://cdn.tailwindcss.com"></script>
    <title>Refined Page</title>
</head>
<body>
    <!-- ... The full, corrected, and improved HTML content ... -->
    <button class="bg-purple-600 text-white px-4 py-2 rounded">Submit</button>
    <!-- ... etc. ... -->
</body>
</html>
```

Figure 21: The prompt used to instruct the Refiner persona for code correction and improvement.

## G.2    EXPERIMENTAL DESIGN

To investigate this, we designed a series of controlled experiments. Each experiment involves ablating one fundamental dimension of the design information from the input provided to the model. We also include a baseline group, which receives the complete, unaltered metadata, serves as the control to establish the model's optimal performance. The experimental groups are defined in Table 4.

## G.3    IMPLEMENTATION METHODOLOGY

To facilitate these experiments, we developed a programmatic data processing pipeline to generate the required input files for each ablation group from a single source Figma JSON file.

### G.3.1    TRANSFORMATION PIPELINE

The pipeline takes the file path of a source JSON as input and applies a series of transformation functions, each corresponding to one of the ablation experiments. For each transformation, it produces a new JSON file where the targeted information has been systematically removed, ensuring a consistent and repeatable process for every sample in our dataset.

### G.3.2    TRANSFORMATION LOGIC FOR EACH ABLATION GROUP

The logic for each ablation is as follows:

- **Geometric Layout Ablation.** The transformation recursively traverses the entire JSON tree and removes all keys related to geometric properties. This includes not only node-level attributes like `x` and `width` but also nested structures such as `absoluteBoundingBox` and transform matrices. The process also cleans up references to these properties in component override lists to ensure their complete removal.

Table 4: Experimental groups for the ablation study.

| Group ID | Ablated Information | Core Research Question Investigated |
|---|---|---|
| **Baseline** | None (complete metadata) | What is the upper-bound performance of the model with full information? |
| **Ablation A** | **Geometric Layout.** `x`, `y`, `width`, `height`, `absoluteBoundingBox`, `transforms`, etc. | Can the model accurately reconstruct the layout solely from visual cues in the rendered image? |
| **Ablation B** | **Visual Styling.** Colors, fonts, effects, corner radii, strokes, opacity, etc. | Can the model correctly infer visual styles (e.g., colors, font sizes) by "sampling" from the image pixels? |
| **Ablation C** | **Image Content.** `imageRef`, `imageHash`, and other references to external image assets. | How critical are explicit image references for correctly rendering image elements, as opposed to treating them as generic placeholders? |
| **Ablation D** | **Hierarchical Structure.** The nested parent-child relationships of all nodes. | How important is the explicit DOM hierarchy for achieving a correct and maintainable layout, versus a "flat" structure? |
| **Ablation E** | **Textual Content.** `characters`, `text`, and other fields containing literal string content. | Does the semantic content of text nodes influence the generation of layout and styling for surrounding elements? |

- **Visual Styling Ablation.** This process performs a comprehensive removal of all styling-related information. The set of ablated keys includes `fills`, `strokes`, `effects`, font properties (`fontName`, `fontSize`, etc.), `cornerRadius`, `opacity`, and references to style libraries (e.g., `fillStyleId`). The removal is applied recursively to ensure that style information is purged from all parts of the JSON, including style override tables.

- **Image Content Ablation.** This transformation targets all references to external image assets. It recursively removes keys such as `imageRef` and `imageHash`. Furthermore, it inspects `fills` and `background` properties and filters out any paint objects of type `IMAGE`, effectively leaving image-filled shapes as empty containers.

- **Hierarchical Structure Ablation**. This process fundamentally alters the node hierarchy to create a flat structure. It operates by first traversing the entire JSON to collect a unique set of all nodes (identified by their `id`). It then reconstructs the `children` list of the root node, populating it with this flattened collection of all other nodes. The `children` property of every node in this new flat list is then explicitly emptied, resulting in a structure with a maximum depth of one.

- **Textual Content Ablation.** This transformation removes all semantic textual content. It deletes the `characters` field from all nodes and, for nodes of type `TEXT`, also removes the `text` field and anonymizes the node's `name` to prevent semantic leakage. The process also clears text values within component properties and removes text-related fields from override lists.

This methodological rigor ensures that each experimental condition is cleanly isolated, allowing for a valid interpretation of the results.

## H MAE_px versus MOS

## I More Experimental Results

In the main text, we report a subset of evaluation metrics for clarity and readability. Here, we provide more experimental results of our benchmarking experiments, including all measured attributes from the raw evaluation files. These results allow for a more detailed examination and potential reproduction of our findings.

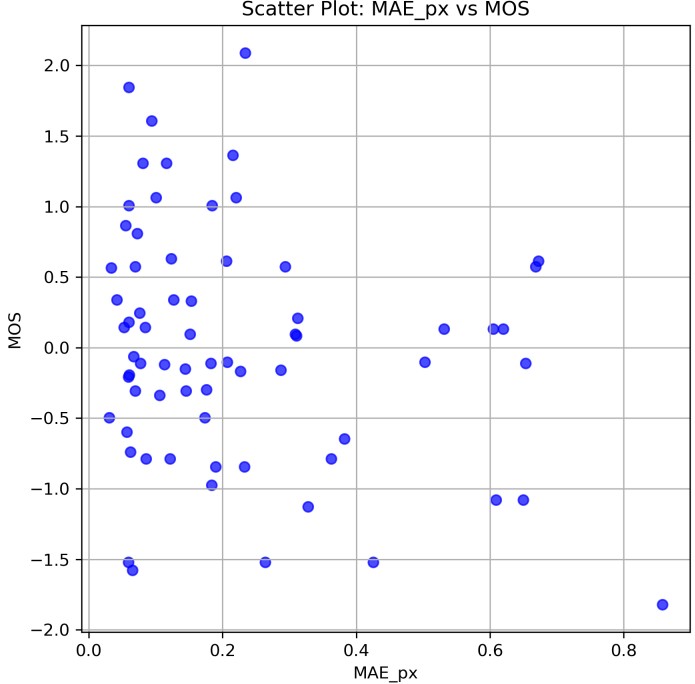

Figure 22: Scatter plot between pixel-level Mean Absolute Error (MAE_px) and Mean Opinion Score (MOS). MAE_px quantifies per-pixel luminance/color deviations between paired images (lower is more similar), whereas MOS reflects human-perceived structural and semantic similarity (higher is more similar). The dispersion illustrates a systematic mismatch between pixel-wise error metrics and perceptual judgments, especially in cases dominated by background intensity differences or localized content shifts.

Table 5: Complete responsiveness results for multimodal code generation models.

| Model | RUR (% ↑) | BC (% ↑) | FU (% ↑) | APR (% ↓) |
|---|---|---|---|---|
| Llama 4 Maverick | 3.30 | 6.72 | **50.37** | 6.09 |
| Llama 4 Scout | 4.72 | **26.06** | 45.89 | **1.42** |
| Qwen 2.5 VL | 4.40 | 8.24 | 42.07 | 2.66 |
| ERNIE 4.5 424B VL | **4.81** | 10.43 | 40.61 | 2.83 |
| Nova Pro v1 | 4.59 | 6.88 | 35.14 | 3.84 |
| Gemini 2.5 Pro | 4.43 | 18.16 | 50.05 | 10.51 |
| Grok4 | 2.30 | 5.97 | 34.95 | 31.09 |
| Claude Opus 4.1 | 1.05 | 2.62 | 42.98 | 9.62 |
| GPT-4o | 3.72 | 1.99 | 46.88 | 2.94 |
| GPT-5 | 1.73 | 17.68 | 35.64 | 14.35 |

## J CASE STUDY

Table 6: Complete maintainability results for multimodal code generation models.

| Model | STR (% ↑) | ISR (% ↓) | AVU (% ↓) | CCR (% ↑) |
|---|---|---|---|---|
| Llama 4 Maverick | 25.28 | **0.16** | 7.71 | 42.94 |
| Llama 4 Scout | **35.76** | 0.33 | 0.87 | **48.71** |
| Qwen 2.5 VL | 29.54 | 0.37 | **0.15** | 44.05 |
| ERNIE 4.5 424B VL | 32.03 | 0.61 | 2.06 | 34.68 |
| Nova Pro v1 | 29.28 | 0.71 | 1.53 | 42.87 |
| Gemini 2.5 Pro | 28.98 | 3.42 | 25.46 | 47.34 |
| Grok4 | 13.88 | 6.91 | 49.97 | 53.52 |
| Claude Opus 4.1 | 19.79 | 1.92 | 23.29 | 40.44 |
| GPT-4o | 25.21 | **0.16** | 2.46 | 34.89 |
| GPT-5 | 15.37 | 4.21 | 37.72 | 46.44 |

Table 7: Complete responsiveness results for different approaches.

| Method | Input | RUR (% ↑) | BC (% ↑) | FU (% ↑) | APR (% ↓) |
|---|---|---|---|---|---|
| Direct Prompting | 🖼 | 6.14 | 0.00 | 0.52 | **0.01** |
| Text-Augmented | 🖼 | **7.02** | 0.00 | 0.43 | **0.01** |
| Direct Prompting | </> | 4.97 | 10.08 | **40.58** | 5.01 |
| Template Conversion | </> | 0.00 | 0.00 | 13.67 | 58.0 |
| Direct Prompting | 🖼 + </> | 4.81 | **10.43** | 40.61 | 2.83 |
| F2CAGENT | 🖼 + </> | 4.69 | 5.75 | 36.63 | 13.57 |

Table 8: Complete maintainability results for different approaches.

| Method | Input | STR (% ↑) | ISR (% ↓) | AVU (% ↓) | CCR (% ↑) |
|---|---|---|---|---|---|
| Direct Prompting | 🖼 | 31.86 | 2.68 | **0.00** | 25.69 |
| Text-Augmented | 🖼 | 29.67 | 2.46 | **0.00** | 25.75 |
| Direct Prompting | </> | 21.46 | 2.17 | 6.09 | 36.49 |
| Template Conversion | </> | 0.01 | **0.00** | 24.94 | **63.32** |
| Direct Prompting | 🖼 + </> | **32.03** | 0.61 | **2.06** | 34.68 |
| F2CAGENT | 🖼 + </> | 28.57 | 0.28 | 16.71 | 38.7 |

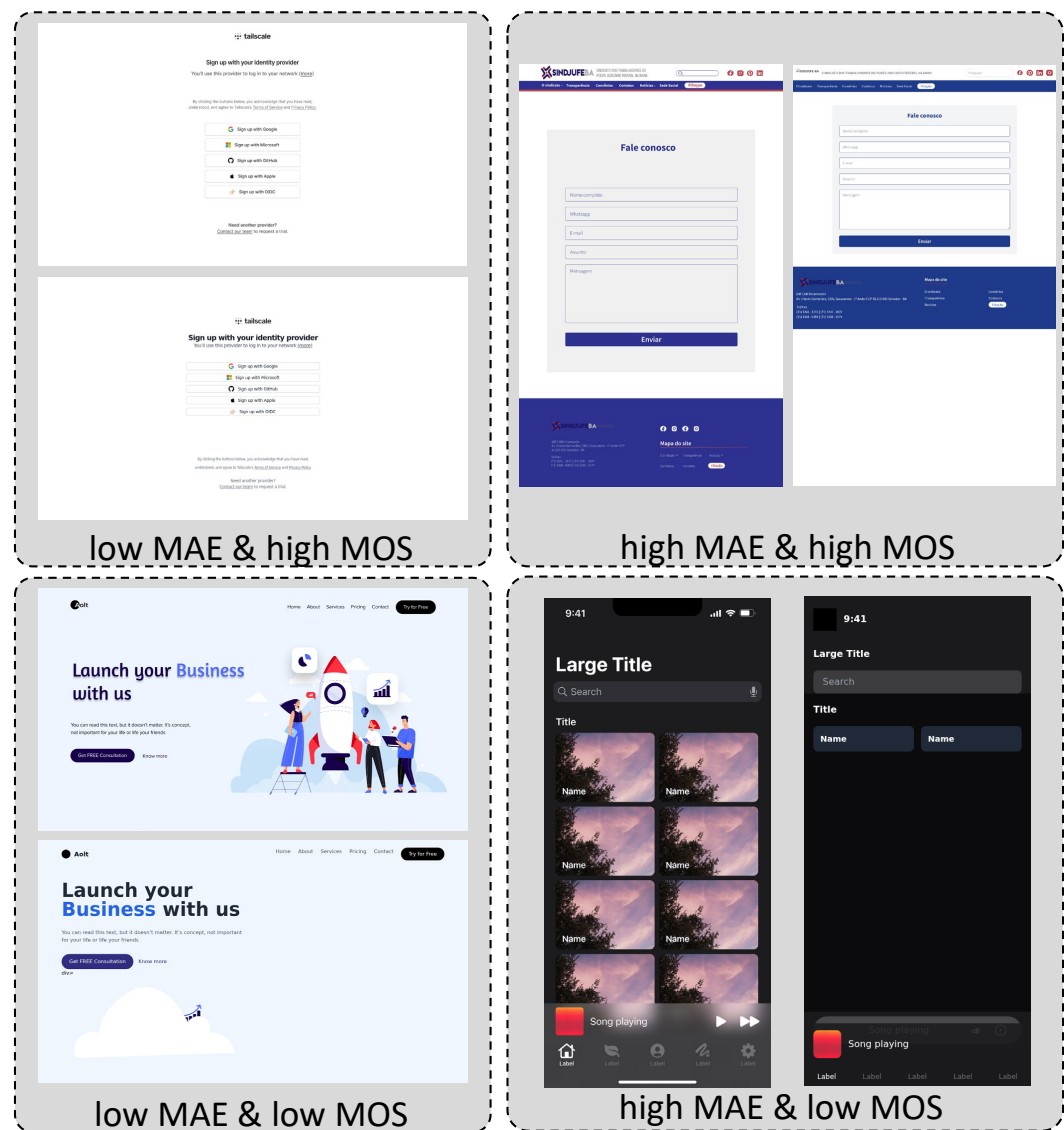

Figure 23: Qualitative cases summarizing the relationship between MAE_px and MOS. (i) **Low MAE, low MOS**: Large structural misalignment or missing content yields poor perceptual similarity, yet MAE remains small when displaced/removed regions share similar background chromaticity or low contrast; pixel-wise differences are masked by homogeneous backgrounds. (ii) **High MAE, high MOS**: Global page layout and semantics are preserved (high MOS), but extensive background color shifts (e.g., dark blue versus white) introduce large, spatially widespread intensity deltas, inflating MAE despite perceptual similarity. (iii) **Low MAE, high MOS** (desired): Minimal pixel deviations with large uniform backgrounds and small foreground components; structural alignment and content consistency lead to strong agreement between MAE and MOS. (iv) **High MAE, low MOS**: Prominent, vividly colored foreground regions (high saturation/low luminance) occupy a substantial area; local additions/removals or misplacements produce large pixel-wise errors and poor perceptual similarity. Overall, MAE is highly sensitive to global intensity and color distribution (background dominance and area coverage), while MOS is driven by structural alignment, object presence, and semantic plausibility. This divergence motivates complementing pixel-level metrics with perceptual or structure-aware measures for robust similarity assessment.

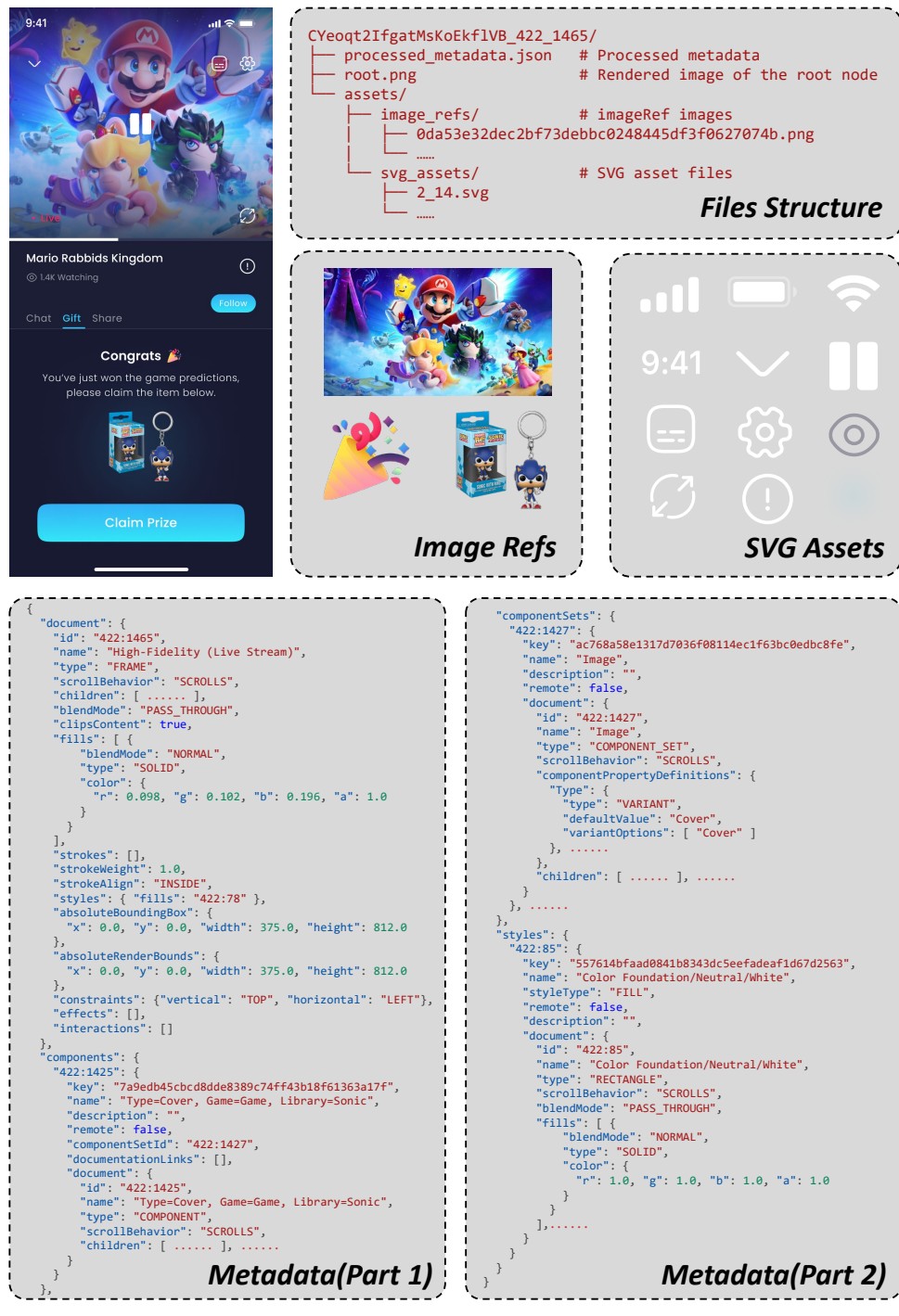

Figure 24: Illustration of a Dataset Sample.

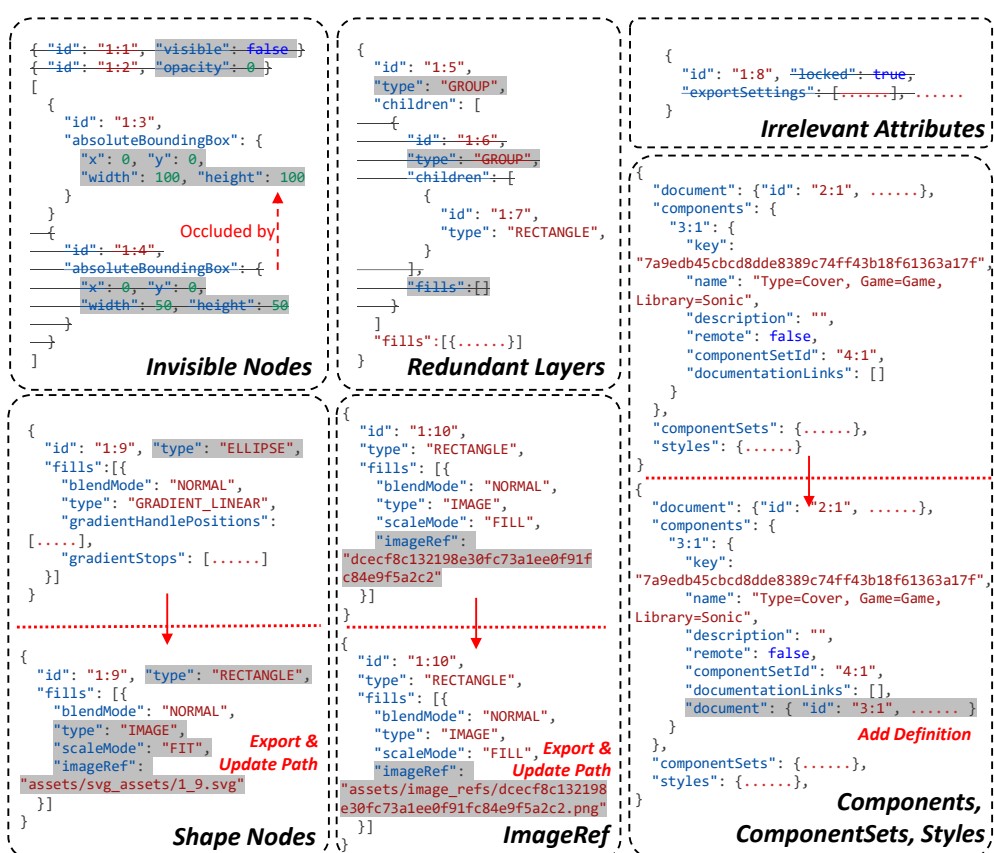

Figure 25: Illustration of Metadata Refinement.

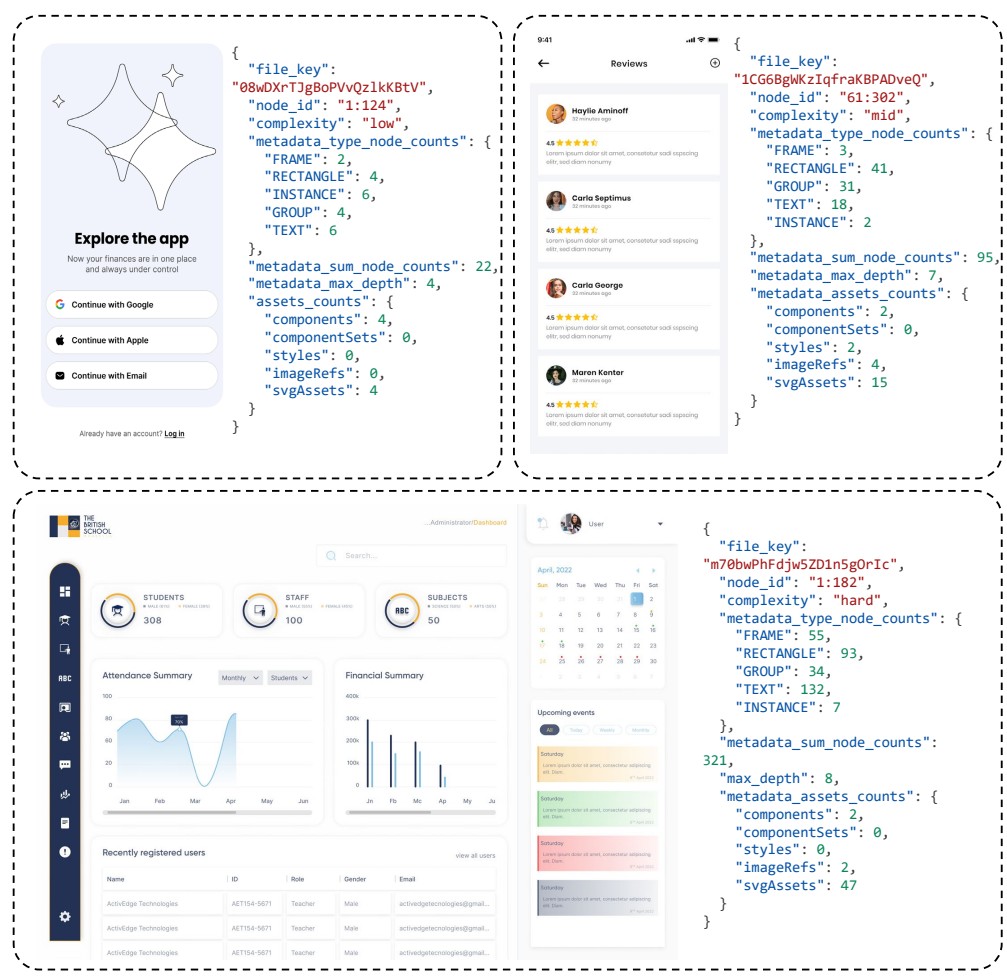

Figure 26: Illustration of Samples of different Complexities.

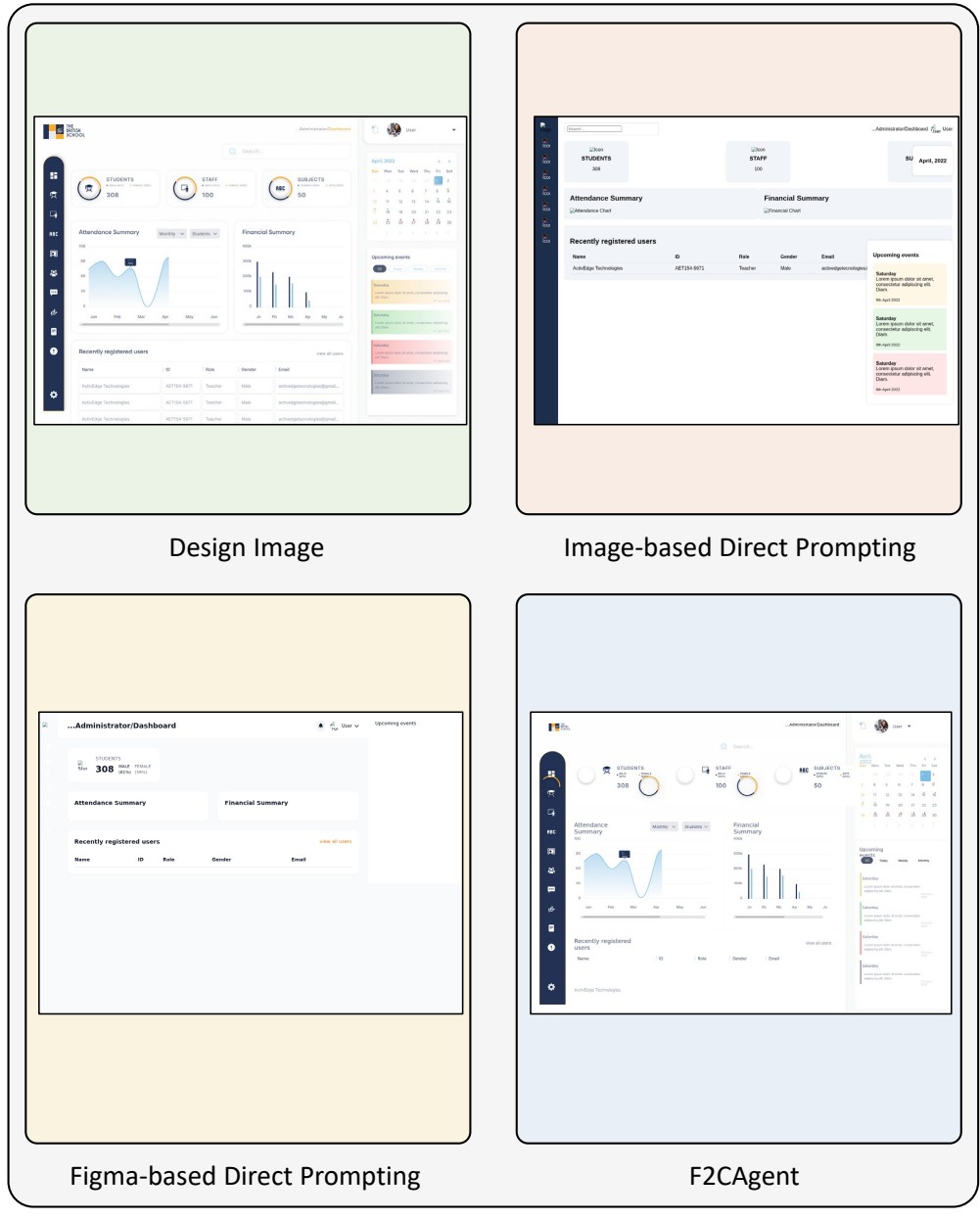

Figure 27: Case Study 1.

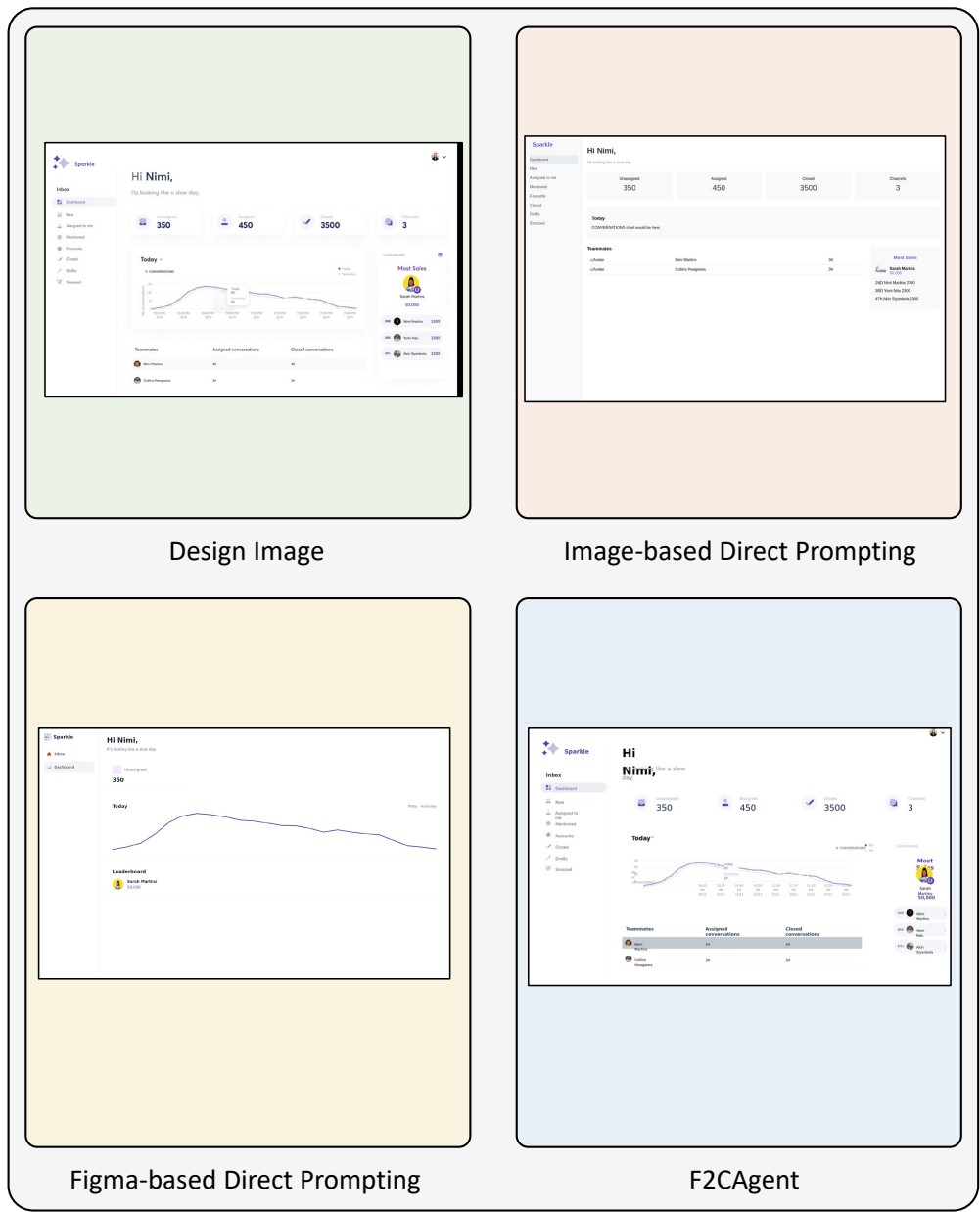

Figure 28: Case Study 2.

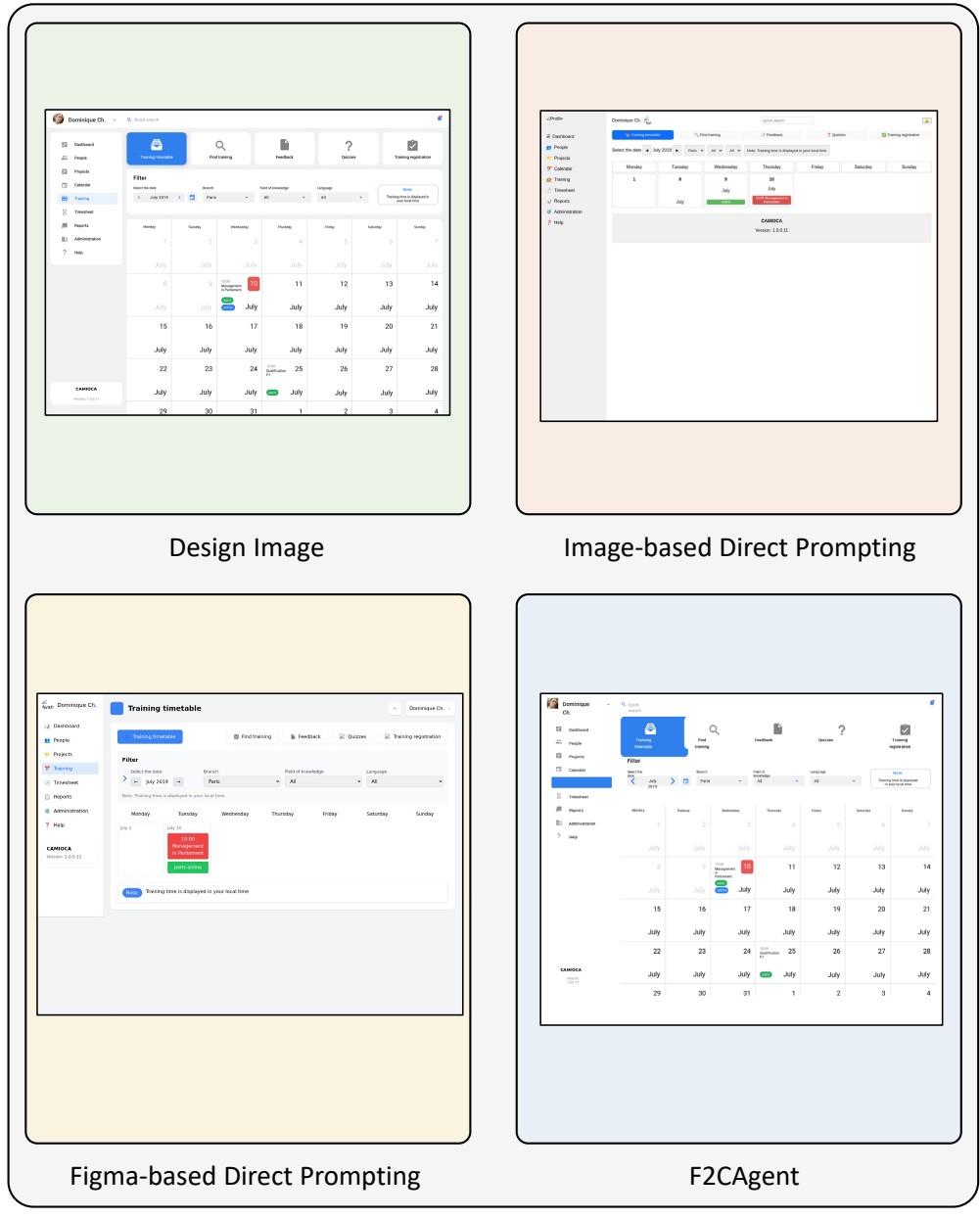

Figure 29: Case Study 3.

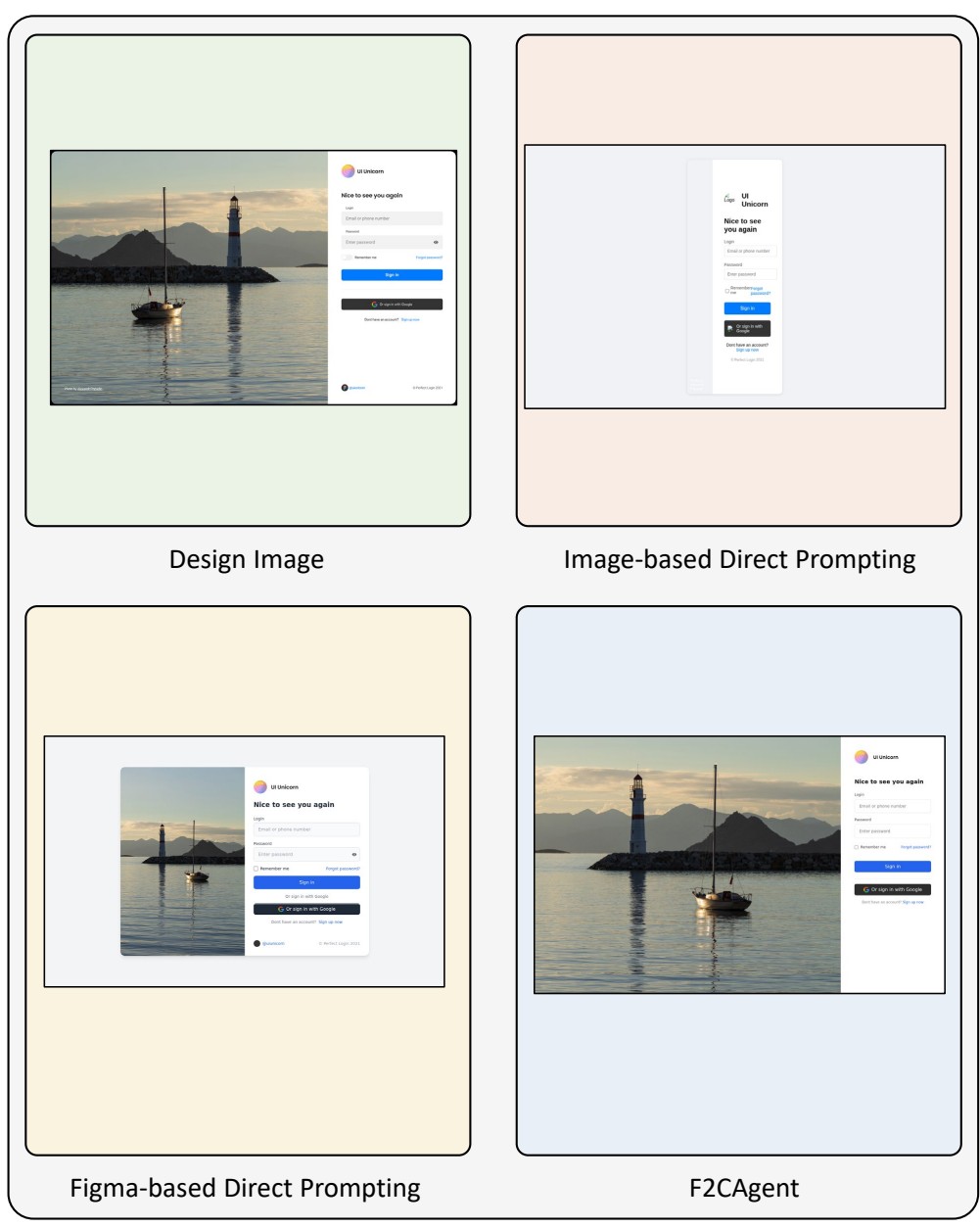

Figure 30: Case Study 4.

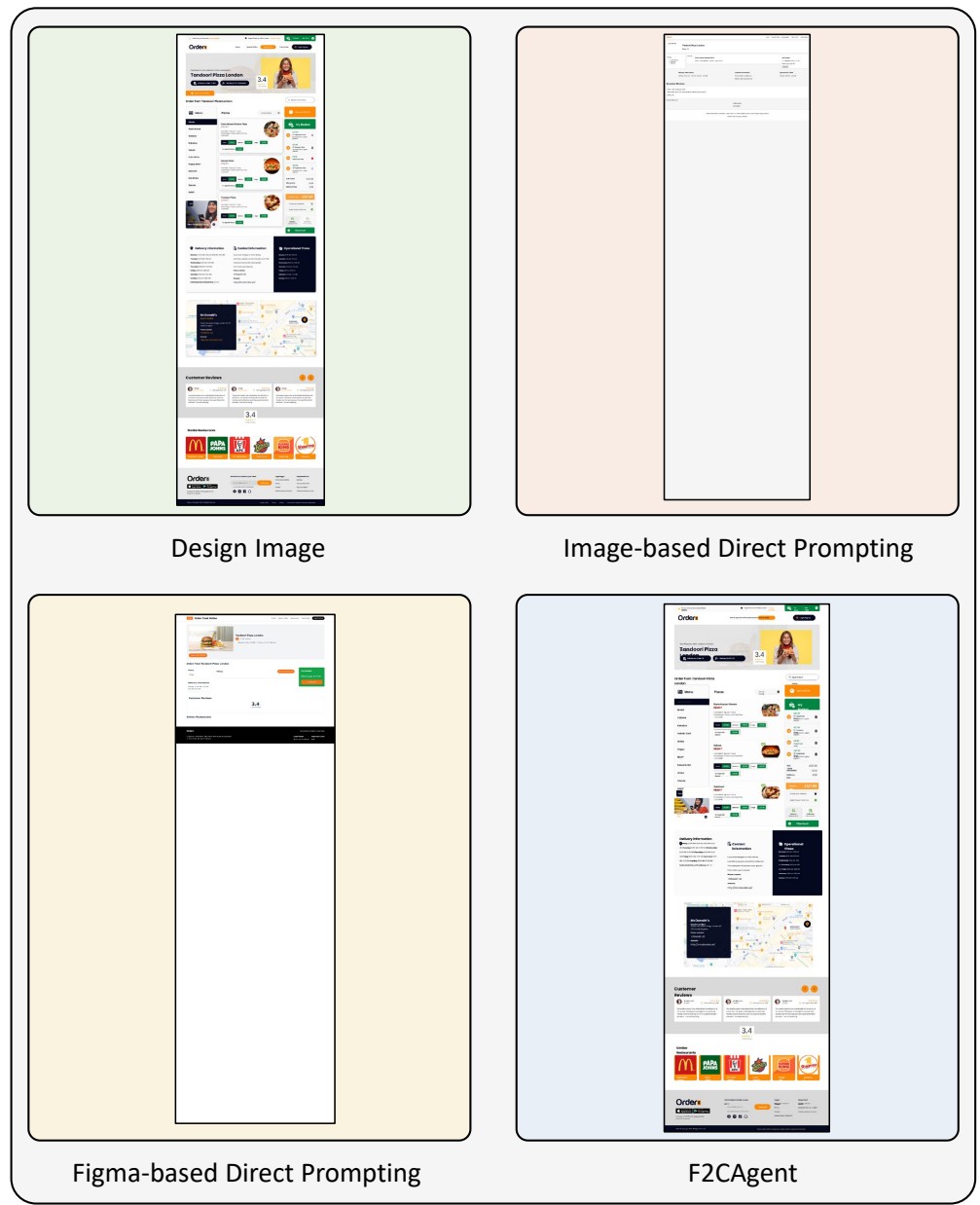

Figure 31: Case Study 5.

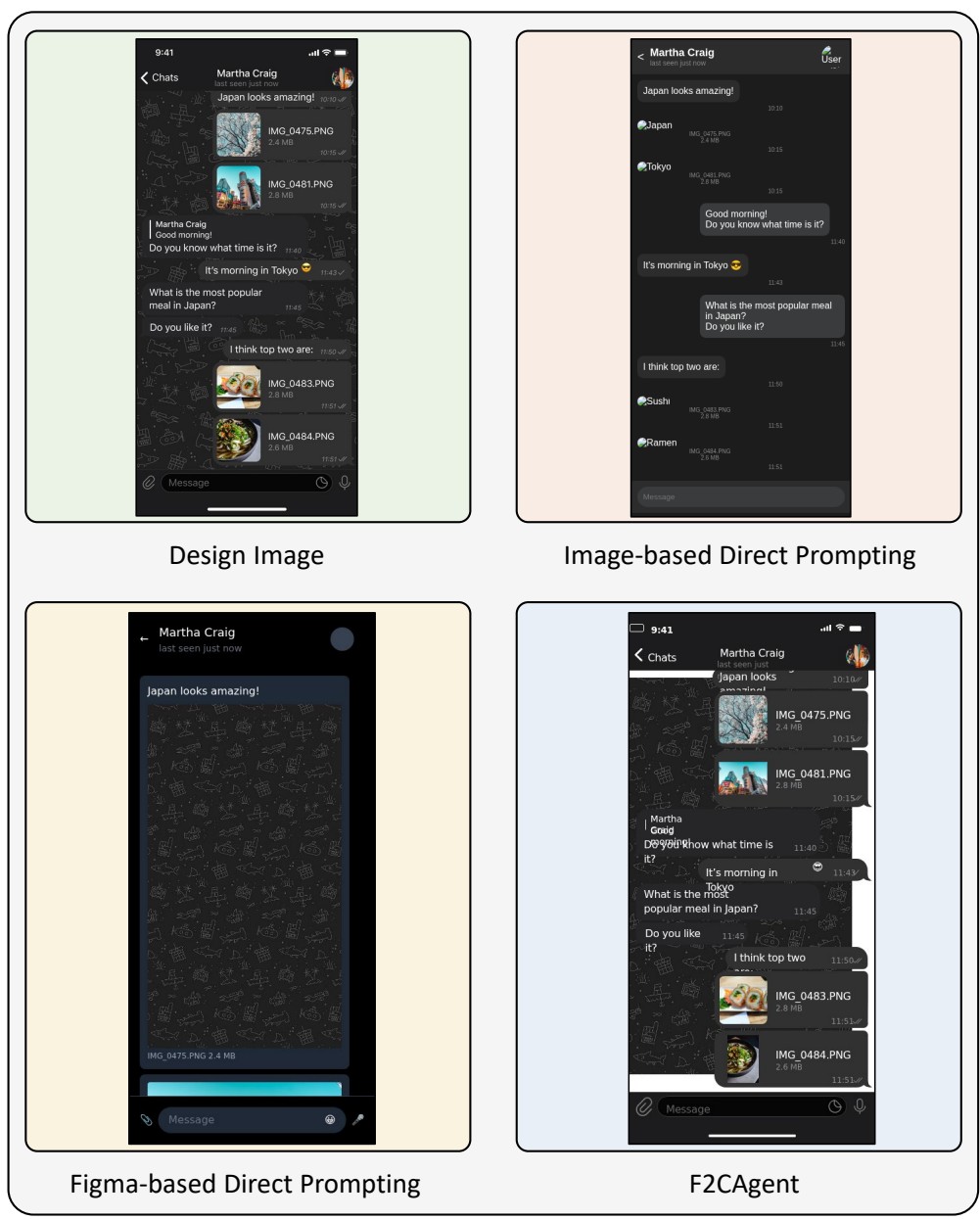

Figure 32: Case Study 6.

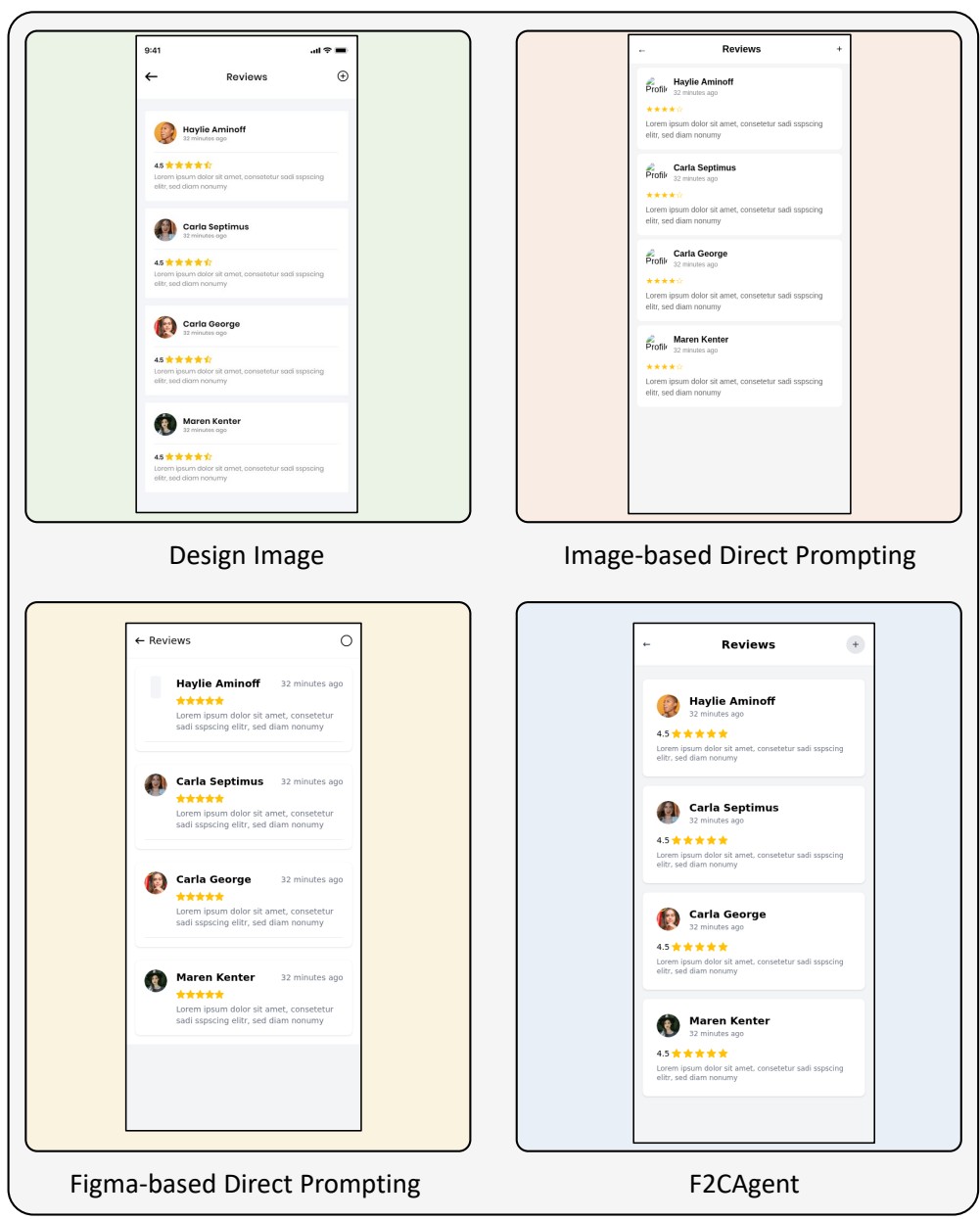

Figure 33: Case Study 7.

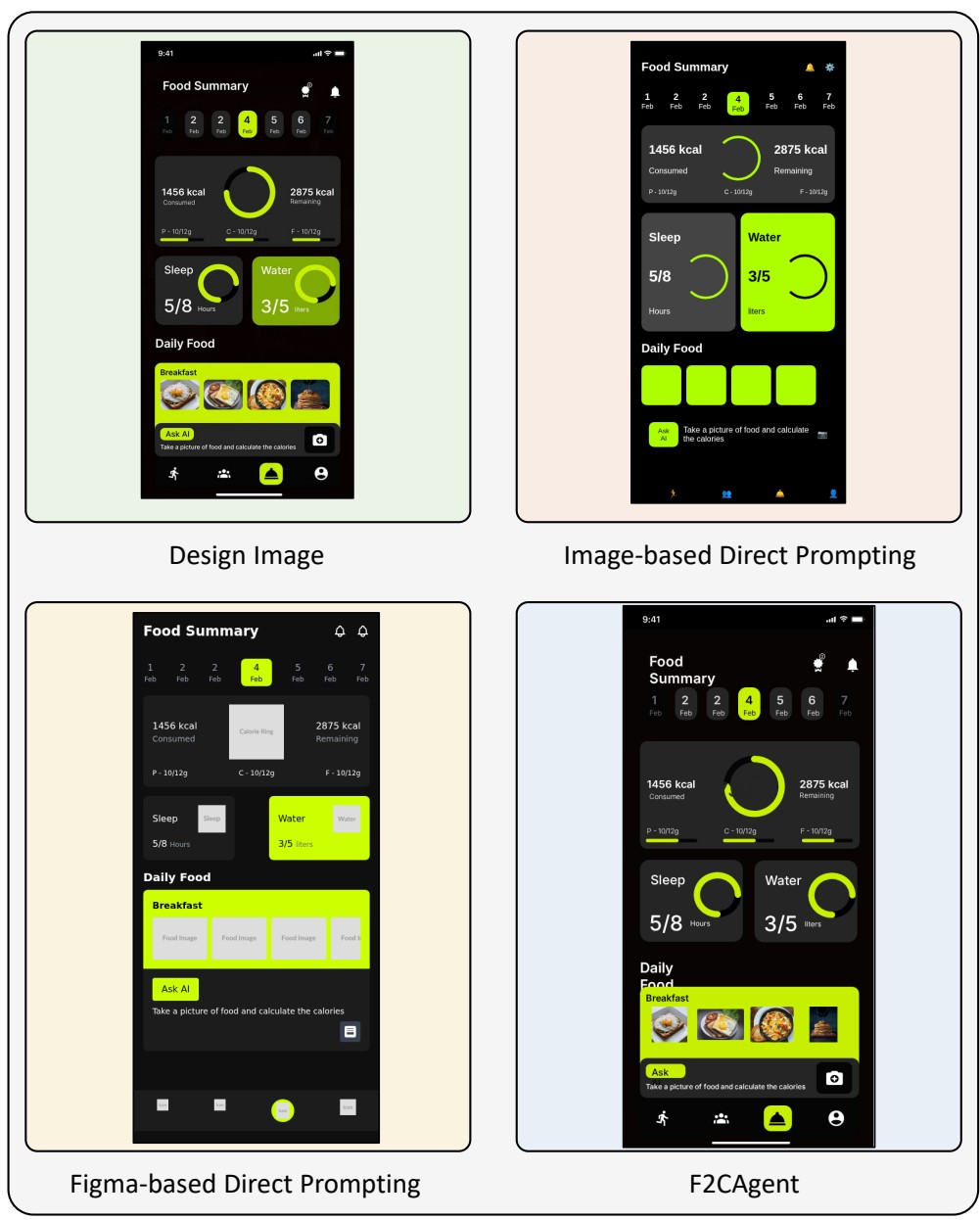

Figure 34: Case Study 8.

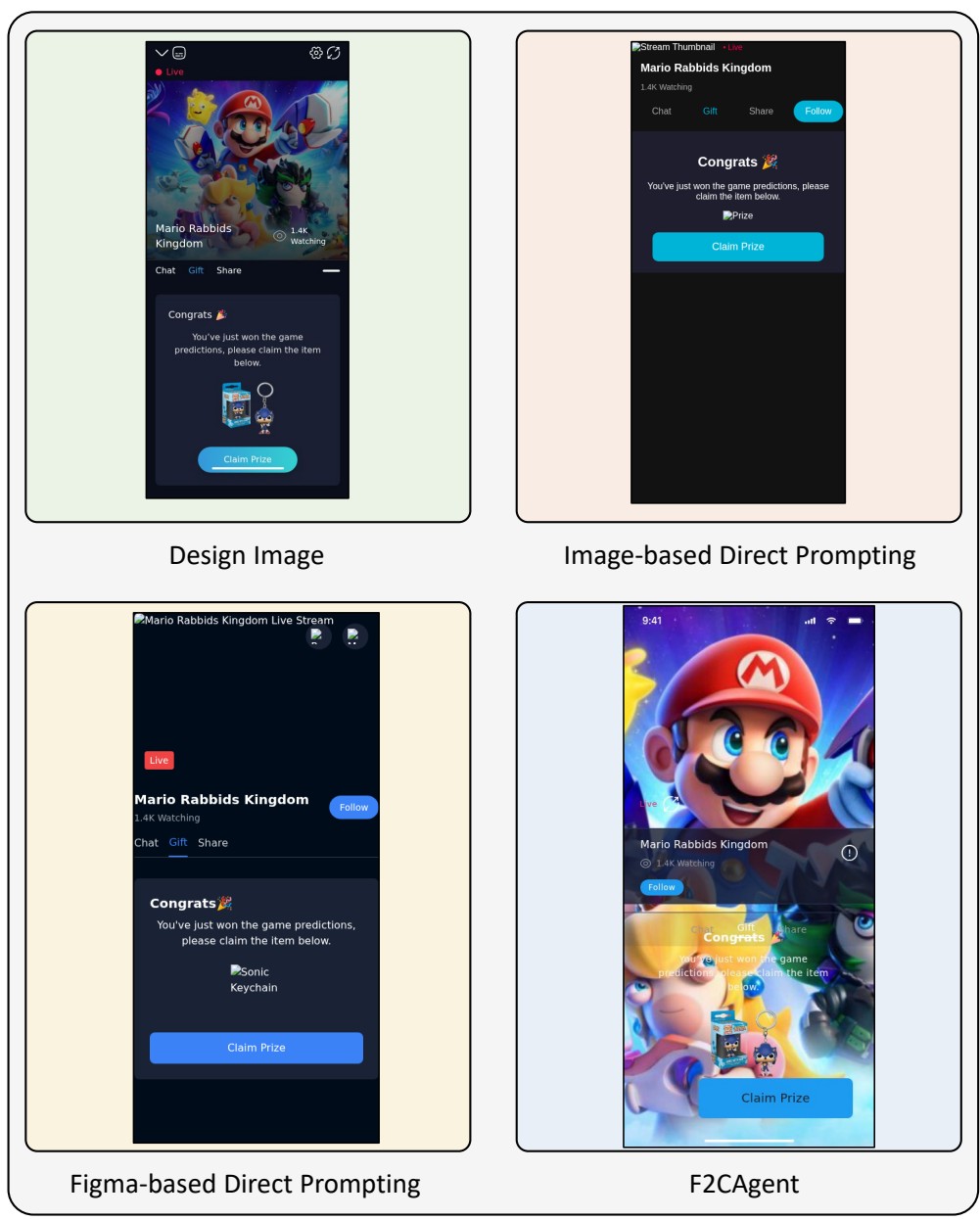

Figure 35: Case Study 9.

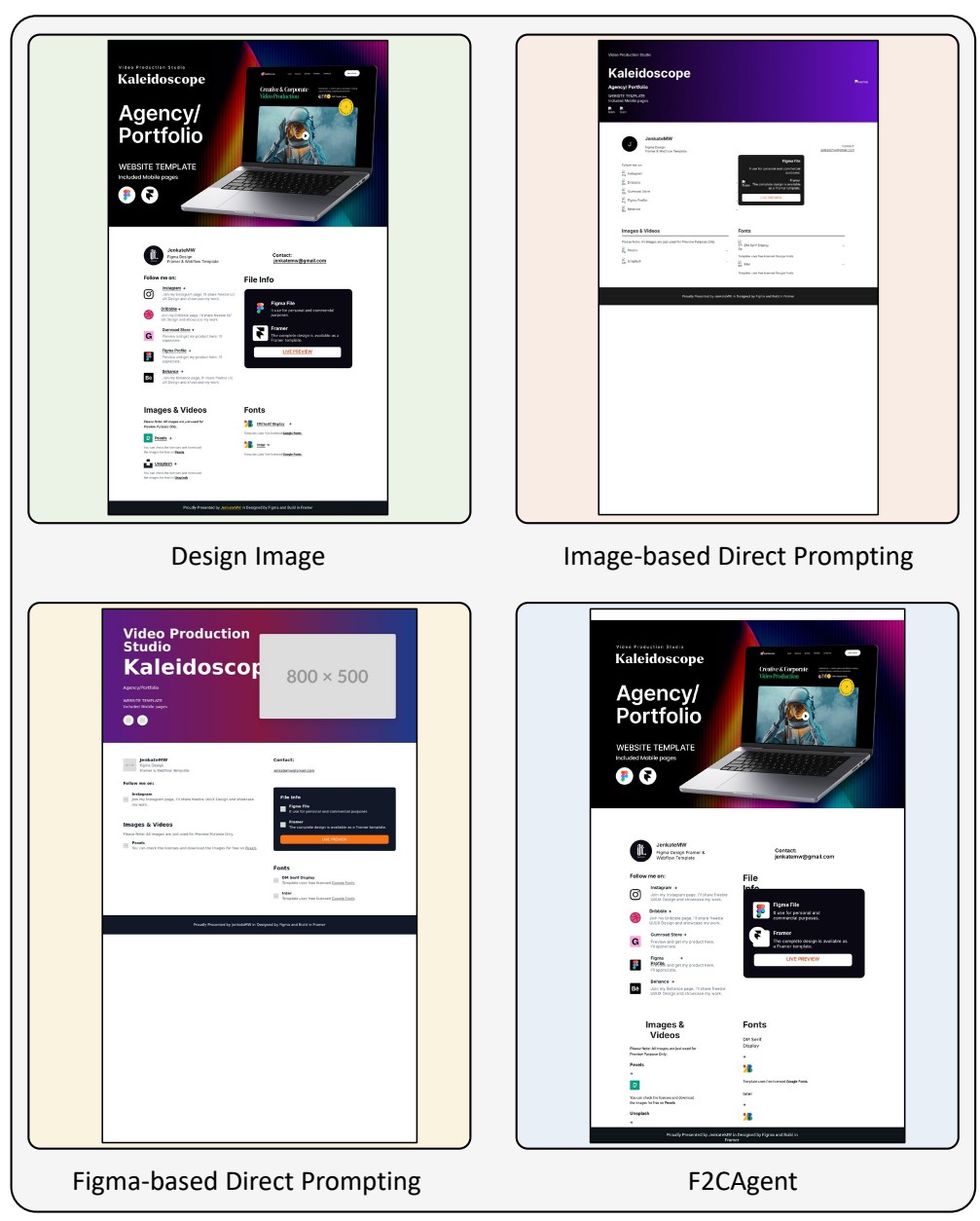

Figure 36: Case Study 10.

