# OpenReview forum: "Figma2Code: Automating Multimodal Design to Code in the Wild"
_ICLR.cc/2026/Conference — ICLR 2026 Poster_

### Official Review · Reviewer_NgVv · 2025-10-21

**Soundness:** 3
**Presentation:** 3
**Contribution:** 2
**Rating:** 4
**Confidence:** 4

**Summary:**

This paper constructs a new benchmark called Figma2Code for the task of converting visual front-end designs into code implementations.

The main difference as compared to prior works is that prior works only considered image inputs (like screenshots of the webpages) while Figma2Code inputs also include images, assets, and metadata, such that the model can possibly reconstruct the full UI from the input.

The raw data are crawled from the Figma user community with multiple stages of filtering and additional metadata refinement. The final benchmark contains 213 processed examples.

The evaluation metrics cover three aspects, including visual fidelity, layout responsiveness, and code maintainability. The authors benchmarked various vision-language foundation models as well as various scaffolds including a simple F2CAgent that does better than naive prompting baselines.

**Strengths:**

- I think including real Figma data as the input is a valuable contribution to the community. And I'm reasonably convinced that the authors have produced a high-quality benchmark (despite the final benchmark being a bit small in size).

- The F2CAgent seems to be a simple but effective agent scaffold for this task.

- Quite thorough experiments.

**Weaknesses:**

- I think you should try to include a few actual examples in the main paper to illustrate both what the benchmark examples look like and also how good the model generations are (I think this would be much more informative than your current Figure 6).

- I'm a little worried about the set of evaluation metrics that you have selected. For example, from Table 2, it looks like these metrics don't necessarily agree with each other (even for metrics in the same category). How do you reconcile this? It'd be nice to have a small-scale human eval to see how well these metrics correlate with human judgement (especially the visual fidelity metrics).

- Do you have more in-depth explanations for why "current MLLMs still struggle to balance visual fidelity with code quality"? Do you think there's a way to finetune base models to get better at both?

- While I recognize the value of this new benchmark, I also recognize that the contribution could be limited in terms of novelty. I'm therefore leaning towards a borderline score.

**Questions:**

- What are you doing with the 2,842 auxiliary samples? Are there finetuning experiments on them?

---

> ### Author Response · Authors · 2025-11-22
> **Response to Reviewer NgVv [1/5]**
>
> ## On the Key Contribution of Our Work
>
> **Weaknesses 4**
> > *While I recognize the value of this new benchmark, I also recognize that the contribution could be limited in terms of novelty. I'm therefore leaning towards a borderline score.*
>
> **Answer**
>
> We sincerely‌ appreciate that you recognize the value of this new benchmark, but would like to clarify that our contribution goes beyond releasing a Figma benchmark. **Our work could be meaningful for moving the design-to-code research agenda toward a more realistic and practically grounded setting—an important step toward fully automated UI generation in the wild**. Prior work in UI code generation has focused mainly on screenshot-to-code scenarios, which only support UI replication from existing UIs rather than the workflows in real development. In contrast, our benchmark evaluates MLLMs on end-to-end code generation from multimodal designs—including rendered visuals, hierarchical structure, geometric attributes, constraints, and style metadata.
>
> Beyond the dataset itself, our work provides:
> (1) a new task definition grounded in practical design workflows;
> (2) a stratified and expert-curated sample construction methodology;
> (3) a comprehensive benchmarking across ten leading MLLMs;
> (4) a maintainability- and responsiveness-oriented metric suite (RUR, APR, AVU, STR, plus FU/BC/ISR/CCR).
> (5) an in-depth analysis of the influence of Figma metadata
>
> Together, these reveal several previously unreported failure modes in current MLLMs—such as geometry “copy-through,” rigid absolute layouts, and poor abstraction of layout semantics—which appear consistently across proprietary and open-source models (Fig. 6, Table 3). We believe these findings highlight important capability gaps and help establish a foundation for future research on more robust design-to-code models.
>
> We hope this clarifies the key novelty of our work. The contribution lies not only in releasing a dataset, but in establishing a well-defined, practically grounded benchmark for end-to-end multimodal design-to-code generation—something that, to our knowledge, has not been systematically formulated or evaluated before. We hope our work offers a meaningful step toward advancing research on realistic UI code generation.

---

> > ### Author Response · Authors · 2025-11-22
> > **Response to Reviewer NgVv [2/5]**
> >
> > ## On the use of 2,842 Auxiliary Samples
> >
> > **Question 1**
> > > *What are you doing with the 2,842 auxiliary samples? Are there finetuning experiments on them?*
> >
> > **Answer**
> >
> > We selected a balanced 213-sample core set to keep evaluation feasible—running only these cases already takes more than six hours on GPT-5 in a single-thread setting. The remaining 2,842 samples are reserved for more extensive evaluation or future research needs. We do not perform any finetuning in this work; all results are based on training-free inference.

---

> > ### Comment · Reviewer_NgVv · 2025-11-25
> >
> > Thanks for the detailed responses. I've raised my overall score to 6.

---

> > > ### Author Response · Authors · 2025-11-25
> > >
> > > Thank you for raising the score. We appreciate your constructive feedback and wish you all the best.

---

> ### Author Response · Authors · 2025-11-22
> **Response to Reviewer NgVv [3/5]**
>
> ## On Including More Qualitative Examples
>
> **Weakness 1**
> > *I think you should try to include a few actual examples in the main paper to illustrate both what the benchmark examples look like and also how good the model generations are (I think this would be much more informative than your current Figure 6).*
>
> **Answer**
>
> Thank you for the suggestion.
>
> In fact, we already provide several representative benchmark cases and corresponding model outputs in Appendix I, because the high-resolution design images are too large to display clearly in the main text. Figure 6 is used to highlight our key finding—the limitations of proprietary MLLMs on the FIGMA2CODE task—within the limited available space. Following your suggestion, we now include explicit references in the main text and add an additional figure (Fig. 24 in Appendix) to show dataset examples.

---

> ### Author Response · Authors · 2025-11-22
> **Response to Reviewer NgVv [4/5]**
>
> ## On Metrics and Human Evaluation
>
> **Weakness 2**
> > *I'm a little worried about the set of evaluation metrics that you have selected. For example, from Table 2, it looks like these metrics don't necessarily agree with each other (even for metrics in the same category). How do you reconcile this? It'd be nice to have a small-scale human eval to see how well these metrics correlate with human judgement (especially the visual fidelity metrics.*
>
> **Answer**
>
> Thank you for the thoughtful question.
>
> To avoid redundancy, we provide a unified discussion on our metrics and human evaluation in the global comment titled **“More Studies about Metrics and Human Evaluation”**. Please kindly refer to that section for the detailed analysis.

---

> ### Author Response · Authors · 2025-11-22
> **Response to Reviewer NgVv [5/5]**
>
> ## On Why Current MLLMs Struggle
>
> **Weakness 3**
> > *Do you have more in-depth explanations for why "current MLLMs still struggle to balance visual fidelity with code quality"? Do you think there's a way to finetune base models to get better at both?*
>
> **Answer**
>
> Thank you for the insightful question.
>
> Our results provide concrete evidence for why current MLLMs still struggle to improve visual fidelity and code quality simultaneously.
>
> - **First**, the quantitative results reveal a consistent trade-off. As shown in Table 3, proprietary MLLMs achieve strong visual performance (VES/MAE) but systematically underperform on responsiveness and maintainability metrics (RUR, APR, AVU, STR). Models that excel at fidelity tend to generate code with more absolute positioning, more fixed pixel values, and fewer relative units.
>
> - **Second**, the qualitative examples in Fig. 6 make this trade-off explicit. Many models “copy through’’ geometric metadata by translating x/y/width/height directly into `position: absolute`, fixed sizes, and large numbers of Tailwind arbitrary values. While this preserves local geometry, it leads to rigid, hard-to-maintain layouts—exactly the failure modes captured by our metrics.
>
> - **Third**, our additional prompt-variation experiment further supports this interpretation (the table below). We strengthened the prompt with explicit, rule-level instructions encouraging flex/grid-first layouts, reduced wrappers, fewer arbitrary values, and better tag semantics. As shown in the table below, these enhanced prompts produce only marginal changes across most metrics, suggesting that the difficulty is not simply a lack of guidance; the model behavior remains strongly biased toward geometry-preserving but rigid patterns.
>
> | Method                            | VES (↑) | MAE (↓) | RUR (↑) | APR (↓) | STR (↑) | AVU (↓) |
> |-----------------------------------|---------|---------|---------|---------|---------|---------|
> | **Direct Prompting (metadata only)**     | 0.6801  | 0.2101  | 4.97    | 5.01    | 21.46   | 6.09    |
> | + Enhanced Prompting                          | 0.6589  | 0.1562  | 2.05    | 3.99    | 22.15   | 3.29    |
> | Δ                                 | -0.0212 | -0.0539 | -2.92   | -1.02   | +0.69   | -2.80   |
> |                                   |         |         |         |         |         |         |
> | **Direct Prompting (metadata + image)**                  | 0.6923  | 0.2228  | 4.81    | 2.83    | 32.03   | 2.06    |
> | + Enhanced Prompting                          | 0.7112  | 0.1509  | 4.33    | 2.75    | 31.91   | 1.72    |
> | Δ                                 | +0.0189 | -0.0719 | -0.48   | -0.08   | -0.12   | -0.34   |
>
> In principle, finetuning could help mitigate this tension—for example, by supervising on high-quality responsive UI code or using our metrics as optimization or reward signals for open-source models.
> Finetuning is indeed promising, though beyond the main focus of our benchmark study. We see it as a natural avenue for future work.

---

### Official Review · Reviewer_5rJz · 2025-10-23

**Soundness:** 3
**Presentation:** 4
**Contribution:** 4
**Rating:** 6
**Confidence:** 3

**Summary:**

The paper proposes a benchmark to evaluate UI code generation reflecting enterprise workflows given Figma artifacts (images and structured metadata) rather than screenshots alone. Across multiple MLLMs, the authors show that models can leverage both image and metadata to get high visual fidelity but still suffer from code responsiveness and maintainability.

**Strengths:**

1. Strong problem formulation. Moving beyond screenshots-only to include metadata that is a part of enterprise UI design process makes the study highly relevant. The benchmark evaluates on axes like code responsiveness and maintainability in addition to visual fidelity.
2. Dataset creation shows good coverage with thorough automatic filtering and human-in-the-loop checks.
3. The metrics introduced for each axis are clear and reproducible.
4. Experiments and ablations are informative. Results show that image + metadata improves visual fidelity. Ablations clearly indicate geometry/hierarchy drive responsiveness and metadata encourages brittle code affecting maintainability. Tables 2 and 3 sufficiently show these patterns across multiple MLLMs.

**Weaknesses:**

1. RUR and APR metrics don’t guarantee cross-device behavior as code-level proxies can miss real rendered issues like overflow/overlap at narrow viewports, failed wrapping/reflow, aspect-ratio distortions among others. Similarly, STR and AVU might only be superficial and ignore semantic correctness and accessibility (e.g.,  heading hierarchy), overlook architectural signals like specificity, duplication and component reuse.
2. The MLLMs considered are predominantly closed source. Adding more open-source models can improve reproducibility. Robustness of findings to prompt variations and per-model instruction following quirks (if any; especially with open-source models) and how they are normalized is unclear.
3. Correlation of the proposed code quality metrics with developer insights and judgements is missing.

**Questions:**

1. Can code maintainability shortcomings be addressed with explicit and detailed instructions? How about few-shot samples? Generally, how robust are the models to system prompt perturbations?
2. Are there any human studies that tie proposed metrics to developer experience?

---

> ### Author Response · Authors · 2025-11-22
> **Response to Reviewer 5rJz [1/4]**
>
> ## On the robustness to explicit prompt perturbations
>
> **Question 1**:
> > *Can code maintainability shortcomings be addressed with explicit and detailed instructions? How about few-shot samples? Generally, how robust are the models to system prompt perturbations?*
>
> **Answer**
>
> Thank you for the insightful question.
>
> As shown in Appendix Fig. 18 and Fig. 19, our original system prompt already included an explicit instruction encouraging maintainable and responsive code generation:
> ```text
> The output must preserve the visual fidelity of the design while improving responsiveness and maintainability.
> ```
>
> To further address this concern, we expanded the prompt with a set of more detailed, rule-level guidelines:
> ```text
> - RESPONSIVE LAYOUTS: Use Tailwind responsive prefixes (sm:, md:, lg:, xl:) so layouts adapt to different screen sizes. Prefer flexible layouts over fixed-width designs.
> - AVOID DIRECT FIGMA COORDINATES: Do not rigidly copy x/y/width/height from Figma. Use natural flex/grid flow unless the design clearly requires fixed positioning.
> - USE FLEX/GRID FIRST: Favor flex or grid for structure, with gap and spacing utilities. Absolute positioning is allowed only when the design explicitly depends on it.
> - REDUCE UNNECESSARY WRAPPERS: Simplify the DOM when possible. Keep structure clean but do not remove elements that are semantically or visually important.
> - REUSE UI PATTERNS: When similar UI fragments repeat, prefer component extraction. Avoid excessive duplication of markup.
> - TAILWIND SCALES PREFERRED: Use Tailwind’s spacing, color, font, and radius scales when reasonable. Arbitrary values are allowed when needed for closer visual match.
> - CLEAN CLASS LISTS: Keep Tailwind classes organized and avoid redundant utilities, but do not over-optimize at the cost of clarity.
> - BALANCED FIDELITY: Match the design visually, but when pixel precision conflicts with readability or maintainability, choose the simpler and clearer implementation.
> ```
> A few-shot setting was not feasible due to limited context length—the multimodal input is long and full code examples easily exceed the available window.
> We therefore conducted additional experiments under two configurations:
> (1) metadata-only, and
> (2) metadata + image.
> The results are shown below:
>
>
> | Method                            | VES (↑) | MAE (↓) | RUR (↑) | APR (↓) | STR (↑) | AVU (↓) |
> |-----------------------------------|---------|---------|---------|---------|---------|---------|
> | **Direct Prompting (metadata only)**     | 0.6801  | 0.2101  | 4.97    | 5.01    | 21.46   | 6.09    |
> | + Enhanced Prompting                          | 0.6589  | 0.1562  | 2.05    | 3.99    | 22.15   | 3.29    |
> | Δ                                 | -0.0212 | -0.0539 | -2.92   | -1.02   | +0.69   | -2.80  |
> |                                   |         |         |         |         |         |         |
> | **Direct Prompting (metadata + image)**                  | 0.6923  | 0.2228  | 4.81    | 2.83    | 32.03   | 2.06    |
> | + Enhanced Prompting                          | 0.7112  | 0.1509  | 4.33    | 2.75    | 31.91   | 1.72    |
> | Δ                                 | +0.0189 | -0.0719 | -0.48   | -0.08   | -0.12   | -0.34   |
>
> Overall, the models appear generally robust to prompt perturbations. The enhanced prompt does not lead to consistent and significant improvements across metrics. Apart from some localized effects, the overall impact remains modest.

---

> ### Author Response · Authors · 2025-11-22
> **Response to Reviewer 5rJz [2/4]**
>
> ## On Metrics and Human Evaluation
>
> **Question 2**
> > *Are there any human studies that tie proposed metrics to developer experience?*
>
> **Weakness 2**
> > *Correlation of the proposed code quality metrics with developer insights and judgements is missing*
>
> **Answer**
>
> Thank you for the thoughtful question.
>
> To avoid redundancy, we provide a unified comment on our metrics and human evaluation in the global comment titled **“More Studies about Metrics and Human Evaluation”**. Please kindly refer to that section for the detailed analysis.

---

> > ### Author Response · Authors · 2025-11-22
> > **Response to Reviewer 5rJz [3/4]**
> >
> > ## On More Metrics for Responsiveness and Maintainability
> >
> > **Weakness 1**
> > > *RUR and APR metrics don’t guarantee cross-device behavior as code-level proxies can miss real rendered issues like overflow/overlap at narrow viewports, failed wrapping/reflow, aspect-ratio distortions among others. Similarly, STR and AVU might only be superficial and ignore semantic correctness and accessibility (e.g., heading hierarchy), overlook architectural signals like specificity, duplication and component reuse.*
> >
> > **Answer**
> >
> > Thank you for the professional comment.
> >
> > We agree that no single metric can fully capture cross-device behavior or the entire spectrum of semantic and architectural correctness. Our aim is not to offer a perfect measure, but to provide a set of objective and reproducible signals. To the best of our knowledge, this is the first attempt in the design-to-code setting to introduce maintainability- and responsiveness-oriented metrics, and we believe these efforts can help inspire more comprehensive evaluation frameworks in future work.
> >
> > In fact, beyond the four metrics in the main text (RUR, APR, AVU, and STR), we also introduce several complementary indicators in the appendix—Flex/Grid Utilization (FU), Breakpoint Coverage (BC), Inline Style Ratio (ISR), and Custom Class Reuse (CCR). Their definitions and results are provided in the appendix due to space constraints.

---

> ### Author Response · Authors · 2025-11-22
> **Response to Reviewer 5rJz [4/4]**
>
> ## On Open-Source Model Inclusion
> **Weakness 3**
> > *The MLLMs considered are predominantly closed source. Adding more open-source models can improve reproducibility. Robustness of findings to prompt variations and per-model instruction following quirks (if any; especially with open-source models) and how they are normalized is unclear.*
>
> **Answer**
>
> Thank you for the suggestion.
>
> We agree that most of the models in our evaluation are closed-source: among the ten MLLMs we tested, three are open-source (Llama 4 Maverick, Llama 4 Scout, and Qwen 2.5 VL). These were selected because they represent the strongest open-source MLLMs available at the time of our experiments. We will consider including additional open-source models in future updates. Due to space limitations, the full prompt and the unified inference protocol are provided in the appendix, and following your suggestion, we have added explicit references to them in the main text.

---

### Official Review · Reviewer_EXnw · 2025-10-26

**Soundness:** 3
**Presentation:** 3
**Contribution:** 3
**Rating:** 6
**Confidence:** 4

**Summary:**

The paper introduces FIGMA2CODE, a realistic multimodal design-to-code task and benchmark built from public Figma community files. Each example pairs a rendered screenshot with cleaned JSON metadata (hierarchy, geometry, styles) and linked assets (icons/images). From ~2.1k files and ~30k pages, the authors curate a 213-sample benchmark for MLLM evaluation.

They evaluate 10 state-of-the-art MLLMs with reference-free metrics for visual fidelity (VES via DINOv2; MAE), responsiveness (RUR, APR), and maintainability (STR, AVU). Results show proprietary models lead in visual similarity but lag on responsiveness/maintainability, whereas some open models produce cleaner, more responsive code. An agentic baseline (F2CAGENT) improves fidelity when using both image and metadata, and an ablation over five metadata components quantifies their differing effects (styles/assets critical for fidelity; geometry/hierarchy for responsiveness).

**Strengths:**

- The paper reframes design-to-code as multimodal (image + metadata + assets) rather than image-only; clearly motivated by real Figma workflows

- The paper shows significant effort in data curation and data quality assurance, including heuristic filters, CLIP-based dedup (0.95), expert screening, metadata pruning, asset unification, resulting in self-contained samples.

- Evaluation metrics go beyond fidelity to responsiveness and maintainability; definitions are explicit and implementable.

- Clear, actionable finding: e.g., metadata boosts fidelity but encourages rigid/absolute layouts; multimodality partly mitigates.

**Weaknesses:**

- While the paper presents a valuable MLLM benchmarking resource, the significance of the Figma2Code task is unclear given the existing commercialized figma-to-code solutions (e.g., Figma’s native export/Dev Mode plugins and third-party tools like Locofy/TeleportHQ). Further discussion is needed to justify how MLLMs may benefit front-end designers beyond existing solutions.

- VES + MAE lack user-centric validation; perceptual ranking or human rater studies would strengthen claims. Both responsiveness and code quality metrics are rather hard-coded and can be “optimized” without genuinely high-quality webpages or maintainable code (e.g., inserting semantic tags mechanically). Calibrate with expert code ratings or lints.

- The core benchmark (213) is relatively small for training-free conclusions and may not capture long-tail UI patterns despite stratified sampling. Recommend expanding the dataset, construct multiple subsets with varying difficulty levels, or reporting variance across multiple random subsets.

- While protocols are unified, randomness of inference-time sampling and absence of statistical significance leave uncertainty about small deltas.

**Questions:**

- Can the authors perform human ratings and preferences as auxiliary metrics? How well does each of the metrics (VES, MAE, RUR, APR) align with empirical human preferences?

- Can the authors provide any "smarter" or more comprehensive evaluations of code quality in addition to the two rule-based maintainability metrics? Any correlation with linter-based or expert maintainability ratings?

- Can the authors report some ablation studies on the proposed agentic workflow?

---

> ### Author Response · Authors · 2025-11-22
> **Response to Reviewer EXnw [1/6]**
>
> ## On Metrics and Human Evaluation
>
> **Question 1**
> > *Can the authors perform human ratings and preferences as auxiliary metrics? How well does each of the metrics (VES, MAE, RUR, APR) align with empirical human preferences?*
>
> **Weakness 2**
> > *VES + MAE lack user-centric validation; perceptual ranking or human rater studies would strengthen claims. Both responsiveness and code quality metrics are rather hard-coded and can be “optimized” without genuinely high-quality webpages or maintainable code (e.g., inserting semantic tags mechanically). Calibrate with expert code ratings or lints*
>
> **Answer**
>
> Thank you for the thoughtful question.
>
> To avoid redundancy, we provide a unified comment on our metrics and human evaluation in the global comment titled **“More Studies about Metrics and Human Evaluation”**. Please kindly refer to that section for the detailed analysis.

---

> ### Author Response · Authors · 2025-11-22
> **Response to Reviewer EXnw [2/6]**
>
> ## On More “Smarter” or Comprehensive Code-Quality Evaluations
>
> **Question 2**
>  > *Can the authors provide any "smarter" or more comprehensive evaluations of code quality in addition to the two rule-based maintainability metrics? Any correlation with linter-based or expert maintainability ratings?*
>
> **Answer**
>
> Thank you for the insightful question.
>
> At the early stage of the project, we explored several alternative metrics, including MLLM-as-a-judge evaluations and weighted composite scores. MLLM-as-a-judge produced highly inconsistent judgments across models, making the results difficult to reproduce and unsuitable for benchmarking. Weighted composite scores may appear “smarter,” but they require manually assigning weights across heterogeneous dimensions, which introduces opacity and makes the aggregated score harder to interpret.
>
> We therefore opted for objective, transparent metrics that directly capture maintainability properties of UI code. We would like to note that our evaluation is more comprehensive than the two metrics highlighted in the main text: in addition to Relative Unit Ratio (RUR) and Absolute Positioning Ratio (APR), we computed two further indicators—Inline Style Ratio (ISR) and Custom Class Reuse (CCR). Their definitions and results are provided in Appendix E.3 and Appendix H due to space limitations in the main paper.
>
> Regarding linter-based evaluation, mainstream linters (e.g., ESLint, Stylelint) are highly configuration-dependent and focus primarily on syntax and general coding conventions rather than UI-specific maintainability patterns. Our metrics are implementation-agnostic statistical measures that could, in principle, be expressed as custom linter rules—a direction we view as promising for future work.

---

> ### Author Response · Authors · 2025-11-22
> **Response to Reviewer EXnw [3/6]**
>
> ## On Ablation of the Agentic Workflow
>
> **Question 3**
>  > *Can the authors report some ablation studies on the proposed agentic workflow?*
>
> **Answer**
>
> Thank you for the insightful question.
>
> Our agentic workflow is a primary reference baseline built on a classic ReAct-style reactive architecture. It is intentionally simple, serving as an initial exploration rather than a fully developed agent system. Because its steps and tool calls unfold in a tightly coupled manner, dissecting it into clean ablation components is not straightforward.
>
> We will acknowledge this limitation in the paper. Exploring richer and more sophisticated agentic workflows is a promising direction for future work.

---

> ### Author Response · Authors · 2025-11-22
> **Response to Reviewer EXnw [4/6]**
>
> ## On the Significance of the Figma2Code Task
>
> **Weakness 1**
> > *While the paper presents a valuable MLLM benchmarking resource, the significance of the Figma2Code task is unclear given the existing commercialized Figma-to-Code solutions (e.g., Figma’s native export/Dev Mode plugins and third-party tools like Locofy/TeleportHQ). Further discussion is needed to justify how MLLMs may benefit front-end designers beyond existing solutions.*
>
> **Answer**
>
> Thank you for the professional comment.
>
> As described in the introduction and illustrated in Fig. 1, **the significance of our work is to formally propose the Figma2Code task and its corresponding dataset, advancing design-to-code research closer to real software development workflows**. In practice, Figma designs are inherently multimodal and include rich structured metadata, whereas image-only code generation is often closer to screenshot-to-code replication of an existing page rather than supporting actual development needs. **Our goal is not to build a standalone Figma-to-code tool, but to advance design-to-code research toward practical and fully automated UI code generation in the wild**.
>
> Our work differs from existing commercialized Figma-to-code solutions in its research focus. Traditional converters—whether Figma’s native export/Dev Mode plugins rely on predefined rules or template-based mappings. As a result, the generated code often exhibits rigid layouts and is difficult for developers to maintain. With recent advances in MLLMs, generating code directly from full Figma designs has become feasible, providing a more flexible approach that can overcome the limitations of template-based conversion. We have included the Figma-to-code plugin as a baseline; Table 3 shows that such rule-based converters perform poorly in both responsiveness and maintainability.

---

> > ### Author Response · Authors · 2025-11-22
> > **Response to Reviewer EXnw [5/6]**
> >
> > ## On the Size of the Benchmark  Dataset
> > **Weakness 3**
> > > *The core benchmark (213) is relatively small for training-free conclusions and may not capture long-tail UI patterns despite stratified sampling. Recommend expanding the dataset, construct multiple subsets with varying difficulty levels, or reporting variance across multiple random subsets.*
> >
> > **Answer**
> >
> > Thank you for the suggestion.
> >
> > As shown in Table 1, the overall scale of our dataset is comparable to existing UI generation benchmarks. Unlike prior work, each of our samples includes not only the design image but also detailed structured metadata, which makes the input itself very long—Fig. 4 shows that a single metadata file averages roughly 300k tokens. In our experiments, running inference on models such as GPT-5 over our benchmark dataset often requires more than six hours in a single-thread setting, making very large evaluation splits impractical.
> >
> > For this reason, we keep the standard test set to 213 stratified and balanced samples, while the remaining 2,842 samples are preserved as a reserve for future extensions or alternative evaluation subsets.

---

> > > ### Author Response · Authors · 2025-11-22
> > > **Response to Reviewer EXnw [6/6]**
> > >
> > > ## On Inference Stability and Reproducibility
> > >
> > > **Weakness 3**
> > > > *While protocols are unified, randomness of inference-time sampling and absence of statistical significance leave uncertainty about small deltas.*
> > >
> > > **Answer**
> > >
> > > Modern LLM inference inevitably contains a degree of non-determinism due to factors such as non-deterministic GPU kernels, floating-point reduction order, and batch-composition effects [1] [2]. As described in Sec *Reproducibility Statement*, we have already minimized such randomness to the greatest extent possible by (1) setting temperature = 0, (2) fixing all seeds, and (3) using a unified inference protocol to ensure identical decoding conditions across runs.
> > >
> > > To further address your concern, we conducted a consistency check using a balanced subset of 40 samples from our benchmark. We evaluated all models twice under **identical** settings and compared results across the two runs. The table below reports the VES and MAE metrics for both runs, along with the absolute difference (Δ). As shown, the variations are very small across all models, and none of them affects model ranking or our conclusions.
> > >
> > > | Model                                   | VES (Run1) | VES (Run2) | Δ VES     | MAE (Run1) | MAE (Run2) | Δ MAE     |
> > > |-----------------------------------------|------------|------------|-----------|------------|------------|-----------|
> > > | ERNIE 4.5 424B VL (Baidu, 2025)         | 0.719276   | 0.725897   | +0.006621 | 0.128644   | 0.136741   | +0.008097 |
> > > | Gemini 2.5 Pro (Google, 2025)           | 0.909367   | 0.899344   | -0.010023 | 0.092283   | 0.103884   | +0.011601 |
> > > | GPT-5 (OpenAI, 2025)                    | 0.851682   | 0.847033   | -0.004649 | 0.124430   | 0.129618   | +0.005188 |
> > > | Grok4 (xAI, 2025)                        | 0.863029   | 0.865202   | +0.002173 | 0.106843   | 0.087177   | -0.019666 |
> > > | Llama 4 Maverick (MetaAI, 2025)         | 0.679704   | 0.653216   | -0.026488 | 0.165844   | 0.181345   | +0.015501 |
> > > | Llama 4 Scout (MetaAI, 2025)            | 0.604854   | 0.582767   | -0.022087 | 0.209980   | 0.208930   | -0.001050 |
> > > | Nova Pro v1 (Amazon, 2025)              | 0.585265   | 0.617855   | +0.032590 | 0.211809   | 0.228909   | +0.017100 |
> > > | Qwen 2.5 VL (Alibaba, 2025)             | 0.711496   | 0.692160   | -0.019336 | 0.146559   | 0.162746   | +0.016187 |
> > >
> > > Across all models, the absolute differences in VES and MAE are small (typically within 0.00–0.03) and do not change any relative ranking or qualitative findings. This demonstrates that the benchmark conclusions are stable and not driven by inference randomness.
> > >
> > > ---
> > >
> > > References:
> > >
> > > [1] Non-Determinism in TensorFlow ResNets
> > >
> > > [2] mlf-core: a framework for deterministic machine learning

---

> > > > ### Comment · Reviewer_EXnw · 2025-11-24
> > > > **Response to the Authors**
> > > >
> > > > Thanks for the detailed rebuttal. The additional rationale for MAE and the reproducibility check are helpful. However, there are several concerns that remain unaddressed/tangentially addressed.
> > > >
> > > > 1. **No human evaluation on this dataset.**
> > > > While MRWeb’s study supports MAE in general, it does not validate your full metric suite (MAE + DINOv2 VES + RUR/APR/ISR/CCR/STR/AVU) on FIGMA2CODE. Please add a small, reproducible human study on a stratified subset and provide
> > > >
> > > >     - Fidelity: pairwise preference of rendered outputs vs. ground truth; report SROCC/CC between human rankings and MAE/DINOv2.
> > > >     - Code quality & responsiveness: expert ratings (front-end practitioners) on maintainability, semantics/accessibility, and responsiveness; report agreement (e.g., Krippendorff’s α) and correlation with RUR/APR/ISR/CCR/STR/AVU.
> > > >
> > > >
> > > > 2. **Code quality metrics lack empirical validations.** I accept the arguments against MLLM-as-a-judge and opaque composite scores. But the current maintainability metrics remain heuristic and potentially gameable. Without *any* external anchor to validate these heuristic-based metrics, the usability of these code quality metrics remain questionable. Can the authors please provide the followings:
> > > >     - Correlate your metrics with (i) expert ratings from (1b) and/or (ii) a fixed, published linter ruleset.
> > > >     - Include a short failure-mode taxonomy
> > > >
> > > >
> > > > 3. The 213-item test set is acceptable given token length and cost, but there’s no notion of difficulty or robustness. Please release different subsets with varying levels of difficulties (e.g., Figma2Code vs Figma2Code-HARD), or at the very least report statistics on the task difficulties with visualized examples.

---

> ### Author Response · Authors · 2025-11-25
>
> Dear Reviewer EXnw,
>
> Thank you for the constructive response.
>
> We have conducted additional experiments on most of the aspects you suggested.
>
> ## On SROCC/CC between human rankings and MAE/DINOv2
> Following your suggestion, we conducted the same evaluation protocol as MRWeb. The results confirm the strong correlation between DINOv2-based similarity and human rankings, while MAE shows only moderate correlation. We performed an in-depth analysis to understand this discrepancy. The full results are presented in the global comment **“Additional Studies on the Alignment Between Visual Fidelity Metrics and Human Ratings in Our Benchmark.”** Please kindly refer to that comment for details.
>
> ---
>
> ## On SROCC/CC between human rankings and code quality metrics
> Following your suggestion, we conducted additional experiments to study the alignment between human rankings and Code quality metrics. The results are presented in the global comment **“Additional Studies about Alignment Between Code Quality Metrics and Human Ratings in Our Benchmark.”** Please kindly refer to that comment for details.
>
> ---
>
> ## On CSS linters
>
> We used stylelint v16.26.0 (with stylelint-config-standard and stylelint-config-tailwindcss) to inspect the code generated by Grok4. Grok4 is selected because, as shown in Table 2, it is a representative model that achieves high visual fidelity but very poor code-quality metrics. Its outputs frequently contain rigid, metadata-literal layouts, for example:
>
> ```html
> <div class="absolute left-[12px] top-[2.886px] w-[185px] h-[28px] overflow-hidden">
>   <div class="absolute left-[0px] top-[3.114px] w-[158px] h-[22px] text-white text-[17px] font-normal">It’s morning in Tokyo</div>
>   <div class="absolute left-[163px] top-[0.614px] w-[17px] h-[17px] text-white text-[17px] font-normal">😎</div>
> </div>
> <div class="absolute left-[207px] top-[15px] w-[28px] h-[13px] text-right text-[rgb(142,142,147)] text-[11px] italic">11:43</div>
> ```
>
> The aggregated stylelint results are:
> | Rule | Count | Files |
> |---|---:|---:|
> | no-invalid-position-declaration | 2806 | 93 |
> | color-function-notation | 308 | 35 |
> | alpha-value-notation | 244 | 33 |
> | color-function-alias-notation | 244 | 33 |
> | font-family-name-quotes | 164 | 69 |
> | font-family-no-missing-generic-family-keyword | 164 | 6 |
> | length-zero-no-unit | 73 | 9 |
> | declaration-property-value-no-unknown | 70 | 4 |
> | at-rule-empty-line-before | 49 | 22 |
> | color-hex-length | 32 | 4 |
> | rule-empty-line-before | 28 | 26 |
> | property-no-vendor-prefix | 8 | 5 |
> | declaration-block-single-line-max-declarations | 3 | 1 |
> | CssSyntaxError | 2 | 2 |
> | comment-empty-line-before | 2 | 2 |
> | function-url-quotes | 1 | 1 |
> | property-no-unknown | 1 | 1 |
>
> Although many warnings appear, they mostly relate to formatting or generic CSS conventions (e.g., color-notation, font-family quoting). None of them captures the issues in our benchmark (e.g., overuse of absolute positioning, rigid pixel values, missing semantic structure, lack of layout abstraction).
>
> As a result, published stylelint rulesets focus on general CSS conventions, and therefore provide only partial coverage of the issues that arise specifically in Figma-to-code generation—such as the MLLM tendency to directly translate Figma coordinates into rigid absolute layouts, overuse pixel-fixed values, and produce structures with poor maintainability.

---

> ### Author Response · Authors · 2025-11-25
>
> ## On the notion of task difficulty
> Thank you for the suggestion. We agree that difficulty-aware subsets are valuable (e.g., design2code-HARD).
> But we would like to note that our dataset already includes an explicit **complexity attribute** for every sample, and its overall distribution is shown in the inner ring of Fig. 3. Each instance also provides fine-grained structural statistics (e.g., node count, depth, component types), which are included in the released metadata.
>
> Following your advice, we have expanded this part in the revision. We now provide **visualized examples for each difficulty level** together with their corresponding JSON-style structural statistics in Appendix Fig. 26. These examples make the differences between difficulty levels more concrete. For instance:
>
> - **Low**:
> ```json
> {
>   "file_key": "08wDXrTJgBoPVvQzlkKBtV",
>   "node_id": "1:124",
>   "page_url": "https://www.figma.com/design/08wDXrTJgBoPVvQzlkKBtV/?node-id=1:124",
>   "annotation": {
>     "platform": "mobile",
>     "complexity": "low",
>     "quality_rating": "2",
>     "theme": "light",
>     "language": "en",
>     "content": "Support, Guidance & Onboarding",
>     "description": "A mobile onboarding screen inviting users to explore the app, featuring options to continue with Google, Apple, or Email, along with a brief description of the app's purpose.",
>     "content_original": "Onboarding"
>   },
>   "statistics": {
>     "type_counts": {
>       "FRAME": 2,
>       "RECTANGLE": 4,
>       "INSTANCE": 6,
>       "GROUP": 4,
>       "TEXT": 6
>     },
>     "node_counts": 22,
>     "max_depth": 4,
>     "downloaded_counts": {
>       "components_json": 4,
>       "components_img": 4,
>       "component_sets_json": 0,
>       "styles_json": 0,
>       "image_refs": 0,
>       "svg_assets": 4,
>       "render_nodes": 6
>     },
>     "total_resources": 4,
>     "total_downloaded": 18
>   },
> }
> ```
> - **Hard**:
>
> ```json
> {
> "file_key": "m70bwPhFdjw5ZD1n5gOrIc",
>   "node_id": "1:182",
>   "page_url": "https://www.figma.com/design/m70bwPhFdjw5ZD1n5gOrIc/?node-id=1:182",
>   "annotation": {
>     "platform": "desktop",
>     "complexity": "hard",
>     "quality_rating": "3",
>     "theme": "light",
>     "language": "en",
>     "content": "Data & Information Presentation",
>     "description": "A desktop dashboard UI for a school management system, featuring attendance and financial summaries, user registration details, and a calendar for upcoming events.",
>     "content_original": "Dashboard"
>   },
>   "statistics": {
>     "type_counts": {
>       "FRAME": 55,
>       "RECTANGLE": 93,
>       "GROUP": 34,
>       "TEXT": 132,
>       "INSTANCE": 7
>     },
>     "node_counts": 321,
>     "max_depth": 8,
>     "downloaded_counts": {
>       "components_json": 2,
>       "components_img": 2,
>       "component_sets_json": 0,
>       "styles_json": 0,
>       "image_refs": 2,
>       "svg_assets": 47,
>       "render_nodes": 77
>     },
>     "total_resources": 51,
>     "total_downloaded": 130
>   },
> }
>
> ```
> As shown in these examples, low-complexity designs typically contain shallow hierarchies and few layout branches, whereas hard cases exhibit deep nesting, heterogeneous component compositions, and substantially more geometric and constraint variability.
>
> We hope the visual examples and the accompanying structural statistics in the appendix help clarify our definition of task difficulty and its relevance to robustness evaluation.

---

### Official Review · Reviewer_EBQS · 2025-11-01

**Soundness:** 2
**Presentation:** 3
**Contribution:** 2
**Rating:** 6
**Confidence:** 2

**Summary:**

Existing methods generate code solely from images, overlooking that design mockups are typically delivered as Figma files—a widely used front-end design tool. This paper introduces a new task, FIGMA2CODE, pushing design-to-code research beyond image-only approaches toward a multimodal, industry-relevant setting. It is the first to establish a systematic evaluation framework that assesses not only visual fidelity but also code quality in a multimodal context.

**Strengths:**

1. The paper is clearly written.

2. The paper presents novelty and comprehensive experiments.

**Weaknesses:**

1. The Method section requires a more concrete and transparent description, especially within the Metadata Refinement subsection. It would be beneficial to include specific figures, examples, or case studies that illustrate the data-cleaning pipeline in practice. For example, showing how noisy examples are identified, filtered, or corrected would help readers better understand the robustness and reliability of the dataset construction process.

2. The definition of “difficulty” remains unclear. Lines 254–255 suggest that complexity is determined by the interface complexity, but it would strengthen the paper to provide concrete examples or ablation studies that demonstrate how tasks differ across varying difficulty levels. Furthermore, the relationship between task difficulty and model difficulty (as opposed to human preference or perception) should be clarified. It would be helpful to quantify how “difficult” tasks challenge the model’s reasoning or generation capabilities, ideally supported by empirical evidence.

3. A core concern lies in the conceptual positioning of the benchmark. Benchmarks are typically designed to evaluate the ability boundaries of large models under consistent and unconditional setups. However, in this work, the proposed benchmark incorporates conditional inputs intended to assist the model in generating UI code. This design choice seems to blend evaluation with task guidance, potentially compromising the benchmark’s intended diagnostic purpose. The paper would benefit from a clearer justification for this approach—explaining whether the goal is to measure raw capability or conditional adaptability—and from additional discussion on how this aligns with broader benchmarking principles in the LLM community.

**Questions:**

1. The authors state (Line 240) that the benchmark dataset was constructed via stratified sampling followed by expert selection, with the goal of maintaining balance across key dimensions such as platform, complexity, and content category. However, it remains unclear how this process quantitatively guarantees diversity and coverage. Specifically, how were the strata defined and weighted, and what measures (e.g., entropy, coverage ratio, distribution analysis) were used to verify that the final dataset indeed represents a broad and balanced distribution? Including a concrete analysis or visualization (e.g., distribution plots or summary statistics) would substantially strengthen this claim.

2. The current evaluation setup seems to rely on a single run per model, which may underestimate model performance due to stochastic generation variability. Have the authors considered adopting pass@k metrics (e.g., pass@1, pass@5, pass@10) as done in standard code generation or reasoning benchmarks? Such an evaluation would provide a more robust and statistically grounded assessment of model capabilities, particularly for generative tasks.

---

> ### Author Response · Authors · 2025-11-22
> **Response to Reviewer EBQS [1/5]**
>
> ## On Data Sampling  Details in Dataset Construction
>
> **Question 1**
> > *The authors state (Line 240) that the benchmark dataset was constructed via stratified sampling followed by expert selection, with the goal of maintaining balance across key dimensions such as platform, complexity, and content category. However, it remains unclear how this process quantitatively guarantees diversity and coverage. Specifically, how were the strata defined and weighted, and what measures (e.g., entropy, coverage ratio, distribution analysis) were used to verify that the final dataset indeed represents a broad and balanced distribution? Including a concrete analysis or visualization (e.g., distribution plots or summary statistics) would substantially strengthen this claim.*
>
>
> **Answer**
>
> Thank you for the helpful suggestion.
>
> The full sampling and curation pipeline is described in Appendix C.5, and in the revision we have added an explicit reference in the main text for clarity.
>
> Briefly, we define the strata along four orthogonal dimensions—platform (mobile/desktop), interface complexity (four levels), quality rating (2–3), and content category (12 classes). We compute the distribution of all 3,055 designs across these strata and sample proportionally according to the target test-set size (≈200), ensuring that the drawn subset mirrors the global distribution. To further guarantee quality and representativeness, we over-sample each stratum (about 2× the required amount) and submit these candidates to design experts, who remove near-duplicate or low-quality samples and retain roughly half. This two-stage process preserves distributional balance while ensuring that each test example is clear and typical.
>
> We have provided quantitative evidence of coverage and balance in the main text (Fig. 3, Fig. 4, and Table 1) and further analyses in Appendix D. These analyses show that the dataset exhibits well-distributed coverage across all key dimensions. We hope this clarifies the completeness and rigor of the sampling procedure.

---

> ### Author Response · Authors · 2025-11-22
> **Response to Reviewer EBQS [2/5]**
>
> ## On the Applicability of pass@k for UI Code Generation
> **Question 2**
> > *The current evaluation setup seems to rely on a single run per model, which may underestimate model performance due to stochastic generation variability. Have the authors considered adopting pass@k metrics (e.g., pass@1, pass@5, pass@10) as done in standard code generation or reasoning benchmarks? Such an evaluation would provide a more robust and statistically grounded assessment of model capabilities, particularly for generative tasks.*
>
> **Answer**
>
> Thank you for the suggestion.
>
> We agree that pass@k is a standard and valuable metric in code-generation benchmarks where correctness is defined by executable test cases. However, UI code generation in our setting is fundamentally different: the generated HTML/Tailwind code is almost always syntactically valid, and quality is evaluated through continuous visual and quality metrics rather than a binary notion of “passing” or “failing” test cases. Consequently, pass@k does not meaningfully reflect the quality dimensions we aim to measure (e.g., visual fidelity, responsiveness, maintainability), and is therefore not well aligned with this task.
>
> Regarding your concern that a single run might underestimate performance due to stochasticity, we conducted two additional evaluation runs on a balanced subset of 40 samples under identical settings. The results across the two runs are highly consistent. For example, the absolute differences in VES and MAE for all models are small (see Table X), and no model’s ranking or qualitative trend changes. This confirms that residual stochasticity has a negligible influence on the reported benchmark conclusions.
> | Model                                   | VES (Run1) | VES (Run2) | Δ VES     | MAE (Run1) | MAE (Run2) | Δ MAE     |
> |-----------------------------------------|------------|------------|-----------|------------|------------|-----------|
> | ERNIE 4.5 424B VL (Baidu, 2025)         | 0.719276   | 0.725897   | +0.006621 | 0.128644   | 0.136741   | +0.008097 |
> | Gemini 2.5 Pro (Google, 2025)           | 0.909367   | 0.899344   | -0.010023 | 0.092283   | 0.103884   | +0.011601 |
> | GPT-5 (OpenAI, 2025)                    | 0.851682   | 0.847033   | -0.004649 | 0.124430   | 0.129618   | +0.005188 |
> | Grok4 (xAI, 2025)                        | 0.863029   | 0.865202   | +0.002173 | 0.106843   | 0.087177   | -0.019666 |
> | Llama 4 Maverick (MetaAI, 2025)         | 0.679704   | 0.653216   | -0.026488 | 0.165844   | 0.181345   | +0.015501 |
> | Llama 4 Scout (MetaAI, 2025)            | 0.604854   | 0.582767   | -0.022087 | 0.209980   | 0.208930   | -0.001050 |
> | Nova Pro v1 (Amazon, 2025)              | 0.585265   | 0.617855   | +0.032590 | 0.211809   | 0.228909   | +0.017100 |
> | Qwen 2.5 VL (Alibaba, 2025)             | 0.711496   | 0.692160   | -0.019336 | 0.146559   | 0.162746   | +0.016187 |

---

> ### Author Response · Authors · 2025-11-22
> **Response to Reviewer EBQS [3/5]**
>
> ## More Details about  Data Processing
>
> **Weakness 1**
> > *The Method section requires a more concrete and transparent description, especially within the Metadata Refinement subsection. It would be beneficial to include specific figures, examples, or case studies that illustrate the data-cleaning pipeline in practice. For example, showing how noisy examples are identified, filtered, or corrected would help readers better understand the robustness and reliability of the dataset construction process.*
>
> **Answer**
>
> Thank you for the suggestion.
>
> Due to space limitations, the detailed data-processing pipeline was originally placed in Appendix C. Following your feedback, we have now added a new figure (Fig. 25) in the appendix to visually illustrate the metadata-refinement process. We hope these additions make the refinement procedure more transparent and easier to understand.

---

> ### Author Response · Authors · 2025-11-22
> **Response to Reviewer EBQS [4/5]**
>
> ## On the Definition of Task Difficulty
> **Weakness 2**
> > *The definition of “difficulty” remains unclear. Lines 254–255 suggest that complexity is determined by the interface complexity, but it would strengthen the paper to provide concrete examples or ablation studies that demonstrate how tasks differ across varying difficulty levels. Furthermore, the relationship between task difficulty and model difficulty (as opposed to human preference or perception) should be clarified. It would be helpful to quantify how “difficult” tasks challenge the model’s reasoning or generation capabilities, ideally supported by empirical evidence.*
>
> **Answer**
>
> Thank you for the helpful comment.
>
> We have clarified the definition of interface complexity in Appendix C.2. Following your suggestion, we have refined the description and added an explicit reference in the main text. In addition, we now provide **visualized examples** across different difficulty levels in Appendix Figure 25. The accompanying statistics clearly show that task “difficulty” primarily manifests in properties such as node count, hierarchical depth, and structural branching. For example:
> ```json
> {
>   "file_key": "2uAU8qEUeDh5b1K8XPByby",
>   "node_id": "101:1712",
>   "page_url": "https://www.figma.com/design/2uAU8qEUeDh5b1K8XPByby/?node-id=101:1712",
>   "annotation": {
>     "platform": "desktop",
>     "complexity": "high",
>     "quality_rating": "2",
>     "theme": "light",
>     "language": "en",
>     "content": "Forms & Interaction Flows",
>     "description": "A desktop contact form UI featuring input fields for name, email, subject, and messages, along with a submit button. Below, there is a subscription section for recipes and a list of featured recipes with images and descriptions.",
>     "content_original": "Form/Input"
>   },
>   "statistics": {
>     "type_counts": {
>       "FRAME": 1,
>       "GROUP": 36,
>       "TEXT": 42,
>       "RECTANGLE": 42,
>       "VECTOR": 4
>     },
>     "node_counts": 125,
>     "max_depth": 4,
>     "downloaded_counts": {
>       "components_json": 0,
>       "components_img": 0,
>       "component_sets_json": 0,
>       "styles_json": 2,
>       "image_refs": 8,
>       "svg_assets": 12,
>       "render_nodes": 31
>     },
>     "total_resources": 33,
>     "total_downloaded": 53
>   },
> }
>
> ```

---

> ### Author Response · Authors · 2025-11-22
> **Response to Reviewer EBQS [5/5]**
>
> ## On the Positioning of the FIGMA2CODE Benchmark
> **Weakness 3**
> > *A core concern lies in the conceptual positioning of the benchmark. Benchmarks are typically designed to evaluate the ability boundaries of large models under consistent and unconditional setups. However, in this work, the proposed benchmark incorporates conditional inputs intended to assist the model in generating UI code. This design choice seems to blend evaluation with task guidance, potentially compromising the benchmark’s intended diagnostic purpose. The paper would benefit from a clearer justification for this approach—explaining whether the goal is to measure raw capability or conditional adaptability—and from additional discussion on how this aligns with broader benchmarking principles in the LLM community.*
>
> **Answer**
>
> Thank you for the thoughtful comment.
>
> Unlike NLP-focused LLM benchmarks, FIGMA2CODE evaluates MLLMs on their ability to generate UI code conditioned on multimodal design inputs  (metadata, assets, and the rendered design), which is inherent to the real design-to-code workflow rather than optional guidance. The input setting of FIGMA2CODE is formally defined in Section 2 (Problem Formulation), where the task is specified as taking the triplet \( I = (M, A, V) \)—the Figma JSON metadata \(M\), associated assets \(A\), and the rendered screenshot \(V\). These three components together constitute the natural representation of a Figma design file and are part of the task definition rather than auxiliary guidance.
>
> The benchmarked models in Section 4.3 (Benchmarking SOTA MLLMs) are all evaluated strictly under this same multimodal input configuration. This ensures consistency across models and reflects real-world design-to-code scenarios in which both visual and structured signals are inherently available.
>
> The comparisons across input modalities in Section 4.4 (Code Generation Performance Across Modalities) are presented solely as an *analysis experiment* to study how different signals contribute to model behavior. They do not form part of the benchmark protocol and do not alter the evaluation setting established in Section 2.

---

### Author Response · Authors · 2025-11-22
**More Studies about Metrics and Human Evaluation**

We sincerely thank all reviewers for the attention and effort in the rebuttal work.

To avoid redundancy, we provide a unified clarification on metrics and human evaluation here. Our metrics fall into two major categories: visual fidelity and code quality, which are presented below.

---

> ### Author Response · Authors · 2025-11-22
> **On the Alignment Between Visual Fidelity Metrics and Human Ratings**
>
> Using visual similarity between the rendered UI and the target design as a measure of visual fidelity has been widely adopted in prior design-to-code work [1–3]. Importantly, MRWeb [4] conducted a dedicated human study comparing major visual similarity metrics with human preference judgments. Their results are reproduced below:
>
> | Metric | **Variance-Weighted Regression** CC ↑ | MAE ↓ | RMS ↓ | OR ↓ | **Non-Linear Regression** CC ↑ | MAE ↓ | RMS ↓ | OR ↓ | **Direct** SROCC ↑ |
> |--------|----------------------------------------|--------|---------|---------|----------------------------------|--------|---------|---------|----------------------|
> | **MAE**   | **0.547** | 4.10 | **1.95** | 0.049 | _0.515_ | 0.646 | 0.765 | 0.013 | **0.542** |
> | **NEMD**  | 0.469 | **3.98** | **1.95** | 0.052 | **0.532** | **0.628** | **0.752** | **0.023** | _0.508_ |
> | **PSNR**  | 0.323 | 5.69 | 2.46 | 0.000 | 0.434 | 0.679 | 0.800 | 0.016 | 0.451 |
> | **CLIP**  | 0.314 | 5.37 | 2.41 | 0.013 | 0.426 | 0.681 | 0.800 | 0.010 | 0.340 |
> | **SSIM**  | 0.305 | 5.47 | 2.42 | 0.010 | 0.381 | 0.699 | 0.817 | **0.000** | 0.218 |
> | **LPIPS** | 0.221 | 5.70 | 2.49 | 0.010 | 0.290 | 0.726 | 0.847 | 0.013 | 0.168 |
> | **Human** | — | — | — | — | — | — | — | — | **0.640** |
>
> For completeness, below we summarize the definitions of the correlation statistics reported in the human study:
>
> - **CC (Correlation Coefficient, |r|):**
>   Measures the absolute linear correlation between the metric predictions and human preference scores.
>
> - **MAE (Mean Absolute Error):**
>   Pixel-level absolute error between the rendered UI and the ground-truth design image.
>
> - **RMS (Root Mean Square Error):**
>   Pixel-level squared error; more sensitive to large deviations in local regions.
>
> - **OR (Outlier Ratio):**
>   The proportion of samples where a metric substantially deviates from human judgment under the regression model.
>
> - **SROCC (Spearman’s Rank-Order Correlation Coefficient, |ρ|):**
>   Measures how well a metric preserves the *ranking* of human preferences.
>   *The table is sorted by SROCC because it best reflects alignment with human-perceived ordering.*
>
> From these results, it is clear that **pixel-level metrics such as MAE** (Mean Absolute Error) correlate most strongly with human preference, outperforming alternatives such as **SSIM** (Structural Similarity), **LPIPS** (Learned Perceptual Image Patch Similarity), and **CLIP-based** visual encoders. Consequently, we adopt **MAE** as a core visual fidelity indicator in our benchmark.
>
> To complement low-level pixel metrics, we additionally include a **high-level embedding similarity** computed using **DINOv2** (a self-supervised vision transformer) as the encoder. This captures semantic and structural similarity that may not be reflected in raw pixel errors.
>
>
>
> ---
>
> References
>
> [1] Design2Code: Benchmarking Multimodal Code Generation for Automated Front-End Engineering
>
> [2] Web2Code: A Large-scale Webpage-to-Code Dataset and Evaluation Framework for Multimodal LLMs
>
> [3] VISION2UI: A Real-World Dataset with Layout for Code Generation from UI Designs
>
> [4] MRWeb: An Exploration of Generating Multi-Page Resource-Aware Web Code from UI Designs

---

> ### Author Response · Authors · 2025-11-25
> **Additional Studies About Alignment Between Visual Fidelity Metrics and Human Ratings in Our Benchmark**
>
> We sampled a nearly balanced subset from the outputs generated by **ERNIE 4.5 424B VL** and followed the evaluation protocol of **MRWeb** to assess the consistency between visual-fidelity metrics and human ratings. The statistics of the sampled subset are summarized below.
>
> | **Platform** | **Count** | &nbsp;&nbsp; | **Complexity** | **Count** | &nbsp;&nbsp; | **Category** | **Count** |
> |--------------|----------:|--------------|----------------|----------:|--------------|--------------|----------:|
> | mobile       | 38        |              | mid            | 24        |              | Data & Information Presentation | 12 |
> | desktop      | 30        |              | low            | 19        |              | E-Commerce & Transactions       | 11 |
> | **Total**    | **68**    |              | high           | 16        |              | Marketing & Landing Content     |  9 |
> |              |           |              | hard           |  9        |              | User Identity & Personalization |  8 |
> |              |           |              |                |           |              | Forms & Interaction Flows       |  6 |
> |              |           |              |                |           |              | Communication & Collaboration   |  5 |
> |              |           |              |                |           |              | Health & Lifestyle              |  4 |
> |              |           |              |                |           |              | Others                          |  4 |
> |              |           |              |                |           |              | Support, Guidance & Onboarding  |  3 |
> |              |           |              |                |           |              | Notifications & States          |  3 |
> |              |           |              |                |           |              | Media & Entertainment           |  2 |
> |              |           |              |                |           |              | Scheduling & Activities         |  1 |
>
>
>
> We then recruited four annotators with computer science backgrounds to compare each generated page with its corresponding design image and assign one of the following similarity labels: *Highly Dissimilar*, *Dissimilar*, *Moderately Similar*, *Similar*, and *Highly Similar*. Following MRWeb, we converted these labels into **Mean Opinion Scores (MOS)** and computed the correlation between MOS and two metrics—**MAE** and **VES** (using DINOv2 or CLIP as the encoder). The correlation results are reported below.
>
>
> | Metric | User | Variance-Weighted Regression CC | Non-Linear Regression CC | SROCC |
> |---|---|---:|---:|---:|
> | VES (DINOv2) | Annotator 1 | 0.3844 | 0.3844 | 0.4282 |
> |  | Annotator 2 | 0.4450 | 0.4155 | 0.4775 |
> |  | Annotator 3 | 0.4215 | 0.3998 | 0.4654 |
> |  | Annotator 4 | 0.3787 | 0.3787 | 0.3995 |
> |  | MOS | 0.4880 | 0.4656 | 0.5180 |
> | VES (CLIP) | Annotator 1 | 0.3549 | 0.3535 | 0.3060 |
> |  | Annotator 2 | 0.2687 | 0.2687 | 0.2219 |
> |  | Annotator 3 | 0.3485 | 0.3485 | 0.3044 |
> |  | Annotator 4 | 0.2126 | 0.0955 | 0.1978 |
> |  | MOS | 0.3547 | 0.2816 | 0.2831 |
> | MAE | Annotator 1 | 0.3177 | 0.3177 | 0.2034 |
> |  | Annotator 2 | 0.1782 | 0.1782 | 0.0543 |
> |  | Annotator 3 | 0.3550 | 0.3550 | 0.2809 |
> |  | Annotator 4 | 0.3378 | 0.3378 | 0.2724 |
> |  | MOS | 0.3560 | 0.3560 | 0.2337 |
> |Human CC  | --- | --- | --- | 0.6623 |
>
> From the results, we observe three main findings:
> - VES (DINOv2 encoder) shows the strongest correlation with human ratings among all visual metrics.
> - MAE exhibits only moderate correlation in our study, which differs from the trend reported in MRWeb.
> - Inter-annotator consistency is high, indicating that the human assessments are stable and reliable.

---

> > ### Author Response · Authors · 2025-11-25
> > **Why MAE Correlates Weakly with Human Ratings on Our Benchmark**
> >
> > To understand the weak MAE–MOS correlation in our benchmark, we first conducted short interviews with annotators and examined representative human-rated examples (Appendix Fig. 23). These analyses revealed consistent rating tendencies:
> >
> > - Annotators focus on **structural alignment**, **component presence**, and **semantic plausibility**.
> > - Widespread but low-salience background variations (e.g., slight tint or brightness shifts) have **minimal impact** on their perceived similarity.
> > - As long as the layout and key UI elements remain correct, annotators generally assign high similarity scores even if pixel-level differences exist.
> >
> > Alongside these preferences, the behavior of MAE becomes clearer. MAE computes **per-pixel intensity differences**, so even subtle global color or brightness drifts—especially across large uniform regions—can accumulate into a large error despite being barely perceptible to human observers. This pixel-level sensitivity accounts for much of the divergence.
> >
> > Consequently, MAE and MOS emphasize different aspects of similarity:
> >
> > - **MAE** captures **local intensity discrepancies** at the pixel level, independent of perceptual significance.
> > - **Human judgments** emphasize **layout fidelity and semantic coherence**.
> >
> > These findings motivate complementing MAE with a structure-aware perceptual metric such as VES (DINOv2-based embedding similarity), which reflects the higher-level visual alignment that human raters rely on. Together, the two metrics offer a more faithful approximation of UI similarity in real design-to-code scenarios.

---

> ### Author Response · Authors · 2025-12-03
> **Additional Studies about Alignment  Between Code Quality Metrics and Human Ratings in Our Benchmark**
>
> To examine how well our proposed code-quality metrics align with human judgments, we recruited three programmers to evaluate a stratified subset of generated code samples. Each annotator rated the *structural clarity* and *readability* of the code.
> The resulting inter-rater statistics and metric–MOS correlations are summarized below:
>
> | Metric | Variance-Weighted Regression CC | Non-Linear Regression CC | SROCC |
> |-------|--------|--------|--------|
> | STR (MOS) | 0.1294 | 0.1294 | 0.1927 |
> | AVU (MOS) | 0.0392 | 0.5196 | 0.4513 |
> | Human (Inter-rater) | — | — | 0.4286 |
>
> From these results, we observe that:
> - Human annotators exhibit reasonably strong agreement, indicating that the evaluation criteria are stable and well understood.
> - AVU shows the highest correlation with human judgments, suggesting that excessive arbitrary-value usage is a salient signal.

---

### Author Response · Authors · 2025-12-03
**Rebuttal Summary**

Dear SACs, ACs, and Reviewers,

We sincerely thank the reviewers for their insightful feedback and constructive suggestions, and we greatly appreciate the  SACs and ACs for their careful attention and hard work during this unusually challenging review period. We are also pleased that all reviewers provided positive assessments of our work (Reviewer NgVv had raised their score to 6 before the OpenReview leak incident).

We are grateful for the acknowledgment of our **high-quality multimodal design-to-code dataset** (Reviewers EXnw, 5rJz, and NgVv), constructed through rigorous stratified sampling and expert curation. We also appreciate the recognition of our **novel metrics** that evaluate not only visual fidelity but also code quality (Reviewers 5rJz and EXnw), as well as our **thorough and informative benchmarking and analysis** (Reviewers EBQS, 5rJz, and NgVv). Additionally, we deeply value Reviewer NgVv's description of our research as ***“a valuable contribution to the community.”***

We have addressed every reviewer’s concern and made improvements to the clarity and readability of the paper. The key efforts made in our rebuttal are:

- Additional studies on the alignment between visual fidelity metrics and human ratings in our benchmark (Reviewers EXnw, 5rJz, NgVv).
-  Additional studies on the alignment between code quality metrics and human ratings in our benchmark (Reviewers EXnw, 5rJz, NgVv).
- Additional experiments demonstrating the robustness of our evaluation protocol, showing that randomness during inference has minimal impact on the reported results (Reviewers EBQS and EXnw).
- Additional experiments on prompt variations, confirming that prompt modifications have very limited influence on MLLMs’ design-to-code performance regarding code quality (Reviewer 5rJz).

Through our rebuttal efforts, Reviewer NgVv have already raised their score. However, the incident abruptly interrupts our ongoing discussions with the reviewers (e.g., Reviewer EXnw), which are essential for improving the paper. Nevertheless, we are concerned that the score rollback may inadvertently penalize honest authors who receive higher scores based on legitimate scientific discussion. In this context, we kindly hope that the ACs and SACs take into consideration the substantial rebuttal efforts we make and the positive reviewer feedback.  **We firmly uphold academic integrity and fairness, and we oppose any misuse of leaked information, as it compromises the review process and the values of our community.**

Once again, we thank SACs, ACs, and Reviewers for their hard work and dedication.

Best regards,

The authors

---

### Meta-Review · Area_Chair_gA8z · 2025-12-28

**Summary:**

The paper introduces Figma2Code, a benchmark for multimodal design-to-code generation that uses real Figma artifacts rather than relying on screenshots alone. The authors build a base dataset of 3,055 designs and curate a 213-sample high-quality benchmark via stratified sampling and expert selection. The benchmark evaluates models not only on visual fidelity, but also on layout responsiveness and code maintainability, and the main takeaway is that while proprietary models often achieve better visual fidelity, they still lag on responsiveness and maintainability, leaving a gap to industrial-quality UI code.

Despite remaining limitations, I believe the benchmark is useful and timely, and the rebuttal meaningfully improves clarity, validation, and robustness. Reviewer sentiment appears to trend more positive after rebuttal. Thus, I recommend acceptance.

**Reviewer Concerns:**

Overall, reviewers view the benchmark as meaningful and practically motivated, but they raise several substantive concerns: 1) insufficiently transparent description of the data cleaning/metadata refinement pipeline, 2) an unclear definition of "difficulty", 3) limited human-centric validation of the proposed metric suite, and 4) questions about whether the responsiveness/maintainability metrics fully capture real cross-device rendering issues and deeper semantic/accessibility concerns. Reviewer NgVv also explicitly flagged moderate novelty and was initially leaning borderline.

In the rebuttal, the authors respond seriously and add several targeted analyses. They clarify the stratified sampling dimensions and curation process in more detail, addressing concerns about coverage/diversity. They also add robustness checks, suggesting minimal sensitivity to inference randomness. For prompt sensitivity, they report that enhanced prompts yield only modest and inconsistent improvements. Importantly, they add small-scale human studies: for visual fidelity, they report that VES (DINOv2) aligns best with human ratings while MAE correlates only moderately in their setting; for maintainability, they recruit programmers and find AVU correlates more strongly with human judgments than STR, with reasonable inter-rater agreement.

That said, these rebuttal additions only address the concerns to a certain extent. The human studies are helpful but remain limited in scale and scope relative to fully validating the entire metric suite.  The benchmark itself is also relatively small (though arguably justified by evaluation cost and long multimodal inputs), and the novelty is best characterized as moderate.

**Reviewer Scores:**

The reviewer NgVv is likely to raise the score as mentioned by the comments. The others will keep their positive scores.

---

### Decision · Program_Chairs · 2026-01-26

Accept (Poster)